# Activity in human dorsal raphe nucleus signals changes in behavioural policy

Luke Priestley[1] ✉, Ali Mahmoodi [1], William D. Reith [2],
Nima Khalighinejad [1,3] & Matthew F. S. Rushworth [1,3]

The dorsal raphe nucleus (DRN) is an important source of serotonin to the human forebrain, however there is little consensus about its behavioural function. We build on recent results from animal models to demonstrate that activity in human DRN implements changes in behavioural policy that reflect the distribution of rewards in the environment. We use a foraging-inspired behavioural task to show that human participants change their policy to pursue or reject reward opportunities as a function of the average value of opportunities in the environment. Activity in DRN—but no other neuromodulatory nucleus—signals such policy changes. Patterns of multivariate activity in dorsal anterior cingulate cortex (dACC) and anterior insular cortex (AI), meanwhile, track the relative value of reward opportunities given the average value of the environment. We therefore suggest that DRN, dACC and AI form a circuit in which dACC/AI construct representations of reward opportunities given the current context, and DRN implements changes in behavioural policy based on these representations.

What makes a reward worth pursuing? For both humans and non-human animals, these judgements depend not just on the reward itself but on the context in which the judgement is made. A quintessential example is the prey selection dilemma set forth in behavioural ecology, in which a predator must decide whether to seize a foreground opportunity available in the present or search the environment for better alternatives[1,2]. According to normative accounts of this scenario, an opportunity should only be pursued if its value meets or exceeds the opportunity cost of obtaining it.

The foraging model implies that organisms should compare each opportunity they come across with the alternative opportunities they expect to encounter in the future. While organisms are unlikely to have precise knowledge of all potential future opportunities, the history of previous opportunities is an approximation for what the future will hold. The model further argues that opportunities with the same value will evoke different behavioural policies depending on the context they arise in: for example, it is rational to pursue even an objectively mediocre opportunity in a poor environment where everything on the

horizon is similar or worse, and yet the same opportunity might be rejected in the context of a rich environment with plentiful high-value alternatives.

How does the brain adapt behavioural policies to the environment in this way? In this study, we argue for the importance of a cortico-subcortical circuit comprised of the dorsal raphe nucleus (DRN)—a brainstem nucleus that is the primary source of serotonin to the forebrain in mammals—and anterior cingulate cortex (ACC) and anterior insular cortex (AI)[3–5]. Although several studies have compared DRN activity with that of other neuromodulatory nuclei[6–9], what DRN activity specifically represents, and how its function differs from other neuromodulatory nuclei in the brainstem, midbrain, and basal forebrain, remains incompletely understood.

Our approach builds on previous studies in both human and non-human primates showing that activity in ACC and AI tracks reward-related statistics that are critical for adaptive decision-making, like the value, valence and background-rate of recent reward outcomes[10–13]. Similarly, it builds upon a rich vein of experiments in animal models

[1]Wellcome Centre for Integrative Neuroimaging, Department of Experimental Psychology, University of Oxford, Oxford, UK. [2]Sainsbury Wellcome Centre for Neural Circuits and Behaviour, University College London, London, UK. [3]These authors contributed equally: Nima Khalighinejad, Matthew F. S. Rushworth. ✉e-mail: luke.priestley@psy.ox.ac.uk

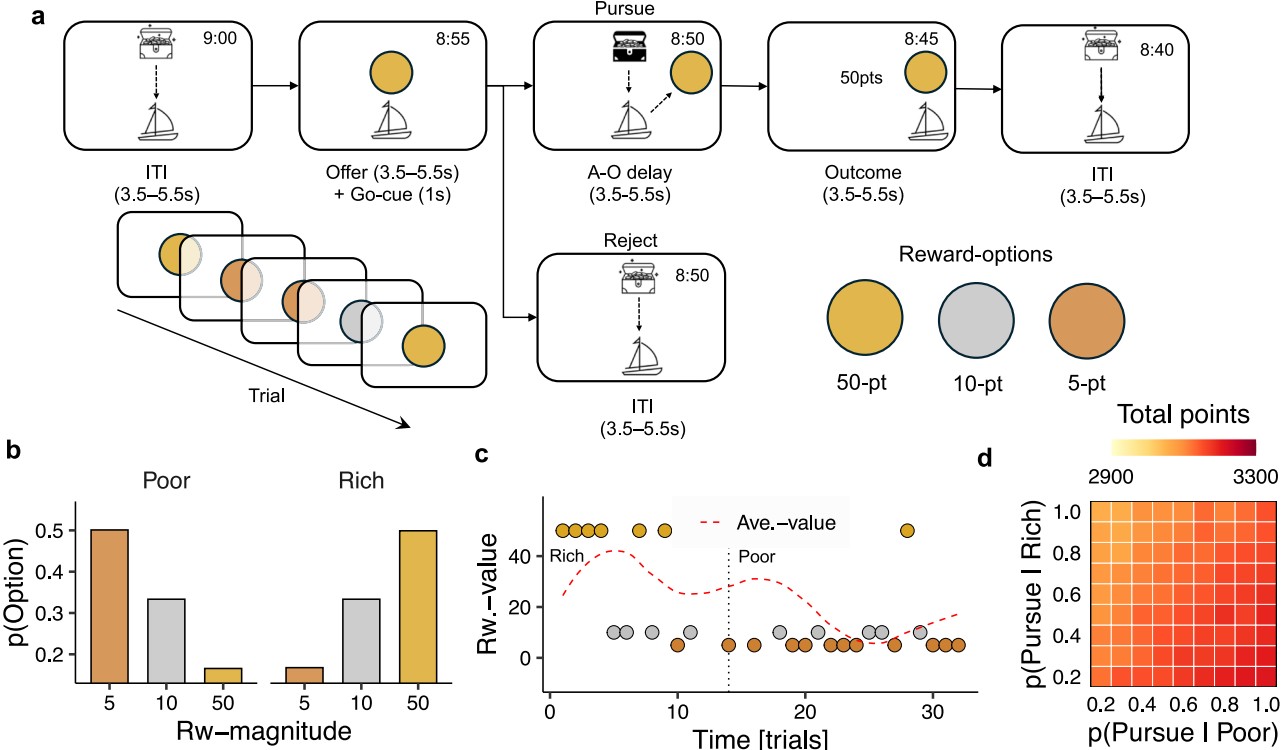

**Fig. 1 | A behavioural task for manipulating the richness of the environment.**
**a** Diagram of the behavioural task. Participants played the role of a ship's captain, deciding whether to pursue or reject sequentially encountered treasure opportunities. Reward magnitude was indicated by gold, silver and bronze medallion stimuli. **b** Relative frequency of 5-point, 10-point and 50-point reward options in rich and poor environments, respectively. **c** Example sequence of rich and poor environments illustrating key reward statistics and their evolution over time. The dashed red line indicates the mean value of rewards over the preceding five trials (i.e., $t{-}1{:}t{-}5$). **d** Points-earned as a function of pursuit rates for the 10-point option in rich (Y axis) and poor (X axis) blocks. Task parameters ensure that the optimal strategy is to pursue the 10-point option in poor environments and reject it in rich environments.

linking DRN activity to learning and decision-making. A consistent theme in these studies is that DRN activity carries information about the value of opportunities and controls the balance between opponent modes of reward-driven behaviour, like patience and impulsivity[14–18]. Here, we propose that these seemingly disparate functions may be linked by a common process of setting a behavioural policy. We employ a foraging-inspired behavioural task that allows us to identify when and how behavioural policies are set and demonstrate that activity in human DRN is associated with policy changes.

We gave participants a foraging-inspired behavioural task in which they encountered a limited set of reward options in a series of rich and poor environments. As expected, participants systematically changed their behavioural policy depending on the richness of the environment and were especially responsive to poor environments, which caused them to become less selective in their choices. We recorded brain activity while participants performed the task using an ultra-high field functional magnetic resonance imaging (fMRI) protocol specifically designed to maximise signal from brainstem and midbrain areas. These recordings demonstrated that DRN—but no other major neuromodulatory nucleus—exhibited distinctive univariate patterns of activity during behavioural-policy changes. In contrast, multivariate patterns of activity evoked by reward options in dACC and AI paralleled the environment-specific policy for each option revealed in each participant's behaviour. Taken together, these results suggest that DRN, dACC and AI form a cortico-subcortical circuit that reconciles the brain's behavioural policy with the distribution of rewards in the environment.

## Results

### A behavioural task for manipulating the richness of the environment

Twenty-seven participants performed a simple decision-making task involving sequential encounters with reward opportunities (Fig. 1a). On each trial, participants were presented with a single reward opportunity drawn from a set of three possible options with known a priori values: a low-value option (5-points; bronze medallion), a middle-value option (10-points; silver medallion), and a high-value option (50 points; gold medallion). Upon each encounter, participants could pursue opportunities by making a button-press response or reject opportunities by withholding their response and waiting for the next trial to begin (see Methods).

Pursuing decisions incurred a temporal opportunity cost equivalent to foregoing a single reward opportunity in the future. It was therefore critical to consider the prospective frequency of each reward option when deciding whether an opportunity was worth pursuing. The frequency of reward-options was systematically manipulated by dividing the task into 4.5-minute blocks of trials with different reward-option distributions (Fig. 1b). There were two types of block; (1) *rich* blocks, where the 50-point option was more frequent than the 10-point and 5-point options [*P(5-point|Rich)* = 0.16, *P(10-point|Rich)* = 0.33, *P(50-point|Rich)* = 0.50)], and; (2) *poor* blocks where the 5-point option was more frequent than the 10-point and 5-point options [*P(5-point|Poor)* = 0.50, *P(10-point|Poor)* = 0.33, *P(50-point|Poor)* = 0.16)]. These blocks allowed us to experimentally control the richness of the environment throughout the task (Fig. 1c). Importantly, the frequency

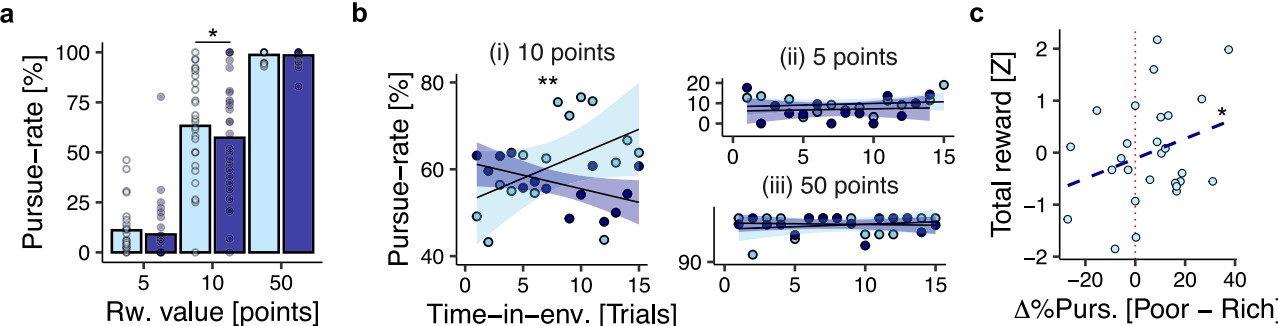

**Fig. 2 | Reward-guided decisions are modulated by the richness of the environment. a** Pursue-rates for 5-point, 10-point, and 50-point reward options as a function of environment-type. Bars indicate sample-level mean pursue-rate; dots indicate participant-level mean pursue-rates. **b** Pursue-rates for 10-point (i), 5-point (ii) and 50-point (iii) options as a function of trial-within-environment in rich and poor blocks, respectively. Lines and shadings indicate regression-line ± SEM; dots indicate sample-level mean pursuit-rate as a function of trial-within-environment. **c** Relationship between behavioural change between environments for the 10-point option (*X* axis) and total points earned during the task (*Y* axis). Dots indicate observations from individual participants. In all panels, *n* = 27; *p < 0.05, **p < 0.01, ***p < 0.001 two-tailed.

of the 10-point option was constant across both rich and poor blocks, meaning that changes in a participant's behaviour toward the 10-point option could only arise from changes in the environmental context in which it occurred.

The parameters of the task were carefully chosen to ensure that the rational task strategy was to accept the 10-point option in poor environments, and to reject it in rich environments (Fig. 1D). Environments were not signalled to participants, and we therefore did not expect participants to exhibit this pattern precisely. Nevertheless, the design ensured that there was a reward-maximising rationale for participants to change their behaviour toward the 10-point option. Accordingly, we focus predominantly on trials featuring the 10-point option in the following behavioural analyses, although the same conclusions obtain in omnibus analyses where all reward options are considered together (GLM1.1a; see supplementary Fig. S1C).

### Reward-guided decisions are modulated by the richness of the environment

To begin, we tested the hypothesis that participants were more likely to pursue the 10-point option in poor environments compared to rich environments. All analyses in this section were implemented using mixed-effect generalised linear models (GLM), which account for subject-specific variation in behaviour, and in which the dependent variable was the binary pursue-vs-reject decision on each trial.

We first tested the hypothesis by asking whether the experimentally manipulated environment type (rich vs. poor) influenced pursue vs. reject decisions. This showed that participants were, indeed, more likely to pursue the 10-point option when it occurred in a poor environment compared to a rich environment (GLM1.1b; $\beta_{\text{env.-type}} = -0.19$, SE = 0.09, $p = 0.038$; Fig. 2a). There was no evidence for similar effects in 5-point (GLM1.1b; $\beta_{\text{env.-type}} = -0.51$, SE = 0.27, $p = 0.064$; Fig. 2a) or 50-point options (GLM1.1b; $\beta_{\text{env.-type}} = -0.11$, SE = 0.24, $p = 0.655$ Fig. 2a). Analysing all options together in a single GLM indicated a two-way interaction between option-value and environment-type that was consistent with the pattern evinced by option-specific analyses (GLM1.1a $\beta_{\text{env.-type*reward-option}} = -3.06$, SE = 0.22, $p < 0.001$). Importantly, pursue rates for the 10-point option in poor environments—but not pursue rates for the 10-point option in rich environments—were positively associated with total reward accumulated in the experiment ($\beta_{\text{Poor}} = 3.94$; SE = 1.63; $p = 0.024$; $\beta_{\text{Rich}} = 1.63$; SE = 1.92; $p = 0.405$). This was complemented by a two-way interaction whereby the benefit of pursuing the 10-point option in the poor environment was inversely proportional to pursuit rates in the rich environment ($\beta_{\text{Poor*Rich}} = -5.42$; SE = 2.29; $p = 0.027$; Fig. 2c). The degree to which participants changed their behaviour between environments

determined their level of performance on the task, as envisaged in the experimental design.

Participants were not informed that there were distinct rich and poor environments, and they received no cues to this effect during the task. Environment-driven changes in behaviour, therefore, involved tracking the frequency of reward options over time. To investigate whether the environment effect emerged over time in this way, we tested the two-way interaction between environment type and time elapsed within an environment, where time was quantified in trials. This revealed that environment-driven changes in behaviour increased as a function of time for the 10-point option (GLM1.2 $\beta_{\text{env.-type*trial-in-env.}} = -0.20$, SE = 0.06, $p = 0.001$; Fig. 2b), but that there were no such changes for the 5-point (GLM1.2; $\beta_{\text{env.-type*trial-in-env.}} = -0.06$, SE = 0.12, $p = 0.629$; Fig. 2b) and 50-point options (GLM1.2; $\beta_{\text{env.-type*trial-in-env.}} = -0.36$, SE = 0.25, $p = 0.139$; Fig. 2b). Simple main-effects analysis for the 10-point option indicated an asymmetric pattern of responsiveness to environments, whereby participants were more likely to pursue the 10-point option as time elapsed in poor environments (GLM1.2; $\beta_{\text{trial-in-env.}} = 0.28$, SE = 0.08, $p < 0.001$), but were not more likely to reject the 10-point option over time in rich environments (GLM1.2; $\beta_{\text{trial-in-env.}} = -0.14$, SE = 0.09, $p = 0.151$). This pattern of results was confirmed by a conceptually similar analysis in which the environment was operationalised as the average value of recently encountered reward-options, as distinct from the experimentally defined environment types (GLM1.3; see supplementary Fig. S1D). Taken together, these analyses show that adaptive changes in behaviour toward the 10-point option emerged gradually within each block, indicating that participants required time and experience to ascertain the richness of the present context.

### Option-specific policies are reconciled with the richness of the environment

Having shown that pursue-vs-reject decisions were influenced by the richness of the environment, we performed a series of complementary analyses focussed on the temporal organisation of behaviour. Because the task featured only three reward options, we reasoned that participants would form a behavioural policy for each option—that is, an option-specific pursue-vs-reject strategy that remained consistent over successive encounters with the option in question (Fig. 3a). We validated this hypothesis by testing whether pursue-vs-reject decisions were autocorrelated across encounters with the same reward option. This confirmed that participants tended to repeat their decisions across successive encounters, consistent with a temporally persistent behavioural policy (GLM2.1; $\beta_{\text{prev.-policy}} = 1.53$, SE = 0.28, $p < 0.001$; Fig. 3b). Importantly, when the pursue-vs-reject decision on the

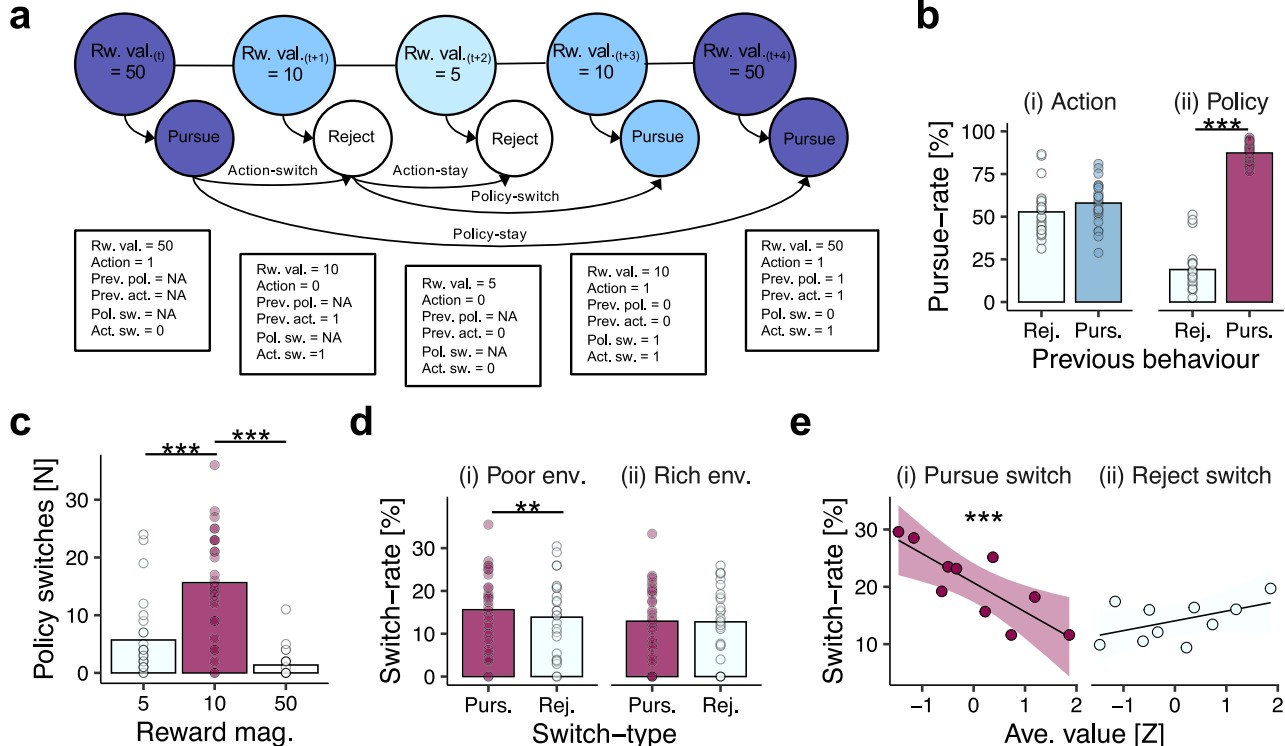

**Fig. 3 | Changes in behavioural policy are modulated by the richness of the environment. a** Example sequence of trials and operationalisation of policy-related variables. Behaviour was characterised along two dimensions: (1) a *policy* dimension reflecting option-specific decisions to pursue-vs-reject the three available reward-options, and (2) an *action* dimension reflecting non-option-specific decisions to pursue-vs-reject an option regardless of the reward option available. Trials were then categorised as (i) *policy-stay* events if the pursue-vs-reject decision was consistent with the previous encounter, or (ii) *policy-switch* events if the pursue-vs-reject decision was different from the previous encounter. Similarly, trials were categorised as (i) *action-stay* if the pursue-vs-reject decision was the same as the preceding trial (regardless of reward option), and (ii) *action-switch* if the pursue-vs-reject decision changed. **b** Pursue-rates as a function of (i) previous action and (ii) previous policy. Bars indicate sample-mean pursuit-rate; dots indicate participant-level pursuit-rates. **c** Policy-switches frequency as a function of reward-option. Bars indicate the mean count of policy-switches per experimental session at the sample-level; dots indicate participant-specific counts. **d** Switch-rate for pursue-switches and reject-switches as a function of environment type. Data shown for the 10-point option only. Bars indicate sample-level mean switch-rate, dots indicate participant-level switch-rate. **e** Probability of (i) pursue-switches and (ii) reject-switches as a function of the average value of recent reward-options. Data shown for the 10-point option only. Lines and shadings indicate the regression line ± SEM. Points and error bars indicate mean ± SEM switch-rates in decile bins of average value. In all panels, $n = 27$; *$p < 0.05$, **$p < 0.01$, ***$p < 0.001$ two-tailed.

preceding trial concerned a different option, it did not influence the pursue-reject decision on the current trial, suggesting that policies were option-specific (GLM2.2; $\beta_{\text{prev.-action}} = 0.05$, SE = 0.17, $p = 0.737$; Fig. 3b; see also Fig. 3a for visual explanation of distinction between switching in relation to a previous-action and switching in relation to a previous-policy).

Adopting the behavioural-policy perspective enabled us to precisely identify changes in behavioural policy (policy-switch, hence). We operationalised policy-switches as trials on which the pursue-vs-reject decision diverged from the policy exhibited the previous time a reward-option was encountered (Fig. 3a). Consistent with earlier analyses showing behavioural changes for the 10-point option, policy-switches were more frequent for the 10-point option than both its 5-point (GLM2.3; $\beta_{\text{middle-vs-low}} = 1.51$, SE = 0.38, $p < 0.001$; Fig. 3c) and 50-point counterparts (GLM2.3; $\beta_{\text{middle-vs-high}} = 3.63$, SE = 0.67, $p < 0.001$; Fig. 3c). Subsequent analysis of policy-switches therefore focused on the 10-point option trials (see supplementary Fig. S2 for analysis of 5-point and 50-point options).

Given our analysis of pursue-vs-reject decisions, we predicted that policy switches would be influenced by the richness of the environment. To test this, we categorised policy-switches into two types depending on the direction of change: (1) *pursue-switches*, when the reward-option was pursued on the current trial after being rejected on the previous encounter, and (2) *reject-switches*, when the reward-

option was rejected on the current trial after being pursued on the previous encounter. Insofar as participants approximated the optimal task strategy (Fig. 1d), we expected that pursue-switches would be more likely in poor environments and reject changes would be more likely in rich environments.

When pursuit-switches were analysed in relation to the experimentally defined environments, they followed the expected pattern: in poor environments, pursue-switches were more likely to occur than reject-switches (GLM2.4a; $\beta_{\text{pursue-switch vs reject-switch}} = -1.38$, SE = 0.52, $p = 0.008$; Fig. 3d), and pursue-switches in poor environments were also more likely to occur than pursue-switches in rich environments (GLM2.5a; $\beta_{\text{rich-vs-poor}} = -0.36$, SE = 0.12, $p = 0.002$). The likelihood of reject-switches, however, did not differ across environment types (GLM2.4b, $\beta_{\text{pursue-switch vs reject-switch}} = -0.69$, SE = 0.57, $p = 0.227$, Fig. 3d; GLM2.5b, $\beta_{\text{rich-vs-poor}} = 0.05$, SE = 0.11, $p = 0.651$). A similar pattern was obtained when policy-switches were compared against the average value of recent reward-options: pursue-switches were more likely as average value decreased (GLM2.6a; $\beta_{\text{ave.-value}} = -0.76$, SE = 0.14, $p < 0.001$; Fig. 2e), but there was no evidence for an effect of reject-switches, which occurred at an approximately constant rate (GLM2.6b; $\beta_{\text{ave.-value}} = 0.13$, SE = 0.15, $p = 0.386$; Fig. 3e; see supplementary Fig. S2 for equivalent analysis of 5-point and 50-point options). In tandem with previous analyses, these results suggest that changes in behavioural policy approximated the rational task

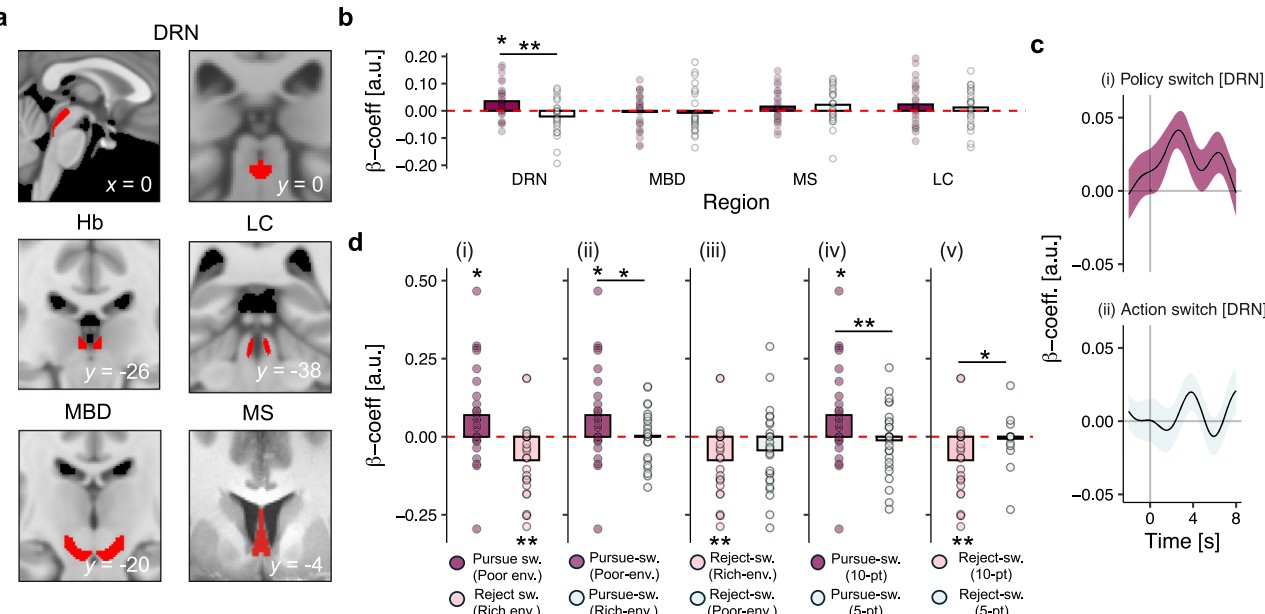

**Fig. 4 | Brain activity in dorsal raphe nucleus represents environment-driven changes in behavioural policy. a** Anatomical regions of interest (ROIs) used in the fMRI analysis, including dorsal raphe nucleus (DRN); habenula (Hb); locus coeruleus (LC); substantia nigra and ventral tegmental area (MBD), and medial septal nucleus (MS). **b** Regression-weights for the effect of policy-switch and action-switch events in subcortical ROIs. **c** Timecourse of the effect size for (i) policy-switch events, and (ii) action-switch events. Time = 0 (x axis) indicates the onset of the reward-opportunity stimulus. Note that due to haemodynamic delay, policy-switch related activity in DRN likely occurs in ITI periods. **d** Regression-weights for the effect of policy-switch events on DRN activity in control analyses. (i) Effect of policy switches that are congruent with the environment—i.e., pursue-switches in poor environments and reject-switches in rich environments. (ii) Effect of pursue-switches in poor environments compared to pursue switches in rich environments; (iii) Effect of reject-switches in rich environments compared to reject-switches in poor environments. (iv) Effect of pursue-switches for the 10-point option compared to pursue-switches for the 5-point option in poor environments; (v) Effect of reject-switches for the 10-point option compared to reject-switches for the 5-point option in rich environments. **c** Lines and shadings indicate mean ± SE of regression-weight across participants. **b**, **d** Bars show sample-level mean peak regression weight; dots indicate participant-level peak regression weights. In all panels, $n = 27$; $*p < 0.05$, $**p < 0.01$, $***p < 0.001$ two-tailed.

strategy and that this pattern was more pronounced in poor environments.

## DRN represents environment-driven policy switches

Participants performed the behavioural task while undergoing ultra-high field (7 T) fMRI recordings of blood oxygen level dependent (BOLD) signal. We used these recordings to investigate brain activity that represented the task environment, and environment-driven changes in behaviour. We focused on a priori regions of interest (ROIs) comprising the ascending neuromodulatory systems (ANS): an assemblage of phylogenetically ancient nuclei in the basal forebrain, midbrain, and brainstem that release fundamental neuromodulatory chemicals via diffuse, whole-brain connections[19–21]. According to a prominent theory, these nuclei broadcast salient low-dimensional signals that orchestrate activity across multiple brain regions[19]. These properties make the ANS ideal candidates for implementing changes in behavioural policy.

We extracted BOLD signal from an anatomical ROI for the DRN, in addition to a combined ROI covering the substantia nigra and ventral tegmental area (midbrain dopaminergic nuclei; MBD), the locus coeruleus (LC), the medial septal nucleus (MS), and the habenula—an epithalamic nucleus characterised by reciprocal interactions with the nuclei comprising the ANS and carrying related activity patterns[22–25]; see Fig. 4a for anatomical illustration of all ROIs and Supplementary Fig. S3 and Methods for details of their construction). Although our key hypotheses concerned DRN, expanding our purview allowed us to compare DRN activity with patterns in other ANS nuclei, which is critical for identifying the specialised function of different brain regions[26].

We first examined brain activity related to policy switches. We did this by contrasting brain activity on policy-switch trials with brain activity on policy-stay trials—i.e., we asked whether specific patterns of brain activity occurred when a policy-switch was exhibited in behaviour. This indicated that activity in DRN—but no other subcortical ROI—represented policy-switch trials (GLM4.1a; $t_{DRN; policy-switch}(26) = 2.84$, $p = 0.043$; Fig. 4b and supplementary Fig. S5A). The DRN effect was significantly different from the effects in MBD ($t_{DRN-vs-MBD; policy-switch}(26) = 3.20$, $p = 0.003$), and habenula ($t_{DRN-vs-Hb; policy-switch}(26) = 2.69$, $p = 0.012$). Although there was no evidence that the effect of policy switches in DRN was, in general, different from LC ($t_{DRN-vs-LC; policy-switch}(26) = 0.99$, $p = 0.330$), environment-driven policy switches (see next paragraph) were, indeed, represented differently in DRN activity compared to LC activity ($t_{DRN-vs-LC; congruent-poor}(26) = 4.79$, $p < 0.001$, $t_{DRN-vs-LC; congruent-rich}(26) = -2.97$, $p = 0.006$). Next, we confirmed that this effect was specific to behavioural-policy switches by testing DRN's relationship with other forms of behavioural change, like action-switches where both the pursue-vs-reject decision and option-identity changed across consecutive trials such that the change in behaviour did not constitute change in option-specific policy (Fig. 3a). There was no evidence for an effect of action switches on DRN activity (GLM4.1b; $t_{DRN; action-switch}(26) = -1.66$, $p = 0.489$; Fig. 4b, c), and the policy-switch representation reported earlier was, moreover, significantly different to the null action-switch signal ($t_{DRN; action-switch vs policy-switch}(26) = 3.53$, $p = 0.002$; Fig. 4b, c). This suggests that the observed activity in DRN is related to sustained changes in policy, but not to rapid changes in action (see also Fig. S5C, D).

Our analysis of behaviour showed that policy switches were driven by the richness of the environment (Fig. 3d, e). Because brain activity in DRN represented policy-switches in general, we therefore next tested whether DRN represented environment-driven policy-switches specifically. We examined DRN activity during policy-switches for the 10-point option that were congruent with the environment—i.e., pursue-

switches in poor environments, and reject-switches in rich environments. DRN activity correlated with both congruent pursue-switches and congruent reject-switches, albeit with oppositely signed patterns of activity (GLM4.2a, $t_{DRN;\ congr.-pursue}(26) = 2.57$, $p = 0.016$; GLM4.2c $t_{DRN;\ congr.-reject}(26) = -3.24$, $p = 0.003$; Fig. 4d, i). Because pursue-switches were correlated with action and reject-switches were correlated with inaction, we ensured that these effects were not artefacts of motor preparation by comparing DRN activity for congruent vs. incongruent pursue-switches and congruent vs. incongruent reject-switches: in other words, we compared events with the same motor-output that occurred in different environmental contexts. Unlike congruent-pursue changes (GLM4.2a, $t_{DRN;\ congr.\ pursue}(26) = 2.57$, $p = 0.016$; Fig. 4d-ii), there was no evidence for an effect of incongruent-pursue changes (GLM4.2b, $t_{DRN;\ incongr.-pursue}(26) = 0.19$, $p = 0.849$; Fig. 4d-ii) on DRN activity, and the congruent pursue-switch signal was stronger than its incongruent-pursue counterpart ($t_{DRN;\ congr.-vs-incongr.\ pursue}(26) = 2.13$, $p = 0.042$; Fig. 4d-ii). Similarly, there was no evidence for an effect of incongruent-reject changes on DRN activity (GLM4.2 d, $t_{DRN;\ incongr.-reject}(26) = -1.78$, $p = 0.090$; Fig. 4d-iii) in contrast to the significant effect of congruent-reject changes (GLM4.2c, $t_{DRN;\ congr.-reject}(26)) = -3.24$, $p = 0.003$; Fig. 4d-iii), although the difference between these patterns was not significant ($t_{DRN;\ congr.-vs-incongr.\ reject}(26) = -0.92$, $p = 0.365$; Fig. 4d-iii). In this way, DRN policy-switch representations paralleled behavioural adjustment to the environment (Fig. 2e): they were only significant during switches to pursuit and switches to rejection that were appropriate for the richness of the environment, and this difference was especially prominent for pursue-switches engendered by poor environments (Fig. 3d).

As a further test of DRN policy-switch signals, we compared congruent policy-switches for 10-point and 5-point options. Behavioural analysis (Fig. 3d, e and supplementary Fig. S2) demonstrated that policy-switches for the 10-point option were modulated by the richness of the environment, whereas policy-switches for the 5-point option were not and instead appeared to be random or exploratory in nature. Comparing the two events, therefore, allowed us to isolate policy-switches driven by the richness of the environment from behavioural switches with other origins. Unlike the significant effect of congruent 10-point option pursue-switches (GLM4.2a, $t_{DRN;\ congr.-pursue}(26) = 2.57$, $p = 0.016$; Fig. 4d-iv), there was no evidence for an effect of 5-point option pursue-switches in poor environments on DRN activity (GLM4.2a, $t_{DRN;\ 5-pt\ pursue-switch}(26) = -0.55$, $p = 0.581$; Fig. 4d-iv). These signals were significantly different from one another ($t_{DRN;\ low-vs-middle\ pursue}(26) = 2.42$, $p = 0.022$; Fig. 4d-iv). Similarly, there was no evidence for an effect of reject-switches for the 5-point option in rich environments on DRN activity (GLM4.2c, $t_{DRN;\ low-option\ reject}(26) = -0.55$, $p = 0.588$; Fig. 4d-v) in contrast to the significant effect of the 10-point option reject-switches (GLM4.2c $t_{DRN;\ congr.-reject}(26) = -3.24$, $p = 0.003$; Fig. 4d-v), and the difference between signals was significant ($t_{DRN;\ low-vs-middle\ reject}(26) = -2.54$, $p = 0.017$; Fig. 4d-v). Furthermore, we found that representations of policy-switching emerged only in late time windows after a block onset (supplementary Fig. S5B) in line with the time taken for participants to learn about the value of the environment (see Fig. 2b). This provides further evidence DRN activity was linked to specifically environment-driven policy-switches.

It is important to note that the absence of policy-switch-related activity in other ANS nuclei was not due to insensitivity in the fMRI signal recorded from these regions. For example, we observed patterns of MDB activity that are strikingly consistent with its role in reward-guided behaviour in previous studies: (i) a pattern related to the pursue-vs-reject decision on each trial, reminiscent of a reward-guided action initiation signal[27,28], and; (ii) a pattern conveying the value-difference between the current reward opportunity and the richness of the environment, akin to a reward prediction-error[29–32] (see supplementary Fig. S6).

Similarly, we found that in contrast to the representation of environment-driven policy-switches in DRN activity, policy-switches that were not appropriate given the richness of the environment were represented in MS activity (GLM S7.1; $t_{MS;\ incongr.-switch}(26) = 2.80$, $p = 0.047$). Such exploratory policy-switches were not represented in DRN activity, and the difference between environment-driven DRN and exploratory-driven MS patterns was, moreover, statistically significant (GLM S7.2; $ANOVA_{ROI*change-type} = 10.39$, $p = 0.001$; see supplementary Fig. S7). This is broadly compatible with the link between acetylcholine and forms of exploratory behaviour, in which an organism deviates from an established reward-maximising policy[26,33]. Taken together, these analyses suggest that activity in DRN−but no other ANS nucleus −represented changes in behavioural policy that aligned behaviour with the environment. The policy-switch effect was specifically linked to environment-driven changes in behaviour and was consistently more prominent for policy-switches engendered by poor environments. In this way, DRN policy-switch representations paralleled behavioural sensitivity to poor environments (Figs. 4di-v and 3d, e).

## Cortical multivariate option representations are modulated by the richness of the environment

Having shown that DRN activity represented environment-driven policy-switches, we expanded our fMRI analysis to cortical areas that might represent the reward options themselves. It has been shown that the brain constructs abstract representational spaces in which options can be compared along multiple dimensions depending on their current behavioural relevance[34–37]. Our behavioural data indicated that participants adaptively changed their behaviour toward the middle-value option depending on the richness of the environment. We therefore hypothesised that in regions encoding a representation of each option, the distance between option-specific representations would change in tandem with the option-specific policy adopted in each environmental context (Fig. 5a). We tested this using representational similarity analysis (RSA) on a series of ROIs derived from functional fMRI analyses[38]. First, we searched for relevant ROIs in our own data using whole-brain fMRI analysis testing for regions that coded: (i) the pursue-vs-reject decision made on each trial, and (ii) the reward magnitude of the opportunity on each trial (GLM3.1 & GLM3.2; regressors time-locked to the time of the opportunity onset; see supplementary Fig. S4). This identified clusters of activity related to the pursue-vs-reject contrast in anterior insular cortex (AI), dorsal anterior cingulate cortex (dACC), and lateral frontopolar cortex (lFPC), and a cluster in subgenual anterior cingulate cortex (sgACC) related to the reward-magnitude contrast (Fig. 5b; see supplementary Fig. S8 and Tables 1 and 2 for full results of whole-brain analyses; see Methods for details of ROI construction). In addition, we investigated ROIs in dorsomedial prefrontal cortex (dmPFC) and pregenual anterior cingulate cortex (pgACC) based on activations reported in previous studies[39]; Fig. 5b).

Our analysis focussed on four representational distances; (i) $d_1$, the distance between 5-point and 10-point options in poor environments; (ii) $d_2$, the distance between 10 and 50-point options in poor environments; (iii) $d_3$, the distance between 5-point and 10-point options in rich environments, and; (iv) $d_4$, the distance between 10-point and 50-point options in rich environments. In each case, the distance was computed as the cosine dissimilarity between the pairs (1-cosine angle between the two offer values). Note that ROIs were identified using univariate changes in BOLD signal at the time of offer presentation, which was temporally decorrelated from behavioural responses. Insofar as the neural representation of reward-options tracked their relative positions in the task space given the environment, we predicted that distances would follow patterns where either: (1) H1: $d_1 - d_3 > 0$−i.e., that the 10-point option's representation was closer to the 5-point in rich environments compared to poor environments, or; (2) H2: $d_4 - d_2 > 0$−i.e., that the 10-point option's

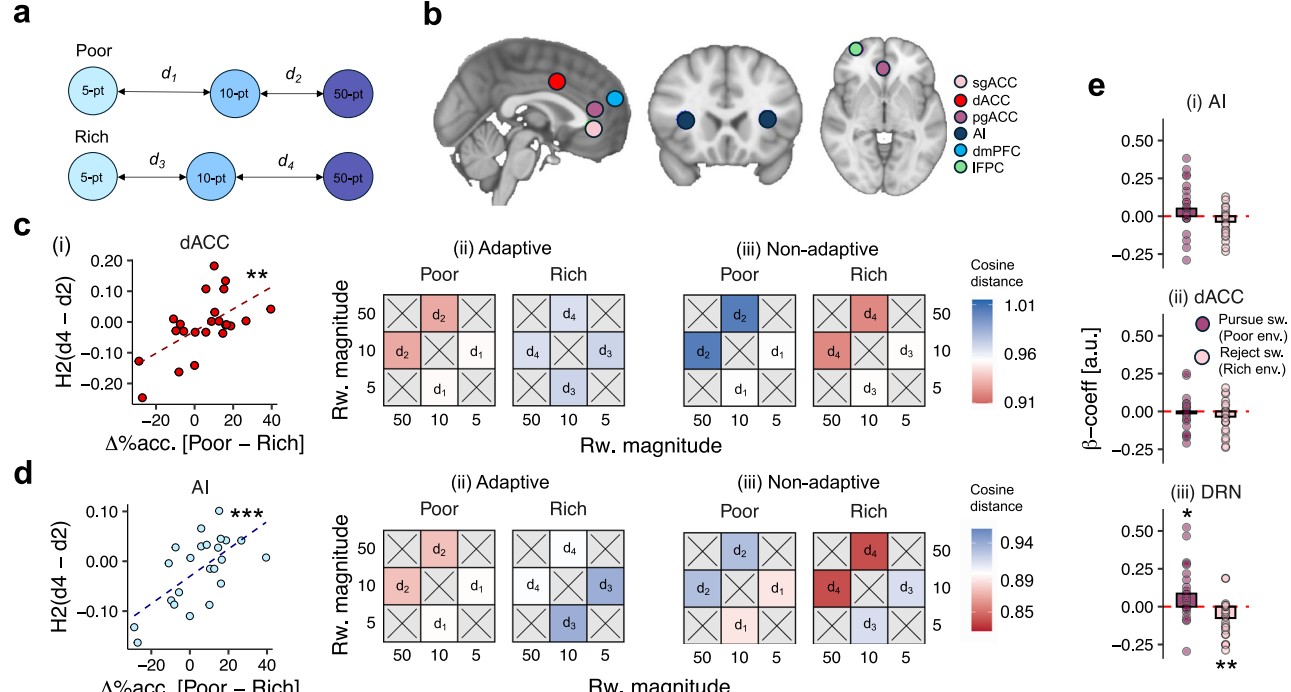

**Fig. 5 | Multivariate representations of value are modulated by the environment in dorsal anterior cingulate cortex and anterior insular cortex. a** Diagram of key representational distances in the representational similarity analysis (RSA). **b** Cortical regions of interest (ROI) used in RSA included subgenual anterior cingulate cortex (sgACC), dorsal anterior cingulate cortex (dACC), pregenual anterior cingulate cortex (pgACC), anterior insular cortex (AI), dorsomedial prefrontal cortex (dmPFC) and lateral frontopolar cortex (lFPC). Colour scale corresponds to ROI identity. **c** RSA results in dACC. (i) Correlation between $d_4-d_2$ and behavioural change between environments. Points indicate participant-level observations; lines indicate line-of-best-fit. (ii) Representational distances between reward options as a

function of environment type in (1) adaptive participants who changed their behaviour toward the 10-point option between environments, and (2) non-adaptive participants who did not. **d** RSA results in AI. (i) Correlation between $d_4-d_2$ and behavioural change between environments. Points indicate participant-level observations; lines indicate line-of-best-fit. (ii) Representational distances between reward options as a function of environment-type in adaptive and non-adaptive participants (c.f. c-ii). **e** Regression-weights for effect of congruent policy-switches in (i) AI, (ii) dACC and (iii) DRN. Bars indicate sample-level mean peak regression weight; dots indicate participant-level peak regression weights. In all panels, $n = 27$; $*p < 0.05$, $**p < 0.01$, $***p < 0.001$ two-tailed.

representation was closer to the 50-point option in poor environments compared to rich environments (Fig. 5a). Importantly, not all participants followed the rational strategy of pursuing the 10-point option more frequently in poor environments (Fig. 2a, b). We therefore anticipated that the magnitude of representational distance changes would be correlated with participant-specific levels of behavioural change between environments.

Comparing neural representations with behaviour demonstrated a striking effect in dACC and AI whereby the magnitude of H2 ($d_4-d_2$) was positively correlated with behavioural change between environments ($r_{ACC} = 0.60$, $t(21) = 3.46$, $p = 0.002$; $r_{AI} = 0.65$, $t(21) = 3.90$, $p < 0.001$; Fig. 5c, d; correlation statistics reported after familywise error control using Bonferroni-Holm method)—in other words, in participants who adopted the rational strategy, neural representations of the 10-point option were closer to the 50-point option in poor environments than they were in rich environments. We replicated these findings using Pearson's correlation rather than cosine dissimilarity, suggesting that the correlation holds regardless of the similarity metric used (Fig. S9). We confirmed this interpretation by contrasting $d_4-d_2$ in participants who were sensitive and insensitive to task environments, which indicated that the $d_4-d_2$ difference was greater in the environment-sensitive group than in the insensitive group ($t_{ACC}(11) = 3.54$, $p = 0.003$; $t_{AI}(11) = 3.31$, $p = 0.003$; Fig. 5c, d). There was no evidence of these patterns in any other ROI. In addition, H1—the representational distance between 10-point and 5-point options—was not correlated with behaviour in dACC or AI ($t_{dACC}(21) = -1.12$, $p = 0.271$; $t_{AI}(21) = -0.45$, $p = 0.650$), nor in any other ROI. Similarly, there was no evidence that the distance between neural

representations of the reward options changed between environments when behaviour was not taken into account (see supplementary Fig. S9). This suggests that the degree of environment-related change in participants' neural representations of the options was linked to their ability to identify the rational task strategy.

Having shown that multivariate patterns of activity in dACC and AI captured critical aspects of the task, we conducted univariate time-course analysis in these regions to relate them to subcortical ROIs. In contrast to the clear evidence of multivariate pattern change, there was no evidence of univariate activity changes in dACC or AI as a function of experimentally manipulated environment-type (GLM4.4; $t_{dACC;\ env.-type}(26) = 1.52$, $p = 0.282$; $t_{AI;\ env.-type}(26) = 0.48$, $p = 0.636$). Similarly, there was very limited evidence that dACC or AI represented environment-driven policy-switches in the manner of DRN (GLM4.2a and 4.2c; $t_{dACC;\ congr.-pursue}(26) = -1.00$, $p = 0.325$; $t_{dACC;\ congr.-reject}(26) = -2.28$, $p = 0.061$; $t_{AI;\ congr.-pursue}(26) = 1.71$, $p = 0.196$; $t_{AI;\ congr.-reject}(26) = -2.11$, $p = 0.061$; Fig. 5e). Indeed, directly contrasting the latter effects indicated that the environment- and congruence-specificity of policy-switch representations was stronger in DRN compared to cortical ROIs (GLM4.5; $t_{DRN-vs-AI}(26) = 4.02$, $p < 0.001$; $t_{DRN-vs-dACC}(26) = 3.22$, $p = 0.003$), consistent with the hypothesis that DRN implemented changes in behavioural policy.

In combination with earlier analyses of DRN activity, we therefore reasoned that DRN and dACC/AI might constitute a circuit in which cortical components constructed an environment-specific representation of each option, while DRN implemented the changes in behavioural policy consequent on these environment-specific changes. We tested this hypothesis with a series of PPI analyses

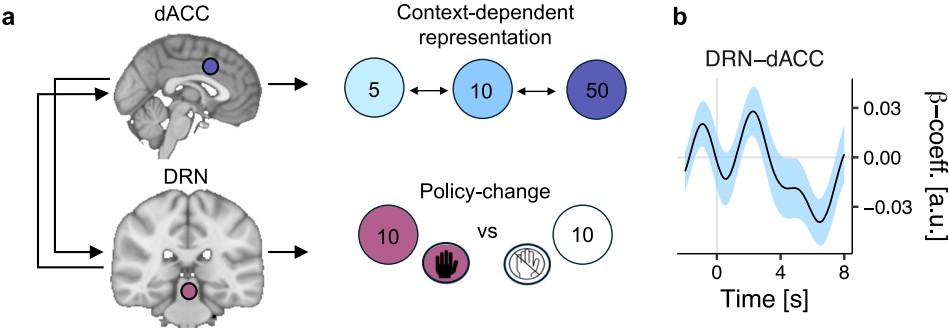

**Fig. 6 | A cortico-subcortical circuit for matching behavioural policies to the environment. a** Diagram of proposed circuit interactions between the cortex and dorsal raphe nucleus (DRN) during reward-guided decision-making. In tandem with previous studies, we propose that regions of cortex like the dorsal anterior cingulate cortex (dACC) and anterior insula (AI) compute the relative value of reward options given the environment, and the DRN implements and/or facilitates changes in behavioural policy arising from relative value computations.
**b** Psychophysiological interaction between dACC and DRN as a function of environment-type. Line and shading indicate mean ± SE of regression weight across participants; $n = 27$; *$p < 0.05$, **$p < 0.01$, ***$p < 0.001$ two-tailed.

quantifying changes in DRN−dACC/AI connectivity that might reflect communication between regions that varied as a function of the environment. Consistent with this hypothesis, there was a PPI between dACC and DRN whereby coactivation between DRN and dACC was stronger in poor environments, in which environment-driven behavioural changes were most prominent ($t_{PPI; dACC-DRN}(23) = -2.46$, $p = 0.021$; GLM4.3; Fig. 6b). There was no evidence of the same pattern between DRN and AI ($t_{PPI; AI-DRN}(23) = -1.18$, $p = 0.248$; GLM4.3).

## Discussion

Normative theories of reward-guided behaviour suggest that reward opportunities should be evaluated relative to the context they occur in ref. 40. Here, we show that human behaviour obeys this principle. We gave participants a foraging-inspired behavioural task involving repeated encounters with the same set of reward options and found that participants were, *ceteris paribus*, more likely to pursue opportunities in poor environments with few high-value alternatives. fMRI recordings of brain activity linked successful task performance to a distributed neural circuit comprising DRN in the brainstem, and dACC and AI in the cerebral cortex. Within this group, activity in DRN− but no other subcortical nucleus investigated−signalled changes in option-specific behavioural policies, but only insofar as those switches were appropriate to the richness of the environment. By contrast, elsewhere in the neuro-modulatory system, activity in MS (supplementary Fig. S7), was linked to other types of behavioural change reminiscent of exploratory switches while activity in MDB was linked to action initiation, and to prediction-error-like responses.

Within the DRN-dACC-AI circuit, multivariate activity in dACC and AI paralleled environment-driven changes in behaviour for each option; in other words, dACC and AI represented options in a context-dependent manner. Co-activation between DRN and cortical regions like dACC changed as a function of the environment, and was particularly strong during poor environments where changes in behaviour and brain activity were most prominent. We therefore suggest that DRN, dACC and AI constitute a circuit for reconciling behaviour with the richness of the environment−a circuit in which cortical components maintain context-specific representations of each choice option, and DRN implements changes in behavioural policy that reflect the current context (Fig. 6).

The present work adds to a growing corpus of experiments inspired by behavioural ecology[11,28,41–44]. These studies have shown that human participants are keenly sensitive to background reward-rates during foraging-style scenarios, where their behaviour approximates the predictions of optimal foraging theory. Here, we provide further evidence to this effect by showing that participants pursued a moderately valuable reward option−the 10-point option−more frequently in poor environments compared to rich environments. A key feature of our task design was that participants repeatedly encountered the same reward options many times over, and that changes in the environment were engendered by simply manipulating the frequency of each individual option. This meant that we could explicitly track option-specific behavioural policies and their relationship with the richness of the environment. Policy switches were consistent with the behavioural patterns described above, whereby participants switched toward pursuing the 10-point option in poor environments.

Pinpointing the trials when policy switches occurred allowed us to identify changes in DRN activity that signalled these switches. This pattern was specific to DRN and specific to switches that were appropriate to the richness of the environment. Together with previous studies, this suggests that DRN and serotonin have a conserved role in controlling switches between fundamental aspects of motivated behaviour. For example, Marques and colleagues identified a population of dorsal raphe neurons in zebrafish in which activity covaried with transitions between food-oriented foraging and exploration[45]; Priestley and colleagues identified DRN activity in the macaque during transitions between periods of engagement and disengagement with reward-oriented behaviour[18]; Trier and colleagues linked activity changes in DRN to transitions between reward-oriented behaviour and threat-oriented behaviour[46]; finally, Ahmadlou and colleagues linked the median raphe nucleus−an adjacent nucleus with prominent serotonergic projections−to controlling the balance between persisting with the current behaviour or switching to an exploratory mode[47]. The aspect of behavioural policy investigated in the present study−whether to accept a reward opportunity of a given magnitude−was comparatively simple and specific, but has nevertheless been a fundamental concern for foraging organisms throughout evolutionary history.

In the present results, changes in behavioural policy were linked to the task's underlying reward rate. This is consistent with recordings of single-neuron activity implicating DRN in environment-driven changes in behaviour, including changes between patience and impulsivity during intertemporal choice[17,48], exploitation and exploration during reversal learning[16,49–51]. Moreover, it complements a string of recent studies indicating that the reward and/or changes in the reward rate are fundamental determinants of DRN activity[13,52,53]. The implications of the background reward rate for adaptive behaviour have long been emphasised in behavioural ecology, and the present

study was indeed inspired by the so-called diet-selection task—a canonical ecology scenario which involves deciding which food items are worth pursuing and which are not[54]. In a previous experiment, Lottem and colleagues drew inspiration from a different foraging paradigm called the patch leaving problem, in which animals must decide how long to persist with food-seeking at one location given the possibility of foraging elsewhere[16]. Optogenetic stimulation of DRN neurons in this task promoted active foraging in the current patch for further potential rewards, similar to the way that DRN activity increased in the present study when participants actively sampled a moderate-value option that was worthwhile only in poor environments. Taken together, these observations emphasise that DRN activity's behavioural effect is not reducible to behavioural inhibition, as posited in classic theories of serotonin function[55]. Instead, they cohere with an emerging body of work in which DRN and adjacent raphe nuclei control changes along fundamental dimensions of behavioural policy[18,45–47].

Our results complement recent computational work demonstrating that dynamic neuromodulatory mechanisms—specifically opponent interactions in dopaminergic circuits—support flexible policy adaptation in response to environmental reward structure[56]. Here, we suggest that serotonergic systems may play an analogous role in implementing such adaptive policy switches in the human brain. Similarly, our results are broadly in line with Ligneul and Mainen's hypothesis that DRN mediates behavioural adaptation to changing environments[57]. Ligneul and Mainen emphasise the role of prediction errors in this function and we, also, have reported prediction error-like activity patterns in DRN in our previous work[46]. Nevertheless, the present results suggest that this is only one aspect of DRN function. The moderate-value options to which DRN responded in the current study always occurred at the same rate in both poor and rich environments, and so, in a simple sense, were no more surprising in one environment than the other. The moderate-value options did, however, constitute deviations from the long-term reward rate of the environment that prompted periodic changes in behavioural policy. In the present results, DRN activity was most strongly associated with these events.

Intriguingly, environment-related modulation of both behaviour and DRN activity was more prominent in poor environments compared to rich environments. Given that DRN is the main source of serotonin to the mammalian forebrain, it is tempting to relate this asymmetry to classic theories implicating serotonin in aversive processing[55,58–62], and the ability to adopt aversion- and/or threat-oriented modes of behaviour[46]. In line with the finding that DRN activity positively covaries with reward value[14], the present results do not implicate DRN activity in aversion per se, however. For example, one of the critical DRN activity patterns observed here signalled *pursuits of*—rather than aversion from –the 10-point option in poor environments. Nevertheless, this pattern is broadly consistent with the notion that DRN facilitates behavioural adaptation to adverse circumstances, because pursuits of the 10-point option were more common in poor environments. It is unclear whether this function arises from serotonin neurons specifically, as many DRN neurons are non-serotonergic, and it is not possible to discriminate serotonergic from non-serotonergic populations in fMRI recordings[62,63]. Moreover, the BOLD signal may not only reflect activity from serotonergic neurons but also local GABAergic inhibition, making the direction of the effect difficult to interpret. Future experiments in human participants could address this by examining whether pharmacological manipulations of the serotonin system influence patterns of behavioural policy change.

In contrast to DRN's role in policy switches, dACC and AI activity reflected options and their similarity to one another, given the richness of the environment. This accords with previous evidence that these regions track rewards over multiple timesteps and, in dACC's case, that

it evaluates rewards relative to the counterfactual alternatives that might be pursued in the current context[10,11,64–66]. Here, we probed this role by testing whether the representational distance between reward options changed across rich and poor environments. This demonstrated that individual differences in the distance between 10-point and 50-point options paralleled the degree of behavioural adaptation to rich and poor environments, suggesting that successful performance of the task was fundamentally linked to warps in the neural representations of options—warps which brought the 10-point and 50-point options closer together when they evoked the same behavioural policy, and pushed them further apart when behaviour diverged. Strikingly similar patterns have been observed in experiments applying RSA to other cognitive domains like navigation, where neural representations of spatial location appear to warp according to an agent's current and goal locations[67]. The present results show that the same warping phenomenon occurs in an abstract task space and suggest that warping is a common neural motif of context-dependent computation.

Notably, dACC and AI were the only ROIs with task-related changes in representational distances, and they were selected as ROIs in the first instance because they showed univariate increases in activity during reward-pursuit decisions. It is important to highlight, therefore, that we measured the distance between multivariate patterns of reward-evoked activity with cosine dissimilarity and Pearson's correlation—methods that obviate the influence of univariate changes on representational distance. This means, for example, that environment-driven changes in the representational distance between 10-point and 50-point options is not reducible to the fact that: (i) both 10-point and 50-point options are pursued in poor environments, and therefore evoke similar levels of univariate activity, and (ii) the 50-point option is pursued more than the 10-point option in rich environments, leading to divergence in univariate activity. Moreover, the multivariate patterns of activity analysed in dACC and AI were time-locked to option presentation, which was temporally decorrelated from behavioural responses. Nevertheless, given the presence of pursuit-vs-rejection-related activity in dACC and AI, it may be more appropriate to conceive the neural representations that these regions carry not simply as reflecting the options per se, but option-linked behavioural policies. Such a view is consistent with the interactions between DRN and dACC that occurred during the updating of behavioural policies.

Finally, it is worth briefly noting how the present results might relate to affective disorders. Affective disorders are marked by an inability to adapt behaviour to negative outcomes and have been linked to serotonin dysfunction through the efficacy of serotonin-specific reuptake inhibitor (SSRI) treatments[68,69]. Although the relationship between serotonin and affect is poorly understood[70], the fact that DRN activity signalled behavioural adaptations to poor environments in the present results is broadly consistent with the hypothesis that such disorders can be linked to serotonergic dysfunction. Furthermore, the present focus on general features of behaviour like behavioural policies might be a useful framework for operationalising critical aspects of affective disorders, such as apathy and anhedonia, in psychopharmacology or computational psychiatry experiments. Investigations linking MDB activity to action initiation and reward prediction errors in the healthy brain have generated new insights into the dopaminergic system's role in mood and emotion[71,72]. Attaining a better understanding of how DRN activity serves to link behavioural policies with the current environment might similarly shed light on serotonin's role in regulating subjective well-being.

## Methods

### Participants
Thirty-one participants (11 self-reported men and 20 self-reported women) performed the experiment. All were between 18 and 40 years

of age, reported normal or corrected-to-normal vision, and had no current diagnosis or treatment for psychiatric or neurological disorder. Participants received £40 for completing the experiment and could earn an additional payment of up to £10 depending on their performance in the decision-making task. All relevant ethical regulations governing research with human participants were observed, and participants provided written informed consent at the beginning of each session. Ethical approval was given by the Oxford University Central University Research Ethics Committee (CUREC) (Ref-Number: MSD-IDREC-R55856/RE001). Two participants were excluded from behavioural analyses because they did not perform the task correctly. In both cases, participants clearly failed to follow instructions and attempted to respond to stimuli on every trial. An additional two participants were excluded from fMRI analyses due to excessive head motion, which prevented accurate registration (motion outliers > 15% of total fMRI volumes). Self-reported sex of participants was not considered during analysis.

### Behavioural task

Before performing the experiment in the MRI scanner, participants received written instructions and completed a short practice version of the task. The experiment proper involved five rounds of the behavioural task, and each round was ~9 minutes (supplementary Fig. 1A). Participants were given a 60 s break between rounds, meaning that the total duration was approximately 50 minutes.

The behavioural task involved a series of encounters with sequential reward opportunities (Fig. 1a). The aim was to earn as many points as possible in the fixed task (45 minutes), and participants were incentivised to do this because their payment was partly determined by their performance. In brief, participants were presented with a single reward option on each trial and needed to decide whether to pursue (i.e., accept) or reject it. Pursuing an option incurred a temporal opportunity cost. Given the time-limited nature of the experiment, it was therefore critical to track the frequency of reward options over time—i.e., to anticipate what kind of reward option they might get in future trials—when deciding whether the opportunity cost was worth incurring. To make the task intuitive, it was presented as a treasure-hunt scenario in which the participant played the role of a ship's captain searching for treasure opportunities (i.e. reward options; Fig. 1a). Upon encountering a treasure opportunity, the captain needed to decide whether it was worth the time and effort to sailing toward it (pursue-decision), or to continue to explore the seas elsewhere (reject-decision).

Reward opportunities were drawn from a set of three, discrete options represented by visual stimuli: (1) a low-value option (5 points; bronze medallion), (2) a moderate value option (10 points; silver medallion), and (3) a high value option (50 points; gold medallion). The value of options was deterministic, constant over time, and known a priori to participants before they performed the experiment. The task, thus, did not involve learning about stimulus-reward relationships, or uncertainty about the payout of an option if/when it was chosen. Although option values were constant, the frequency of each option was systematically manipulated to create differences in the richness of the environment. Environments were operationalised via blocks of trials that were time-limited to 4.5 minutes. In rich blocks, the 50-point option was more frequent than the 10-point and 5-point options (Fig. 1b; $P(Low|Rich) = 0.16$, $P(Middle|Rich) = 0.33$, $P(High|Rich) = 0.50$). In poor blocks, the 5-point option was more frequent than the 10-point and 50-point counterparts (Fig. 1b; $P(Low|Poor) = 0.50$, $P(Middle|Poor) = 0.33$, $P(High|Poor) = 0.16$].

There were no cues indicating the environment during the task, and participants were not informed that the task was divided into discrete rich and poor environments—all they were told was that the frequency of the options might vary over time. As such, participants

needed to learn the richness of the environment by tracking the frequency with which specific options occurred.

Per above, the experiment consisted of five 9-minute rounds, and each round consisted of two 4.5-minute blocks (supplementary Fig. S1A). The amount of time remaining in each round was signalled using a timer in the top right-hand side of the screen. Each round featured one rich block and one poor block, and the order of rich/poor blocks within each round was random (supplementary Fig. S1A). The precise number of trials completed in each block—and therefore each round—varied depending on how many options were pursued during the allotted time (supplementary Fig. S1B; $\mu(trials)_{rich} = 15.95 \pm 1.06$, $\mu(trials)_{poor} = 18.06 \pm 1.27$, $t(26)_{rich-vs-poor} = 15.52$, $p < 0.001$). Each trial began with an inter-trial interval (ITI; $ITI \sim TruncExp(\mu = 4.5$, $min = 3.5$, $max = 5.5$), during which a treasure-chest visual stimulus representing an upcoming reward opportunity descended from the top to the centre of the screen. A visual stimulus—a picture of the medallion—replaced the treasure chest when the chest reached the centre of the screen, thereby revealing the reward value of the opportunity. Participants needed to consider the reward option for a short duration (Opportunity $\sim TruncExp(\mu = 4.5$, $min = 3.5$, $max = 5.5$) before it displaced either left- or rightward from the centre. The displacement functioned as a go-cue, which indicated that the reward option was eligible for pursuit. Imposing the go-cue delay created a temporal dissociation between brain activity related to pursue-vs-reject decisions and the motor processes undertaken to execute the decision via button press. Participants could pursue options by making a button press response within 1 s of the go-cue. Different responses were required for leftward and rightward pursuits to prevent anticipatory planning of movements during the go-cue delay. To reject a reward option, participants needed to withhold any button-press response and let the reward option pass.

Pursuing an opportunity incurred a temporal opportunity cost via action-outcome (Action-outcome $\sim TruncExp(\mu = 4.5$, $min = 3.5$, $max = 5.5$) and reward feedback delays (Rw-feedback $\sim TruncExp(\mu = 4.5$, $min = 3.5$, $max = 5.5$). The duration of these delays was equivalent to the time taken by one future encounter—in other words, pursuing an opportunity on the current trial meant foregoing a reward opportunity in the future. The magnitude of this cost was explained to participants during the instructions and visually indicated during the task via a passing treasure-chest stimulus during the action-outcome delay, which represented an opportunity forgone.

The parameters of the task were carefully chosen to ensure that the optimal strategy was to always pursue the 10-point option in rich blocks and to never pursue the 10-point option in the poor blocks. Per above, blocks were not signalled to participants, and we therefore did not expect them to display the optimal pattern of behaviour immediately. Instead, we expected that they would move towards the optimal pattern after accumulating experience of each block-based environment over time. Designing the task in this way ensured that there was a reward-maximising rationale for participants to change their behavioural policy toward the 10-point option.

The task was implemented in MATLAB R2022b (Mathworks Inc.) using the Psychophysics Toolbox v3 extension[73].

### Behavioural analysis

Our initial analysis of behaviour quantified the factors influencing the pursue-vs-reject decision made on each trial. We accomplished this with a series of mixed-effect binomial GLMs with participant identity as a random variable. All GLMs, therefore, accounted for inter-subject variability in effects of interest.

We first investigated whether pursue-vs-reject decisions were influenced by the richness of the environment, where the richness of the environment was operationalised as the experimentally controlled environment-type (rich-vs-poor block) on a given trial:

**GLM1.1a.**

$$logit(decision_t) = \beta_o + \beta_1 reward - magnitude_t + \beta_2 environment - type_t$$
$$+ \beta_3 trial - number_t + \beta_4 (reward - magnitude_t^* environmen_t - type_t) + \mu_0$$
$$+ \mu_1 (reward - magnitude_t^* environment - type_t) + \varepsilon$$

Where $\beta_{0-4}$ are fixed-effects and $\mu_{0-1}$ are random-effects within each subject. We followed this with a GLM that tested the effect of the environment separately for each reward option:

**GLM1.1b.**

$$logit(decision_t) = \beta_o + \beta_1 environment - type_t + \mu_0 + \mu_1 environment - type_t + \varepsilon$$

Where $\beta_{0-1}$ are fixed-effects, $\mu_{0-1}$ are random-effects within each participant. Per above, GLM1.1b was fit separately to subsets of trials featuring the 5-point, 10-point and 50-point options, respectively.

We then investigated whether the effect of environment type on pursue-vs-reject decisions was modulated by the amount of time elapsed within a block (*trial-within-block*). This interaction is consistent with the hypothesis that participants incrementally learned the richness of the environment with experience:

**GLM1.2.**

$$logit(decision_t) = \beta_o + \beta_1 environment - type_t + \beta_2 trial - within - block_t$$
$$+ \beta_3 (trial - within - block_t^* environment - type_t) + \mu_0 + \varepsilon$$

Where $\beta_{0-3}$ are fixed-effects and $\mu_0$ is a random-effect within each subject. GLM1.2 was fit separately to subsets of trials featuring the 5-point, 10-point and 50-point options, respectively.

We then tested the same hypothesis from a complementary perspective by operationalising the richness of the environment as the average value of the reward options encountered on the previous five trials (average-value):

**Average-value.**

$$average - value_t = mean(offer - value_{t-1} : offer - value_{t-5})$$

Operationalising the richness of the environment in this way captured a subject's experience of the task, instead of relying on the task's underlying generative structure. We investigated the effect of average value on pursue-reject decisions with the following GLM:

**GLM1.3.**

$$logit(decision_t) = \beta_o + \beta_1 average - value_t + \beta_2 trial - number_t$$
$$+ \mu_0 + \mu_1 average - value_t + \varepsilon$$

Where $\beta_{0-2}$ are fixed-effects and $\mu_{0-1}$ are random-effects within each subject. GLM1.3 was fit separately to subsets of trials featuring the 5-point, 10-point and 50-point options, respectively.

Because the task featured three discrete reward options that occurred many times over, we were able to precisely identify changes in a subject's behavioural policy toward each option (where behavioural policy indicates an option-specific pursue-vs-reject strategy; Fig. 2a). We first tested the hypothesis that participants formed option-specific policies by examining whether pursue-vs-reject decisions were autocorrelated over consecutive encounters with a particular reward option:

**GLM2.1.**

$$logit(decision_t) = \beta_0 + \beta_1 reward - magnitude_t + \beta_2 previous - policy_t$$
$$+ \beta_3 trial - number + \mu_0 + \mu_1 reward - magnitude_t + \mu_2 previous - policy_t + \varepsilon$$

Where $\beta_{0-3}$ are fixed-effects and $\mu_{0-2}$ are random-effects within each subject. Here, previous-policy indicates the pursue-vs-reject decision the last time that the reward option on trial $t$ was encountered. We confirmed that behavioural policies were option-specific by testing the effect of the pursue-vs-reject decision taken on the preceding trial (i.e., $t$-1) regardless of which reward option occurred:

**GLM2.2.**

$$logit(decision_t) = \beta_0 + \beta_1 reward - magnitude_t + \beta_2 previous - action_t$$
$$+ \beta_3 trial - number + \mu_0 + \mu_1 previous - action_t + \varepsilon$$

Where $\beta_{0-3}$ are fixed-effects and $\mu_{0-1}$ are random-effects within each subject. Together, GLM2.1 and GLM2.2 allowed us to test whether temporal structure in behaviour was option-specific, or whether participants also had autocorrelated predispositions to pursue vs. reject options in general, regardless of reward-option identity.

We then operationalised policy-switches as trials on which a subject changed their pursue-vs-reject decision across successive encounters with a particular option—for example, if a subject rejected the 10-point option on trial $t = 1$ and subsequently accepted the 10-point option on trial $t = 5$, trial $t = 5$ was coded as a policy-switch event. We also defined trials along an action dimension, where an action-switch occurred when participants changed their pursue-vs-reject decision across consecutive trials (regardless of which reward option appeared). This allowed us to compare option-specific tendencies (i.e., policy dimension) in purse/reject decisions with general tendencies (i.e., action dimension) in pursue-vs-reject decisions.

After defining each trial along policy-switch and action-switch dimensions, we conducted a series of analyses to identify the factors predicting policy-switches. Because the policy-switch status of each trial was binary (switch=1, stay =0), we used mixed-effect binomial GLMs where the dependent variable was the policy-switch status of each trial, and participant-identity was a random variable.

We first investigated whether the probability of policy-switches differed as a function of reward-option:

**GLM2.3.**

$$logit(policy - switch_t) = \beta_o + \beta_1 reward - magnitude_t + \beta_2 trial - in - block_t \mu_0$$
$$+ \mu_1 reward - magnitude_t + \mu_2 trial - in - block_t + \varepsilon$$

Where $\beta_{0-2}$ are fixed-effects and $\mu_{0-2}$ are random-effects within each subject.

We then investigated whether the direction of policy-switches for the 10-point option was modulated by environment type. To do this, we separated policy-switches into two different types: (1) *reject-switches*, when subjects rejected the 10-point reward-option but pursued it on the preceding encounter, and (2) *pursue-switches*, where subjects pursued the 10-point option but rejected it on the previous encounter. The optimal strategy for performing the task involved pursuing the 10-point option in poor environments and rejecting it in rich environments. Insofar as behaviour approximated this strategy, we expected that pursue-switches would occur more frequently in poor environments and reject-switches would occur more frequently in rich environments. We tested these hypotheses with the following GLMs:

**GLM2.4a.** ▨

$$logit(policy-switch_{t;\,poor})=\beta_o+\beta_1 previous-policy_t+\mu_0+\mu_1 previous-policy_t+\varepsilon$$

**GLM2.4b.** ▨

$$logit(policy-switch_{t;\,rich})=\beta_o+\beta_1 previous-policy_t+\mu_0+\mu_1 previous-policy_t+\varepsilon$$

Where $\beta_{0-1}$ are fixed-effects and $\mu_{0-1}$ are random-effects within each participant. GLM2.4a and GLM2.4b were fit to trials featuring the 10-point option. Note that although GLM2.4a and GLM2.4b have the same formula, GLM2.4a was fit only to trials from poor blocks and GLM2.4b was fit only to trials from rich blocks. We then tested the same hypothesis in a complementary way with the following GLMs:

**GLM2.5a.** ▨

$$logit(pursue-switch_t)=\beta_o+\beta_1 environment-type_t+\mu_0+\mu_1 environment-type_t+\varepsilon$$

**GLM2.5b.** ▨

$$logit(reject-switch_t)=\beta_o+\beta_1 environment-type_t+\mu_0+\mu_1 environment-type_t+\varepsilon$$

Where $\beta_{0-1}$ are fixed-effects and $\mu_{0-1}$ are random-effects within each participant. GLM2.5a and GLM2.5b were fit to trials featuring the 10-point option. We then tested the same hypotheses in an analysis where the richness of the environment was operationalised as the average value of the reward options encountered on the previous five trials:

**GLM2.6a.** ▨

$$logit(pursue-switch_t)=\beta_o+\beta_1 average-value_t+\mu_0+\mu_1 average-value_t+\varepsilon$$

**GLM2.6b.** ▨

$$logit(reject-switch_t)=\beta_o+\beta_1 average_t-value_t+\mu_0+\mu_1 average-value_t+\varepsilon$$

Where $\beta_{0-1}$ are fixed-effects and $\mu_{0-1}$ are random-effects within each participant. GLM2.6a and GLM2.6b were fit separately to subsets of trials featuring the 5-point, 10-point and 50-point options, respectively. All behavioural analyses were performed in *R* v4.2.2 (R Core Team, 2023) using the *lme4* v1.1.31 implementation of mixed-effects GLMs[74].

### Imaging data acquisition

Structural and functional MRI data were collected with a Siemens 7 Tesla MRI scanner. High-resolution functional data were acquired with a multiband gradient-echo T2* echo planar imaging sequence with 1.5 mm isotropic voxels, multiband acceleration factor 2, repetition time (TR) = 1.775 s, echo time (TE) = 17.8 ms, flip angle = 66°, and GRAPPA acceleration factor 2. The parameters were selected to maximise signal-to-noise ratio in subcortical areas. To accommodate the high temporal and spatial resolution of the protocol, functional scans had a limited field of view (FOV) oriented at 30 degrees with respect to the AC-PC line (66 slices with a coverage of 99 mm). The FOV captured all ROI in the midbrain, brainstem and cortex. Before acquiring the task-related functional scan, we acquired a pre-saturation single-measurement, whole-brain functional scan with the same orientation. The pre-saturation scan was used to facilitate registration of the limited-FOV task-related functional scan to the whole brain.

Structural data were acquired using a T1-wieghted MP-RAGE sequence with 0.7 mm isotropic voxels, GRAPPA acceleration factor 2, TR = 2200 ms, TE = 3.02 ms, and; inversion time (TI) = 1050 ms. To correct distortions arising from inhomogeneities in the magnetic field, a fieldmap sequence was acquired with 2 mm isotropic voxels, TR = 620 ms, TE1 = 4.08 ms, and TE2 = 5.1 ms. To account for the effects of physiological noise on functional MRI data, participants were fitted with a pulse oximeter and respiratory bellows that acquired cardiac and respiratory timeseries at 50 Hz using a BioPac MP160 device (BIOPAC Systems Inc., USA).

### fMRI data preprocessing

Preprocessing of fMRI data was performed with the FMRIB Software Library[75,76]. The Brain Extraction Tool[77] was used to separate brain from non-brain matter in structural and functional images. Functional images were normalised, spatially smoothed (Gaussian kernel with a 3 mm full-width half-maximum) and temporally high-pass filtered (3 dB cut-off = 100 s), and artefacts arising from head motion were removed using MCFLIRT[78]. Registration of task-related functional images to Montreal Neurological Institute (MNI)-space was performed in three stages:

1. The task-related limited-FOV EPI was registered to the pre-saturation whole-brain EPI using FMRIB's Linear Image Registration Tool with 6 degrees of freedom transformation.
2. The whole-brain EPI was registered to the subject-specific structural images using Boundary-Based Registration (BBR) incorporating fieldmap correction[79].
3. Subject-specific structural images were registered to a 1 mm resolution Standard MNI template with FMRIB's Non-linear Registration Tool (FNIRT[75]).

### Whole-brain fMRI data analysis

Statistical analysis of whole-brain functional data was performed at two levels using FMRIB's Expert Analysis Tool (FEAT[75]). In the first level, a univariate general linear model was used to compute parameter estimates for each regressor in each participant[80]. Contrast and variance estimates for each parameter in each participant were subsequently combined in a mixed-effects analysis conducted at the second level (FLAME 1 + 2), where subject-identity was a random effect[81]. Significance testing was performed with cluster-correction, a cluster significance threshold of $p = 0.001$, and a voxel inclusion threshold of $z = 3.1$. Data were pre-whitened before analysis to account for temporal autocorrelations in the BOLD signal.

We performed two whole-brain analyses. The first GLM identified voxels where the BOLD signal represented the pursue-vs-reject decision made on each trial. Note that the pursue-vs-reject decision made on each trial is regressed at two different timepoints: (1) at the time of opportunity onset, and (2) at go-cue onset. These events were separated in time by a jittered interval drawn from *Opportunity - TruncExp($\mu = 4.5$, min = 3.5, max = 5.5)*, meaning that neural activity at each time was temporally decorrelated. The second GLM identified voxels where the BOLD signal represented the reward magnitude of the reward option on each trial. We could not include both pursue-vs-reject at the time of opportunity onset and reward-magnitude regressors in the same GLM because they were highly correlated ($r = 0.64$). The formulas for GLMs were:

**GLM3.1.** ▨

$$BOLD_{v,t}=\beta_1 opportunity-onset+\beta_2 trial-number+\beta_3 decision(opportunity-onset)+\beta_4 go-cue-onset+\beta_5 decision(go-cue-onset)+\beta_6 feedback-onset+\beta_7 reward-feedback$$

**GLM3.2.** ▨

$$BOLD_{v,t}=\beta_1 opportunity-onset+\beta_2 trial-number+\beta_3 reward-magnitude+\beta_4 go-cue-onset+\beta_5 decision(go-cue-onset)+\beta_6 feedback-onset+\beta_7 reward-feedback$$

Where $BOLD_{v,t}$ is a $t \times v$ matrix containing the BOLD signal measurements at time $t$ in voxel $v$. The regressors are as follows: *opportunity-onset* is an unmodulated regressor representing the occurrence of the reward-option stimulus on each trial; *trial-number* is a parametric regressor representing the number of trials completed in the

experiment as a proxy for the passage of time, time-locked to the opportunity onset; *decision(opportunity-onset)* is a binary regressor reflecting the pursue-vs-reject decision on each trial, time-locked to the opportunity onset; *reward-magnitude* is a parametric regressor reflecting the value $\in \{5, 10, 50\}$ of the reward-option time-locked to the opportunity onset; *go-cue-onset* is an unmodulated regressor representing the occurrence of the go-cue stimulus on each trial; *decision(go-cue-onset)* is a binary regressor reflecting the pursue-vs-reject decision on each trial, time-locked to the onset of the go-cue stimulus; *feedback-onset* is an unmodulated regressor representing the occurrence of reward-feedback received on trials when the opportunity was accepted; *reward-feedback* is a parametric regressor with three levels (5, 10, 50) representing the reward-option, and time-locked to feedback presentation on trials where opportunities were accepted.

All regressors were boxcar functions with a constant duration of 0.1 s and convolved with a double-gamma hemodynamic response function. Further non-task confound regressors were added to reduce noise in BOLD signal, including: (1) head motion parameters estimated using MCFLIRT during pre-processing[78]; (2) regressors for voxel-wise estimates of physiological noise arising from cardiac and respiratory activity, estimated using FSL's Physiological Noise Monitoring (PNM) tool[82], and; (3) regressors for motion outliers, indicating volumes with head motion that could not be corrected with linear methods.

### ROI timecourse fMRI analysis

Additional analysis of brain activity was conducted on epoched time-course activity extracted from ROIs. Anatomical ROIs were constructed for a series of brainstem and midbrain nuclei previously implicated in reward-guided decision-making, including DRN, Locus Coeruleus (LC), Medial Septum (MS), a combined mask covering the Substantia Nigra and Ventral Tegmental Area (MDB), and Habenula (Hb). The LC mask was derived from a brain atlas developed by Pauli and colleagues[83]. DRN, MDB, Hb and MS masks were manually drawn in MNI-space based on an atlas of the human brain atlas[84]. The DRN mask was drawn in consultation with the anatomical guidelines of Kranz and colleagues[85] to ensure specificity to the dorsal portion of the raphe nucleus. Standard-space masks were transformed to subject-specific structural space by applying a standard-to-structural warp field. To ensure conformation between anatomical masks and subject-specific neuroanatomy, subject-specific masks were manually checked and edited by two experimenters and evaluated for inter-rater reliability by a third experimenter. Structural-space masks were then transformed from structural to functional space by applying a structural-to-functional affine matrix, and were subsequently thresholded, binarized and dilated by one voxel.

To prepare ROI data for analysis, the filtered time-series of BOLD signal from each voxel was averaged, normalised (such that the time-course's mean activity was centred at 0 and its standard deviation was 1) and up-sampled by a factor of 20 with spline interpolation[86]. Upsampled timeseries were then epoched in 10 s windows starting 2 s before and ending 8 s after reward-option onset on each trial. Time-series data was analysed by fitting a GLM with ordinary least squares (OLS) at each time point in each epoch. The statistical significance of parameters in GLMs was assessed using a leave-one-out procedure that avoided temporal biases in the selection of peak-effects[42]. The procedure involved estimating, for each regressor $r$ in each subject $s$, the time $t$ at which the mean effect of $r$ was greatest in the remaining $N$-1 participants. The effect of $r$ in $s$ was then estimated by calculating the parameter estimates for $r$ at time $t$ in subject $s$. This was repeated iteratively for each subject, leaving an $N$x1 vector of parameter estimates that were not biased by the time course of the data of individual participants. We assessed the statistical significance of subject-specific parameter estimates by running a single-sample t-test on the vector of

parameter estimates. All $t$ tests were two-sided. Whenever analyses involved three or more ROIs, we maintained the family-wise error rates at $\alpha = 0.05$ using the Holm-Bonferroni procedure for multiple comparisons[87]. Time-course analysis was performed in MATLAB 2022b.

We tested which ROIs represented policy-switches with the following GLM:

**GLM4.1a.** ▨

$$BOLD = \beta_0 \text{constant} + \beta_1 \text{policy} - \text{switch} + \beta_2 \text{trial} - \text{number}$$

Where *policy-switch* is a binary variable indicating whether a policy-switch occurred on trial $t$. We then tested which ROIs represented action-switches with the following GLM:

**GLM4.1b.** ▨

$$BOLD = \beta_0 \text{constant} + \beta_1 \text{action} - \text{switch} + \beta_2 \text{trial} - \text{number}$$

Where *action-switch* is a binary variable indicating whether an action-switch occurred on trial $t$ (see Fig. 2a for definitions).

To investigate brain activity related specifically to task-driven policy-switches in behaviour, we classified policy-switch events as either *congruent* or *incongruent* in relation to the environment they occurred in. Recall that the optimal strategy for performing the task was to pursue the 10-point opportunity in poor environments and reject it in rich environments. Accordingly, we coded pursue-switches in poor environments as congruent and pursue-switches in rich environments as incongruent. Correspondingly, reject-switches in rich environments were congruent and reject-switches in poor environments were incongruent.

We used these four event types as regressors in the following series of GLMs:

**GLM4.2a.** ▨

$$BOLD = \beta_0 \text{constant} + \beta_1 \text{congruent} - \text{switch} - \text{poor} + \beta_3 \text{trial} - \text{number}$$

**GLM4.2b.** ▨

$$BOLDi = \beta_0 constant + \beta_1 \text{incongruent} - \text{switch} - \text{rich} + \beta_3 \text{trial} - \text{number}$$

**GLM4.2c.** ▨

$$BOLD = \beta_0 \text{constant} + \beta_1 \text{congruent} - \text{switch} - \text{rich} + \beta_3 \text{trial} - \text{number}$$

**GLM4.2 d.** ▨

$$BOLD = \beta_0 constant + \beta_1 \text{incongruent} - \text{switch} - \text{poor} + \beta_3 \text{trial} - \text{number}$$

Where *congruent-switch-poor* is a binary variable indicating whether a congruent policy-switch in a poor environment occurred on trial $t$, *incongruent-switch-poor* is a binary variable indicating whether an incongruent policy-switch in a poor environment occurred on trial $t$, *congruent-switch-rich* is a binary variable indicating whether a congruent policy-switch in a rich environment occurred on trial $t$, and *incongruent-switch-rich* is a binary variable indicating whether an incongruent policy-switch in a rich environment occurred on trial $t$. We tested differences in neural activity related to these event-types by performing paired-sample t-tests on the effect sizes for different event types.

Per above, the optimal strategy in the task involved changing behavioural policy toward the 10-point option depending on the environment. Conversely, there was no task-driven reason for changing the behavioural policy toward the 5-point option, which should always have been rejected. To further test for brain activity related to

task-driven policy-switches, we separately analysed the effect of policy-switches on trials featuring the 10-point and 5-point options. We did this by performing the analysis in GLM4.2a and GLM4.2c for sub-samples of data containing 10-point option trials and the 5-point option trials, respectively.

We tested for PPI between DRN and dACC/AI as a function of environment type with the following GLM:

**GLM4.3.** ▨

$$BOLD_{ROI} = \beta_o + \beta_1 BOLD_{seed} + \beta_2 PPI + \beta_3 environment - type + \beta_4 trial - number$$

Where $BOLD_{seed}$ is time-series data for the seed region in PPI analysis (here, DRN), $PPI$ is the interaction between $BOLD_{seed}$ and *environment-type* regressors, and *environment-type* is the experimentally specific block-type (i.e., rich-vs-poor).

We tested whether univariate activity in dACC and/or AI represented rich-vs-poor environments with the following GLM:

**GLM4.4.** ▨

$$BOLD = \beta_o + \beta_1 environment - type + \beta_2 trial - number$$

Where *environment-type* was the experimentally programmed environment (rich-vs-poor).

Finally, we compared environment-specific representations of policy switches in DRN with dACC and AI with the following GLM:

**GLM4.5.** ▨

$$BOLD = \beta_o + \beta_1 congruent - change + \beta_2 trial - number$$

Where *congruent-change* contrasted pursue-switches in poor environments with reject-switches in rich environments (congruent-pursue=1; congruent-reject = −1).

## Representational similarity analysis

We further investigated neural representations of the task environment using representational similarity analysis (RSA[38]). RSA was performed on the following cortical ROIs: anterior insular cortex (AI), dorsal anterior cingulate cortex (dACC), lateral frontopolar cortex, subgenual anterior cingulate cortex (sgACC) and pregenual anterior cingulate cortex (pgACC) and dorsomedial prefrontal cortex (dmPFC). AI, dACC and lFPC were selected because they exhibited clusters of activity related to the pursue-vs-reject contrast in whole-brain fMRI analysis (GLM3.1; see supplementary table 1). Similarly, sgACC was selected because it exhibited activity related to the reward-magnitude contrast in a whole-brain fMRI analysis (GLM3.2; see supplementary table 2). dmPFC and pgACC were selected based on functional activations in previous studies. Each ROI was constructed as a sphere with a 7 mm radius centred on the voxel with the highest β-weight in the area of activity identified after cluster-correction.

We used RSA to compare differences in the neural representation of reward options between rich and poor environments. We predicted that the distance between option-specific neural representations would mirror changes in the relative-value of reward options given the environment—i.e., the 10-point would be closer to the high-option in poor environments compared to rich environments, and closer to the low-option representations in rich environments compared to poor environments (Fig. 4a). The analysis consisted of two stages: (1) quantifying the multi-voxel patterns of activity evoked by 5-point, 10-point and 50-point reward options in rich and poor blocks of the experiment, and; (2) computing the distance between multi-voxel representations, and quantifying how distances changed as a function of environment-type.

We measured multi-voxel representations of reward options using a GLM implemented in FSL FEAT. fMRI data pre-processing followed the pipeline described in *fMRI data preprocessing* above, except that spatial smoothing was omitted to maximise voxel-specific information. Omitting spatial smoothing prevented accurate registration for two participants, who were therefore excluded from the analysis ($N_{RSA} = 25$). The GLM quantified the multi-voxel patterns of activity evoked by each reward option with separate binary predictors indicating the onset of each option in each block (i.e., 30 predictors in total). It had the following formula:

**GLM5.1.** $BOLD_{v, t, r} =$

$\beta_1$op.-onset(low; block-1) + $\beta_2$op.-onset(mid; block-1) + $\beta_3$op.-onset(high; block-1) +

$\beta_4$op.-onset(low; block-2) + $\beta_5$op.-onset(mid; block-2) + $\beta_6$op.-onset(high; block-2) +

$\beta_7$op.-onset(low; block-3) + $\beta_8$op.-onset(mid; block-3) + $\beta_9$op.-onset(high; block-3) +

$\beta_{10}$op.-onset(low; block-4) + $\beta_{11}$op.-onset(mid; block-4) + $\beta_{12}$op.-onset(high; block-4) +

$\beta_{13}$op.-onset(low; block-5) + $\beta_{14}$op.-onset(mid; block-5) + $\beta_{15}$op.-onset(high; block-5) +

$\beta_{16}$op.-onset(low; block-6) + $\beta_{17}$op.-onset(mid; block-6) + $\beta_{18}$op.-onset(high; block-6) +

$\beta_{19}$op.-onset(low; block-7) + $\beta_{20}$op.-onset(mid; block-7) + $\beta_{21}$op.-onset(high; block-7) +

$\beta_{22}$op.-onset(low; block-8) + $\beta_{23}$op.-onset(mid; block-8) + $\beta_{24}$op.-onset(high; block-8) +

$\beta_{25}$op.-onset(low; block-9) + $\beta_{26}$op.-onset(mid; block-9) + $\beta_{27}$op.-onset(high; block-9) +

$\beta_{28}$op.-onset(low; block-10) + $\beta_{29}$op.-onset(mid; block-9) + $\beta_{30}$op.-onset(high; block-10)

$BOLD_{v,t,r}$ is a $t$ x $v$ x $r$ matrix containing the BOLD signal measurements at time $t$ in voxel $v$ in ROI $r$. $\beta_{1-30}$ are unmodulated regressors representing the occurrence of a given reward-option in a given block (e.g., option-onset(low; block-1) indicates onsets of low-option in the first block of the experiment). All option-onset regressors were implemented as boxcar functions with a constant duration of 0.1 s and convolved with a double-gamma hemodynamic response function. Confound regressors for physiological noise and motion artefacts were included, as per whole-brain fMRI analysis.

For each ROI $r$, and each predictor $i$ in $\beta_{1-30}$, GLM5.1 produced a vector of $v$ regression weights capturing the effect of $\beta_i$ at each voxel $v$ in $r$. Each vector of regression weights constituted a multivariate neural representation of a reward option. We then quantified pairwise distances between multivariate representations using cosine dissimilarity and Pearson's correlation to avoid artefacts arising from univariate differences in activity (see Discussion for further details). Cosine dissimilarity is unaffected by multiplicative scaling, while Pearson's correlation is unaffected by both multiplicative scaling and additive shifts. As a result, both measures are insensitive to absolute differences in the magnitude of the vectors.

Our analysis focussed on four representational distances; (i) $d_1$, the distance between 5-point and 10-point options in poor environments; (ii) $d_2$, the distance between 10-point and 50-point options in poor environments; (iii) $d_3$, the distance between 5-point and 10-point options in rich environments, and; (iv) $d_4$, the distance between 10-point and 50-point options in rich environments.

To avoid biases arising from the temporal autocorrelation of the BOLD signal, distance calculations were performed by comparing option-specific representations between non-contiguous blocks of the same environment type (see supplementary Fig. S9). For example, to calculate $d_1$—the distance between 5-point and 10-point options in poor environments—we might compare $\beta_{11}$op.-onset(mid; block-4) vs $\beta_{16}$opportunity-onset(low; block-6), where blocks 4 and 6 are poor

environments that are 1-block apart (block-space, hence). Because the analysis was performed to compare changes in representational distance between environments, we excluded representational distance scores if their block-space was not present for both rich and poor environments. To avoid overrepresentation of specific block-space values in representational distance scores, we calculated the mean representational distance at each value of block-space for each environment. For final representational distances, we calculated the mean representational distance over all levels of block-space in each environment. This process was performed separately for each ROI in each participant.

Once subject-specific $d_1$, $d_2$, $d_3$, and $d_4$, scores were calculated, we tested our hypotheses by taking the difference between environment-specific distance scores (e.g., $H1_{10point-vs-5point} = d_{1-3}$; $H2_{10point-vs-50point} = d_4 - d_2$). We tested the relationship between representational distances and behaviour by performing Pearson's $R$ correlations between H1 and H2 scores and the degree of behavioural change for the 10-point option between environments [$\Delta$(pursue-rate $=$ pursue-rate$_{poor}$ $-$ pursue-rate$_{rich}$]. We supplemented this by dividing participants into two groups based on their degree of behavioural change: (i) a sensitive group, where (pursue-rate$_{poor}$ $-$ pursue-rate$_{rich}$ $\geq 0$), and (ii) an insensitive group, where (pursue-rate$_{poor}$ $-$ pursue-rate$_{rich}$ $< 0$). We then tested whether H2 scores were different between groups using a two-sample t-test. In addition, we investigated differences in representational distance regardless of behaviour by testing H1 and H2 against 0 using single-sample t-tests. These analyses were performed separately for each ROI. The family-wise error rate was controlled separately for t-test and correlation analyses using the Bonferroni-Holm correction for multiple comparisons[87].

### Reporting summary
Further information on research design is available in the Nature Portfolio Reporting Summary linked to this article.

## Data availability
Data generated in this study have been deposited at: https://github.com/lpriestley/drn_policy_change/. Source data are available with this paper. Source data are provided with this paper.

## Code availability
Code to replicate the analyses and figures is available at: https://github.com/lpriestley/drn_policy_change/.

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

## Acknowledgements

The study was funded by BBSRC grant BB/W003392/1 and Wellcome Trust grant 221794/Z/20/Z to M.F.S.R., BBSRC Discovery Fellowship BB/W008947/1 to N.K., and Clarendon Scholarship to L.P.

## Author contributions

L.P., A.M., N.K., and M.F.S.R. conceptualised the study; L.P. collected the data; W.D.R. and N.K. pre-processed the data; L.P., A.M., W.D.R., N.K., and M.F.S.R. analysed and interpreted the data; L.P. drafted the initial manuscript; all authors revised and approved the final manuscript.

## Competing interests

The authors declare no competing interests.
