## [Transparent Peer Review file · Nature Communications]

ACTIVITY IN HUMAN DORSAL RAPHE NUCLEUS SIGNALS CHANGES IN BEHAVIOURAL POLICY

Corresponding Author: Mr Luke Priestley

Version 0:

Reviewer comments:

Reviewer #1

(Remarks to the Author)

In this paper, results from an ultra-high resolution fMRI experiment in a moderately sized sample of healthy volunteers are described, which lead to the conclusion that BOLD signaling in the dorsal raphe nucleus (DRN) represents changes in 'behavioural policy' (whether to accept/make a Go response or reject/make a Nogo response) with respect to an option that promises reward, in a manner that is appropriate given the average richness of the environment. Conversely, cortical regions including the ACC and insula are described to represent the environment-specific value of the options (and 'their similarity to one another given the richness of the environment').

The study sheds light on a timely and important issue, not least given the presence of deficits in environment-specific adaptation of behavioral policy to fluctuations in environmental richness in a variety of mental health disorders that implicate this neural circuit that connects the DRN with dACC/AI. The paper is well written and well structured, the figures are informative, and the (behavioral, univariate and multivariate fMRI) analyses are clever. Moreover, the conclusion that 'DRN and dACC/AI constitute a circuit in which cortical components construct an environment-specific representation of each option, while DRN implements the changes in behavioural policy consequent on these environment-specific changes' is plausible, given existing theorizing, and prior knowledge.

There might be a few ways, however, to increase the confidence in conclusions based on the evidence presented here.

First, definition of the key ROIs including the DRN, but also the other neuromodulatory nuclei i.e., LC and dopaminergic midbrain is not trivial. Traditionally, special dedicated MR sequences are used to identify the individually different locus of these small regions. The ROIs were traced manually. How can the authors convince the readers that the signals reported to be centered on the DRN, LC etc are indeed focused on those regions? What were the exact criteria for drawing the masks the way they are drawn? Were ROIs traced by at least two different researchers? What was the inter-rater reliability? What do the masks shown in the Fig 3a represent? An example individual mask? A group mask?

Second, rather than just showing signal averaged across ROIs, displays of whole-brain maps of the key effects of interest (e.g., policy-switches as a function of environmental richness, the PPI etc) can significantly increase confidence, also because those enable assessment of the physiological plausibility of the reported effects (are effects clustered, do they look artefactual, are they bilateral etc).

Third, were effects in DRN compared directly with effects in other ROIs (lc, dopa midbrain etc)?

Fourth, some of the key conclusions seem grounded in p-values that hover around 0.05, with great emphasis on effects just below the 0.05 threshold (such as the environment-driven policy switch effect in DRN), and dismissal of effects just above that 0.05 threshold (such as the environment-driven policy-switch effect in some cortical regions). Can the authors convince the readers that their key take-home message (which implies a double dissociation) that 'cortical components [but not DRN] construct an environment-specific representation of each option, while DRN [but not cortical regions] implements the changes in behavioural policy consequent on these environment-specific changes' still stands despite this reliance on this arbitrary threshold? Presentation of relevant whole-brain maps (of both the policy-switch contrast, as well as the RSA) and/or direct comparisons of ROIs might help?

Finally, the wordings of the key conclusion in the abstract ('that the human DRN represents changes between general behavioural policies') and in some parts of the discussion ('that DRN has a conserved function of matching the general policy governing behaviour with the general reward statistics of the environment') seem overly general, and might benefit from specification. Many neuromodulatory systems have been argued to implement behavioral strategy change, and there are many different statistical parameters that can describe the reward structure of an animal's environment. The key question seems to be *which type* of behavioral strategy decision is carried by which system, and as a function of *which specific* feature of the statistical structure of its environment.

Other comments:

Jaskir and Frank (2023, Elife) have put forward a relevant theoretical (normative) framework for predicting effects of environmental richness of behavioral activation (go responses) vs inhibition (nogo responses). Seems relevant and can the results be discussed? In this context, I wondered, would the same pattern of results have been predicted if subjects would have had to choose between 2 active response options instead of a Go vs Nogo response?

To what degree is it possible that the effect of environmental richness surfaces for the 10p option most clearly because of greatest dynamic range/uncertainty?

Stating that there was 'a null effect' require statistical testing of null effects. In the absence of such explicit (Bayesian) testing, it would be more accurate to state that 'there was no evidence for an effect...')

Two participants were excluded from behavioural analyses because they did not perform the task correctly. Please clarify

(Remarks on code availability)

Reviewer #2

(Remarks to the Author)

See attachment.

(Remarks on code availability)

I viewed the code but did not try to run it myself. It seems clean and well documented.

Reviewer #3

(Remarks to the Author)

In this paper the authors use 7T fMRI and a novel behavioral task to show a role of Dorsal Raphe Nucleus (DRN) in signaling in policy changes, and effects of reward context on representations in dorsal Anterior Cingulate Cortex (dACC) and Anterior Insula (AI).

They further show that the reward context influences connectivity between DRN and dACC. I particularly liked this part since context and context switching is an important (low order) approach to understanding human cognitive representations.

This is an interesting paper that makes a very significant contribution to understand the role of DRN signaling in human decision making. 7T functional imaging lends extra weight to the findings. The paper is well-written and provides an appropriate amount of detail.

There are only a few minor points to clarify. Also, it is interesting that the analysis relies on variables that are not based on any particular model of the behavioral data. While this avoids debates about such models it would be interesting to see if, for example, a hidden Markov model could capture the switches in behavioral policy.

Minor Points

Line 233. "high field" – Why not just sat 7T? I don't think this is stated until the methods.

Line 260. "fig.3D-E" There is no figure 3E. Also, it is not clear what the GLM (and beta) is used in (the submitted) figure 3C.

Line 357. It would be clarifying to add here that the event is "opportunity onset".

Line 424. There is no GLM 4.3 in the methods. It is not clear what is being shown in figure 5B.

Line 1143. "normalized" – how?

Line 1146. Should say the GLMs were performed.

Line 1151. I think everything here was done within subject and then averaging beta values over the N-1 subjects then maximizing. I think this should be clarified. Also, I think "effect size for" could be confusing. I think beta value is what is meant

since it is later stated “N x 1 vector of parameter estimates”.

Line 1232. “cluster-corrected Beta-weight”. I think cluster corrected p-value (or T or z value) is meant.

Line 1276. “vector of v ...” v is an index, not the number of voxels.

(Remarks on code availability)

Reviewer #4

(Remarks to the Author)

In the current study, Priestley and colleagues investigated the neural correlates of behavioral policy change in a foraging-like task. In the task, participants chose between pursuing vs. rejecting rewards in either reward-rich or -poor environments. Pursuing entailed opportunity costs of time, so that participants had to estimate whether pursuing lower-level rewards was worth the time cost of doing so. The authors report that activity in the dorsal raphe nucleus (DRN), but not in other nuclei in the ascending neuromodulatory systems tracks environment-congruent policy switches. Moreover, activity in the dorsal anterior cingulate cortex (dACC) and anterior insula (AI) seemed to represent offer value in relation to the average reward rate of the environment.

Overall, I think that this experiment was expertly conducted and that the results are convincing. The findings would be of broad appeal across disciplines interested in decision-making and fit the scope of the journal very well. Before I can fully endorse the manuscript for publication, however, I have a few comments and concerns, of which I am certain the authors can adequately address.

My most major concern relates to the conceptualization of policy switching in the behavioral task. The authors defined switches as changes in pursue vs. reject responses in consecutive trials of the same reward magnitude. On page 8, lines 323-337, the authors report that DRN activity reflected environment-congruent switches only, while medial septal nucleus (MS) activity tracked environment-incongruent switches, which they think of as explorative. However, especially early in a block, all switches, including those that are environment-congruent, might reflect exploratory behavior (and learning) instead of deviations from an environment-driven optimal policy (which is indicated by the increasing/decreasing acceptance rates of the 10-point option over time). Unless I misread, all trials were used in the analyses of DRN activity. But how could DRN activity truly reflect environment-driven policy changes at a time when participants could not know yet whether they were currently in a rich or a poor block? Or in other words, how could DRN discriminate whether a chosen action represents an environment-congruent or -incongruent switch when the nature of the environment is yet unknown? As far as I could tell, there was no analysis of how DRN activity changes over time, similar to the behavioral adaptation to the environment. For example, it would be interesting to see how long it takes DRN activity to adapt to a change in the environment (i.e., how many trials it takes until DRN activity reliably scales positively with pursuing vs. rejecting 10-point offers when changing from a poor to a rich environment).

A second potential problem with the interpretability and generalizability of the study is the relatively low sample size ($n = 29$, of which only 25 were included in the fMRI analyses). I understand the practical challenges of testing large samples in fMRI studies. However, I was wondering if there were any a priori considerations about statistical power or whether sensitivity analyses have been conducted to ensure the study was sufficiently powered?

More minor points:

- I would greatly appreciate if the authors could include some descriptive statistics about participants choice behavior in the text. Specifically, it would be great to see mean (+/-SEM) pursue/reject rates for each environment as well as for the total number of policy switches especially in the 10-point condition. I know the former can be inferred from Figure 1E but it would still be interesting to have the exact numbers. Given that only certain trials (those with policy switches) were included in most analyses, it is important to know how many of these existed per participant, to make sure the analyses are adequately powered.

- I would encourage the authors to please report participants' sex or gender, even if no analysis were done to investigate differences based on these variables.

- Similarly, I would appreciate a more detailed explanation for the exclusion of 2 participants based on task performance. How exactly did they perform incorrectly?

Typos:

- Page 3, line 102: “Upon each, encounter,... (first comma should be deleted)
- Page 6, lines 221-222: “A similar pattern obtained...” (WAS obtained)
- Page 9, line 353: “...on a series of ROIs deriving from...” (I think this should be DERIVED)
- Page 13, lines 551-552: “...might similarly shed light serotonin's role...” (shed light ON)

(Remarks on code availability)

Version 1:

Reviewer comments:

Reviewer #1

(Remarks to the Author)

The authors have done a great job revising the paper and responding to the issues raised, and I have no further comments! The study represents an important contribution to the rich literature

(Remarks on code availability)

Reviewer #2

(Remarks to the Author)

See attachment.

(Remarks on code availability)

I did not review the code in detail. I attempted to find the script that calculates representational distances for RSA, but could only locate the spreadsheet with the final d values.

Reviewer #4

(Remarks to the Author)

The authors have thoroughly and satisfactorily addressed all of my concerns. I have no further comments other than that I think this is a very interesting manuscript that will be greatly appreciated by other researchers in the field.

(Remarks on code availability)

From Nature Communications guideline:

1. **If the research findings apply to only one sex or gender, that must be indicated in the title and/or abstract.**

We confirm that our study included participants who self-reported as being women (N=20) or men (N=11). Therefore, no change to the title or abstract is necessary in this regard.

2. **For studies involving vertebrates animal and cell lines- The Reporting Summary should include whether sex was considered in the study design.**

Not applicable

3. **For studies involving human research participants- The Reporting Summary should include whether sex and/or gender was considered in the study design and whether sex and/or gender of participants was determined based on self-report or assigned (and methodology used).**

We confirm that sex and gender were not considered in the study design. Participant sex was determined based on self-report, as specified in the "Participants" section of the Methods.

4. **Data should be reported disaggregated for sex and gender where this information has been collected and consent has been obtained for reporting and sharing individual-level data; disaggregated numbers for individual experiments must be provided in the source data as appropriate whereas overall numbers may be provided in the Nature Portfolio Reporting Summary.**

In addition, please note that if sex- and gender-based analyses have been performed a priori, results should be reported regardless of positive or negative outcome. We discourage conducting post hoc sex- and gender-based analysis if the study design is insufficient (for example, low sample size) to enable meaningful conclusions.

Information on the points above should be included in the revised manuscript and detailed in the cover letter.

If no sex- and gender-based analyses have been performed, please indicate the reasons for the lack of these analyses in the Reporting Summary.

We confirm that sex information was collected based on self-report, but consent was not obtained for reporting and sharing individual-level data and sex was not included as a factor in the study design.

REVIEWER COMMENTS

*We thank the editor for sending the manuscript out for review. We have revised the manuscript in the light of the reviewers' comments. We have explained how we have done this in detail below where the reviewer comments are shown in **bold**, our response in italics, and the manuscript text is shown in a standard font with new sections highlighted in red.*

Reviewer #1 (Remarks to the Author):

In this paper, results from an ultra-high resolution fMRI experiment in a moderately sized sample of healthy volunteers are described, which lead to the conclusion that BOLD signaling in the dorsal raphe nucleus (DRN) represents changes in 'behavioural policy' (whether to accept/make a Go response or reject/make a Nogo response) with respect to an option that promises reward, in a manner that is appropriate given the average richness of the environment. Conversely, cortical regions including the ACC and insula are described to represent the environment-specific value of the options (and 'their similarity to one another given the richness of the environment').

The study sheds light on a timely and important issue, not least given the presence of deficits in environment-specific adaptation of behavioral policy to fluctuations in environmental richness in a variety of mental health disorders that implicate this neural circuit that connects the DRN with dACC/AI. The paper is well written and well structured, the figures are informative, and the (behavioral, univariate and multivariate fMRI) analyses are clever. Moreover, the conclusion that 'DRN and dACC/AI constitute a circuit in which cortical components construct an environment-specific representation of each option, while DRN implements the changes in behavioural policy consequent on these environment-specific changes' is plausible, given existing theorizing, and prior knowledge.

There might be a few ways, however, to increase the confidence in conclusions based on the evidence presented here.

- 1. First, definition of the key ROIs including the DRN, but also the other neuromodulatory nuclei i.e., LC and dopaminergic midbrain is not trivial. Traditionally, special dedicated MR sequences are used to identify the individually different locus of these small regions. The ROIs were traced manually. How can the authors convince the readers that the signals reported to be centered on the DRN, LC etc are indeed focused on those regions? What were the exact criteria for drawing the masks the way they are drawn? Were ROIs traced by at least two different researchers? What was the inter-rater reliability? What do the masks shown in the Fig 3a represent? An example individual mask? A group mask?**

We thank the reviewer for raising this important point. They are correct that specialized MRI sequences are sometimes used to identify individual nuclei. However, these sequences are typically optimized for specific nuclei— for example, a sequence designed for the VTA would not be suitable for identifying the DRN. This creates a trade-off between anatomical specificity and the ability to compare across multiple neuromodulatory nuclei. In our study, we prioritized the latter by using a common sequence to sample multiple regions of interest, which allows for cross-nucleus comparisons. However, we also ensured the anatomical specificity of our effects by using rigorous approaches in designing our masks. Below, we outline how our masks were designed and describe additional analyses we have now performed to demonstrate the anatomical specificity of our main effects:

- As explained in the Methods section, the anatomical masks were manually drawn in MNI-space based on an atlas of the human brain atlas (Mai et al., 2016). The DRN mask, in particular, was drawn in consultation with the anatomical guidelines of Kranz and colleagues (Kranz et al., 2012). To ensure conformity between anatomical masks and subject-specific neuroanatomy, subject-specific masks were manually checked and edited by two experimenters and evaluated for inter-rater reliability by a third experimenter. Structural-space masks were then transformed from structural to functional space by applying a structural-to-functional affine matrix.
- In response to the reviewer's request and to show that our signals are indeed from the regions of interest we have performed the following additional analysis:
 1. The masks in Fig.3a show anatomical ROIs in standard space. We now, however, also show anatomical masks from an example session in functional space. In addition, the extracted BOLD signal from each mask is displayed next to its corresponding ROI (supplementary figure S3).

Figure S3 Details of anatomical masks for subcortical regions of interest. Anatomical masks from an example session in functional space. Anatomical masks were designed for each ROI in the standard space (see fig.3A) and were transformed from the standard space to each subject's functional space by applying a standard-to-structural-to-functional warp. The masks are overlaid on functional image from a representative session. The extracted BOLD signal from each mask is displayed next to its corresponding ROI. The lines show the BOLD signal extracted and averaged from each voxel within the ROI, for the whole duration of the scanning session.

2. To demonstrate that the behavioural policy effect is indeed specific to DRN, we have performed an additional analysis of activity in two locations adjacent to the DRN. First, we extracted the time course from the 4th ventricle and, second, we extracted it from the MRN – two immediately adjacent structures to DRN and LC. There were no corresponding patterns of activity in these areas (supplementary fig.S5; $t_{DRN}(26) = 2.85$, $p = .009$, $t_{MRN}(26) = -0.36$, $p = .724$, $t_{Vent.}(26) = -0.83$, $p = .415$). Moreover, the policy-switch effect in DRN was, indeed, greater than the effects in MRN and the 4th ventricle ($t_{DRN-vs-MRN}(26) = 2.08$, $p = .047$, $t_{DRN-vs-Vent}(26) = 2.31$, $p = .028$). These results show that the behavioural policy effect is specific to DRN and could not be observed in immediately adjacent structures. We have added these new results to supplementary figure S5.

Figure S5 Further details of policy-switch effect in DRN. (A) To confirm that the policy-switch effect was spatially specific to DRN, we extracted the time course of BOLD signal from two adjacent anatomical features: (i) the 4th ventricle and; (ii) median raphe nucleus (MRN). There was no evidence that policy-switches affected BOLD signal in these areas ($t_{DRN}(26) = 2.85$, $p = .009$, $t_{MRN}(26) = -0.36$, $p = .724$, $t_{Vent.}(26) = -0.83$, $p = .415$). Moreover, the policy-switch effect in DRN was greater than the effects in MRN and the 4th ventricle ($t_{DRN-vs-MRN}(26) = 2.08$, $p = .047$, $t_{DRN-vs-Vent}(26) = 2.31$, $p = .028$).

2. Second, rather than just showing signal averaged across ROIs, displays of whole-brain maps of the key effects of interest (e.g., policy-switches as a function of environmental richness, the PPI etc) can significantly increase confidence, also because those enable assessment of the physiological plausibility of the reported effects (are effects clustered, do they look artefactual, are they bilateral etc).

We thank the reviewer for their suggestion. We did not report whole-brain maps because our regions of interest consist of small subcortical nuclei. The small size of these structures poses a particular challenge when using conservative cluster-based correction methods, such as those implemented in FSL, which can substantially increase type II error rates.

Nevertheless, to address the reviewer's concern, we conducted a new whole-brain analyses looking for the key effect of policy-switch in the whole-brain using the following GLM:

$$BOLD_{v,t} = \beta_1 \text{opportunity-onset} + \beta_2 \text{trial-number} + \beta_3 \text{policy-switch}(\text{opportunity-onset}) + \beta_4 \text{action-switch}(\text{opportunity-onset}) + \beta_5 \text{go-cue-onset} + \beta_6 \text{decision}(\text{go-cue-onset}) + \beta_7 \text{feedback-onset} + \beta_8 \text{reward-feedback}$$

This GLM corresponds with GLM4.2 in the manuscript but is incorporated into the whole-brain GLM (GLM3.1). As expected, no significant cluster in the brainstem survives correction for the policy-switch effect. However, at a threshold of $Z > 2.3$ (approximately $p = 0.01$), we observe activity centred within our DRN mask (shown in blue). Importantly, this activity does not appear elsewhere in the brain and does not reflect noise or spillover from the ventricles or adjacent regions. Supporting this, when we lower the threshold to $Z = 2$, the activity expands consistently within the anatomical boundaries of the DRN.

These results are added to the manuscript as supplementary figure S4

Figure S4. Effect of policy-switch in the whole-brain. We further examined the relationship between policy-switches and brain activity using a GLM with the same predictors as GLM4.1 in the main manuscript but implemented at the whole-brain level (GLM3.1). For the policy-switch effect no clusters in the brainstem survived correction. However, at a threshold of $Z > 2.3$ (approximately $p = 0.01$), we observed activity centred within our DRN mask (shown in blue). Importantly, at this threshold, there were no statistically significant clusters of activity elsewhere in the brain, and nor did the DRN-related cluster reflect noise or spillover from the ventricles or adjacent regions. Furthermore, lower the threshold to $Z = 2$ indicated that the cluster expanded in a manner that was consistent with DRN's anatomical boundaries.

3. Third, were effects in DRN compared directly with effects in other ROIs (Ic, dopa midbrain etc)?

We found strong evidence for policy-switch in DRN but no evidence for similar effect in other ROIs. However, the reviewer is correct that the DRN effect could also be directly compared with other ROIs. As reported in the Results section on page 7, we already showed a significant difference between activity patterns in the DRN and the MS: the

difference between environment-driven DRN and exploratory-driven MS patterns was statistically significant ($ANOVA_{ROI \times change-type} = 10.39, p = .001$; see supplementary fig.S7).

As requested by the reviewer, we now also compare the activity pattern between DRN and MBD, LC and habenula. We show that the effect of policy-switches on brain activity is significantly different in DRN compared to MBD ($t(26)=3.20, p=.003$), and in DRN compared to Habenula ($t(26)=2.69, p=.012$). Although there was no evidence that policy switches in general differently affected brain activity in DRN and LC ($t(26)=0.99, p=.330$), policy switches that were congruent to the present environment were, indeed, represented differently in DRN activity compared to LC activity ($t_{congruent-poor}(26)=4.79, p<.001, t_{congruent-rich}(26)=-2.78, p=.009$). Furthermore, unlike in DRN, there was no difference in the representation of congruent policy switches between rich and poor environments in LC ($t_{congruent-rich \text{ vs } congruent-poor}(26)=-1.11, p=.273$). This suggests that DRN – and DRN alone – distinctively represented switches in behavioural policy depending on whether the switch was adaptive given the present environment.

We have now added the following paragraph to the results:

This indicated that activity in DRN – but no other subcortical ROI – represented policy-switch trials (GLM4.1a; $t_{DRN; policy-switch}(26)=2.84, p=.043$; fig.3B). The DRN effect was significantly different from the effects in MBD ($t_{DRN-vs-MBD; policy-switch}(26)=3.20, p=.003$), and habenula ($t_{DRN-vs-Hb; policy-switch}(26)=2.69, p=.012$). Although there was no evidence that the effect of policy switches in DRN was in general different from LC ($t_{DRN-vs-LC; policy-switch}(26)=0.99, p=.330$), environment-driven policy switches (see next paragraph) were, indeed, represented differently in DRN activity compared to LC activity ($t_{DRN-vs-LC; congruent-poor}(26)=4.79, p<.001, t_{DRN-vs-LC; congruent-rich}(26)=-2.78, p=.009$).

- 4. Fourth, some of the key conclusions seem grounded in p-values that hover around 0.05, with great emphasis on effects just below the 0.05 threshold (such as the environment-driven policy switch effect in DRN), and dismissal of effects just above that 0.05 threshold (such as the environment-driven policy-switch effect in some cortical regions). Can the authors convince the readers that their key take-home message (which implies a double dissociation) that ‘cortical components [but not DRN] construct an environment-specific representation of each option, while DRN [but not cortical regions] implements the changes in behavioural policy consequent on these environment-specific changes’ still stands despite this reliance on this arbitrary threshold? Presentation of relevant whole-brain maps (of both the policy-switch contrast, as well as the RSA) and/or direct comparisons of ROIs might help?**

We acknowledge that some of our effects are small despite being statistically significant. However, it is important to point out that we follow these results with several follow-up analyses to strengthen our conclusions. For example, we first show that DRN – but no other subcortical ROI – represented policy-switch trials ($p=.043$). We then follow-up this result by showing that this effect was specific to behavioural-policy changes ($p=0.002$), and that the effect was driven by the richness of the environment ($p=0.025$), and that the result was stronger when the policy switch was congruent with the environment ($p=0.019$) and was specific to the 10-point option ($p=0.018$). So, in summary, we show a series of results in which we gradually narrow down an initially coarser test to ascertain more precisely which factors are actually driving activity in a given region.

However, to strengthen our claim we have performed the analyses requested by the reviewer:

1. We present the policy-switch effects at whole brain (see response to comment #2).
2. DRN activity had two critical features: (i) it signalled policy-switches that were congruent to the present environment, and; (ii) the representation of congruent policy-switches in poor environments was distinct from the representation in rich environments. We directly contrasted the strength of this pattern in DRN and cortical (dACC and AI) ROIs by performing the following timecourse analysis:

$$\text{BOLD}_{i,t} = \beta_0 \text{constant} + \beta_1 \text{congruent-poor-vs-congruent-rich} + \beta_2 \text{trial-number}$$

This confirmed that congruent policy-switches in poor and rich environments were accompanied by distinct changes in DRN activity ($t_{\text{DRN}; \text{congr-rich-vs-congr-poor}}(26) = 4.03, p=.001$). There was no evidence of this pattern in either AI ($t_{\text{AI}; \text{congr-rich-vs-congr-poor}}(26)=1.73, p=.189$) or dACC ($t_{\text{dACC}; \text{congr-rich-vs-congr-poor}}(26)=0.28, p=.779$) activity. Indeed, the regression effect contrasting congruent-poor and congruent-rich policy switch events was greater in DRN than in AI and dACC ($t_{\text{DRN-vs-AI}}(26)=4.02, p<.001$; $t_{\text{DRN-vs-dACC}}(26)= 3.21, p=.003$). This suggests that the environment- and congruence-specificity of policy-switch representations was stronger in DRN than in dACC and AI ROIs.

We have added this new analysis to the manuscript:

In contrast to the clear evidence of multivariate pattern change, there was no evidence of univariate activity changes in dACC or AI as a function of experimentally manipulated environment-type ($t_{\text{dACC}; \text{env-type}}(26)= 1.52, p=.282$; $t_{\text{AI}; \text{env-type}}(26)= 0.48, p=.636$). Similarly, there was very limited evidence that dACC or AI represented environment-driven policy-switches in the manner of DRN ($t_{\text{dACC}; \text{congr-pursue}}(26)=-1.00, p=.325$; $t_{\text{dACC}; \text{congr-reject}}(26)= -2.28, p=.061$; $t_{\text{AI}; \text{congr-pursue}}(26)=1.71, p=.196$; $t_{\text{AI}; \text{congr-reject}}(26)=-2.11, p=.061$; GLM4.2a and 4.2c; fig.4E). **Indeed, directly contrasting the latter effects indicated that the environment- and congruence-specificity of policy-switch representations was stronger in DRN compared to cortical ROIs ($t_{\text{DRN-vs-AI}}(26)=4.02, p<.001$; $t_{\text{DRN-vs-dACC}}(26)= 3.22, p=.003$), consistent with the hypothesis that DRN implemented changes in behavioural policy.**

Finally, we note that in the process of the revision we discovered a minor bug in our analysis code. Some of parameter estimates and p-values have changed after rectifying this bug, however all such changes very minor, and there is not a clear trend for results to have become closer or further away from statistical significance. Perhaps most critically, none of the results have changed from being significant to non-significant or vice versa. We note this simply to avoid any confusion that small changes in the numerical values of statistical results may have occasioned if they were indeed noticed.

5. **Finally, the wordings of the key conclusion in the abstract ('that the human DRN represents changes between general behavioural policies') and in some parts of the discussion ('that DRN has a conserved function of matching the general policy governing behaviour with the general reward statistics of the environment') seem overly general, and might benefit from specification. Many neuromodulatory systems have been argued to implement behavioral strategy change, and there are many different statistical parameters that can describe the reward structure of an animal's environment. The key question seems to be *which type* of behavioral strategy decision is carried by which system, and as a function of *which specific* feature of the statistical structure of its environment.**

We thank the reviewer for pointing this out. We have changed the wording in the abstract and discussion to make it more specific.

The sentence ‘that the human DRN represents changes between general behavioural policies’ was changed to:

We build on recent results from animal models to demonstrate that activity in human DRN implements changes in behavioural policy that reflect the distribution of rewards in the environment.

The sentence ‘that DRN has a conserved function of matching the general policy governing behaviour with the general reward statistics of the environment’ has been expanded upon in the following passages of the discussion:

Pinpointing the trials when policy switches occurred allowed us to identify changes in DRN activity that signalled these switches. This pattern was specific to DRN, and specific to switches that were appropriate to the richness of the environment. Together with previous studies, this suggests that DRN and serotonin have a conserved role in controlling switches between fundamental aspects of motivated behaviour. For example, Marques and colleagues identified a population of dorsal raphe neurons in zebrafish in which activity covaried with transitions between food-oriented foraging and exploration (Marques et al., 2020).; Priestley and colleagues identified DRN activity in the macaque during transitions between periods of engagement and disengagement with reward-oriented behaviour (Priestley et al., 2025); Trier and colleagues linked activity changes in DRN to transitions between reward-oriented behaviour and threat-oriented behaviour (Trier et al., 2025); finally, Ahmadlou and colleagues linked the median raphe nucleus – an adjacent nucleus with prominent serotonergic projections – to controlling the balance between persisting with the current behaviour or switching to an exploratory mode (Ahmadlou et al., 2025). The aspect of behavioural policy investigated in the present study – whether to accept a reward opportunity of a given magnitude – was comparatively simple and specific, but has nevertheless been a fundamental concern for foraging organisms throughout evolutionary history.

In the present results, changes in behavioural policy were linked to the task’s underlying reward rate. This is consistent with recordings of single-neuron activity implicating DRN in environment-driven changes in behaviour, including changes between patience and impulsivity during intertemporal choice (K. Miyazaki et al., 2011; K. W. Miyazaki et al., 2014b), exploitation and exploration during reversal learning (Clarke et al., 2004; Grossman et al., 2022; Lottem et al., 2018b; Matias et al., 2017). Moreover, it complements a string of recent studies indicating that the reward and/or changes in the reward rate are fundamental determinants of DRN activity (Harkin et al., 2025; Luo et al., 2016; Wittmann et al., 2020). The implications of the background reward rate for adaptive behaviour have long been emphasised in behavioural ecology, and the present study was indeed inspired by the so-called diet-selection task – a ecology canonical scenario which involves deciding which food items are worth pursuing and which are not (Stephens & Krebs, 1986b). In a previous experiment, Lottem and colleagues drew inspiration from a different foraging paradigm called the patch leaving problem, in which animals must decide how long to persist with food-seeking at one location given the possibility of foraging elsewhere (Lottem et al., 2018a). Optogenetic stimulation of DRN neurons in this task promoted active foraging in the current patch for further potential rewards, similar to the way that DRN activity increased in the present study when participants actively sampled a moderate-value option that was worthwhile only in poor environments. Taken together, these observations emphasise that DRN activity’s behavioural effect is not reducible to behavioural inhibition, as posited in classic theories of serotonin function (Soubrié, 1986a). Instead, they cohere with an emerging body of work in which DRN and adjacent raphe nuclei control changes along fundamental dimensions of behavioural policy (Ahmadlou et al., 2025; Marques et al., 2020; Priestley et al., 2025; Trier et al., 2025).

Other comments:

- 6. Jaskir and Frank (2023, Elife) have put forward a relevant theoretical (normative) framework for predicting effects of environmental richness of behavioral activation (go responses) vs inhibition (nogot responses). Seems relevant and can the results be discussed? In this context, I wondered, would the same pattern of results have been predicted if subjects would have had to choose between 2 active response options instead of a Go vs Nogo response?**

This is an interesting point and we thank the reviewer for this. Our findings provide fMRI evidence in humans for a context-sensitive neuromodulatory mechanism, complementing the computational predictions of the OpAL model in the cited paper. However, while Jaskir and Frank emphasise dopaminergic opponency in the striatum, our data suggest that the serotonergic systems (DRN) might play a parallel or upstream role, particularly in implementing rather than evaluating policy changes. In particular, our observed DRN-dACC circuit aligns with OpAL*'s separation between value representation and policy execution. We have now discussed the cited paper in the manuscript on page 13:*

Our results complement recent computational work demonstrating that dynamic neuromodulatory mechanisms—specifically opponent interactions in dopaminergic circuits—support flexible policy adaptation in response to environmental reward structure (Jaskir & Frank, 2023). Here, we suggest that serotonergic systems may play an analogous role in implementing such adaptive policy switches in the human brain. Similarly, our results are broadly in-line with Ligneul and Mainen's hypothesis that DRN mediates behavioural adaptation to changing environments (Ligneul & Mainen, 2023). Ligneul and Mainen emphasise the role of prediction errors in this function and we, also, have reported prediction error-like activity patterns in DRN in our previous work (Trier et al., 2025). Nevertheless, the present results suggest that this is only one aspect of DRN function. The moderate-value options to which DRN responded in the current study always occurred at the same rate in both poor and rich environments, and so in a simple sense were no more surprising in one environment than the other. The moderate-value options did, however, constitute deviations from the long-term reward rate of the environment that prompted periodic changes in behavioural policy. In the present results, DRN activity was most strongly associated with these events.

The reviewer also asks whether the same pattern would have emerged if participants were choosing between two options, rather than making a Go vs. NoGo response. This is an interesting question, and we believe the pattern would persist, as our policy-change effects appear to be independent of specific motor actions. We hope that this apparent from reconsideration of figure 2A. However, this remains an empirical question, and we hope to address it soon through follow-up studies that are currently underway.

- 7. To what degree is it possible that the effect of environmental richness surfaces for the 10p option most clearly because of greatest dynamic range/uncertainty?**

We thank the reviewer for their question. However, we would like to clarify that the value of each option was deterministic, constant across the task, and known to participants in advance. The task did not involve any uncertainty about the payout of options, nor did it require learning stimulus-reward associations. Instead, the richness of the environment was manipulated solely through changes in the frequency of different options. Therefore,

we do not believe that the 10-point option was associated with greater uncertainty than the others. Rather, it was uniquely positioned to require a context-dependent policy: it was optimal to pursue in poor environments and to reject in rich ones. This strategic ambiguity — not increased uncertainty — likely explains why it was most sensitive to environmental richness.

- 8. Stating that there was ‘a null effect’ require statistical testing of null effects. In the absence of such explicit (Bayesian) testing, it would be more accurate to state that ‘there was no evidence for an effect...).**

We agree with the reviewer and have changed the wording accordingly throughout the manuscript.

- 9. Two participants were excluded from behavioural analyses because they did not perform the task correctly. Please clarify**

We have now added this information to the Methods section.

Two participants were excluded from behavioural analyses because they did not perform the task correctly. **In both cases participants clearly failed to follow instructions and attempted to respond to all stimuli on nearly every trial.** An additional two participants were excluded from fMRI analyses due to excessive head motion which prevented accurate registration (motion outliers > 15% of total fMRI volumes). **Self-reported sex of participants was not considered during analysis.**

Reviewer #2 (Remarks to the Author):

Serotonin is thought to play an important but poorly-defined role in valence-related aspects of behaviour and learning. In this work, Priestley et al. investigate the relationship between the activity of the DRN (the main source of forebrain serotonin) and valence-related changes in human behaviour using high-resolution fMRI and a task inspired by the diet selection problem of foraging theory. Compared with the related literature, this study makes three main contributions: First, the authors directly implicate the DRN in long-term (multi-trial) valence-related changes in behaviour. Long-term aspects of reward-seeking/foraging-like behaviour are difficult to study and have received relatively little attention in the literature, making this result quite significant on its own. Second, the ability to record the output of multiple neuromodulatory nuclei at a time raises exciting possibilities to clearly differentiate their functions, and the authors show that the signal driving the long-term behaviour changes that they study is specific to the DRN. Third, this study tackles a diet selection problem with an interesting inter-temporal choice component that is overall quite different from the waiting, patch residence, bandit, and conditioning tasks in the existing literature, possibly highlighting previously unstudied aspects of DRN function.

On a technical level, the analyses are generally sound, thorough, and support the main conclusions. The methods are generally quite clear, the data are nicely formatted, and the code looks clean (although I didn't run it myself). It is great to see that the authors have put genuine effort into increasing the reproducibility of their work.

For now, I am left feeling that the potential of this manuscript was not fully realized. Although the main text tells a clear story in which policy switches are driven by changes in the environment, a sceptical reader who only skims the figures might get a different impression. Discussion of non-serotonergic neuromodulators is very limited apart from confirmatory results about dopamine. The novel task is not used to rule out any of the many existing theories of serotonergic function. By introducing an intriguing policy switch-based model of DRN function that does not seem to build on previous models, the authors have set themselves up for the daunting task of making a convincing case for a new interpretation of serotonergic function in an already crowded field. In my opinion, harmonizing the figures and main text, placing the new model in greater context, and clarifying the implications of these data for existing theories would significantly strengthen the paper and help move the serotonin field towards consensus.

We are very grateful for the reviewer's careful and thoughtful comments. We are especially grateful for the reviewer's comments regarding the novelty of our perspective and results and the soundness of our analyses. We understand that the reviewer has written about the way in which many aspects of the results are interesting and we are grateful that they have done so. We can see also that the reviewer has suggested that we have done so by constructing a "policy switch-based model of DRN function that does not seem to build on previous model". However, we would contend that our paper, while the first to investigate this issue in humans does link up to an emerging body of work implicating the DRN or adjacent median raphe nucleus (MRN) in behavioural switching in non-human primates (Priestley et al., Science Advances, 2025) and mice (Ahmadlou et al., Nature, 2025). Interestingly, a recent paper on the local circuit organisation of DRN suggests a winner-take-all computation between ensembles of recurrently connected 5-HT neurons in

the DRN, operating over protracted timescales. At a behavioural level, this type of computation is ideal for facilitating transitions between behavioural states that require sharp thresholds (Lynn et al., 2025, Nature Neuroscience). We concede that this work is very novel and is only now emerging since the submission of our manuscript but we think that our report is a part of a coherent set of results that are now being reported across species. We have attempted to make greater reference to these studies, which are now published or in press, in our revised manuscript. We therefore note in our revised manuscript:

Instead, they cohere with an emerging body of work in which DRN and adjacent raphe nuclei control changes along fundamental dimensions of behavioural policy (Ahmadlou et al., 2025; Marques et al., 2020; Priestley et al., 2025; Trier et al., 2025).

Major

1. Implications for existing theories of serotonergic function

The authors are correct to say that there is “little consensus” about the behavioural function of the DRN, and it would be fair to say that there is similarly little consensus about what DRN activity represents. These data seem potentially quite interesting in terms of their implications for existing theories of serotonergic function (see below), and it would be nice to see a clearer and more direct presentation of any specific results that do or don’t support these theories. It is up to the authors to decide how to present their story, but my suggestion would be to start by preparing a table listing their main results and whether or not they are consistent with each of the dominant theories. Once this is done, the authors could consider incorporating key points from this table into the results and/or discussion, potentially supported by new analysis. I believe that emphasizing the contribution of the present work to the evidence base for and against existing theories would increase the impact of the paper by helping to move the field towards consensus.

A few theories that I think would be particularly important to consider:

- Behavioural inhibition (Soubrié, Behav Brain Sci, 1986):** Since the behavioural task involves withholding an action in order to proceed to the next opportunity, the authors could comment on whether their data support a general role for the DRN in behavioural inhibition. My initial interpretation is that these results are probably not consistent with behavioural inhibition because DRN activity does not correlate with accept/reject decisions on a trial-by-trial basis.
- Discounting (Doya, Neural Networks, 2002):** Since the task has an intertemporal choice component, it would be interesting to consider whether a change in discounting initiated by the DRN could explain some of their results. The authors already touch on some relevant work from the Doya lab, and it would be great to expand on this. In my view, a persistent decrease in discounting in poor environments seems like a plausible way to explain the present results.
- Surprise/prediction error (Ligneul and Mainen, Current Biol, 2023):** Serotonin has been proposed to signal surprising/salient events in general in order to facilitate behavioural adaptation to changing environments. Since some offers in this task are rarer than others, the authors could comment on whether DRN activity is correlated with surprising offers.
- “Beneficialness” (Luo et al., Neurobiol Learn Memory, 2016):** As the authors note, serotonin has been repeatedly implicated in signalling positive valence (along the lines of value or reward). Since policy switches should reflect a change in the estimated long-term value of the environment, long-term value of accepting a particular option, or both, it would be good to consider whether the observed

results might be consistent with one of these. In light of the fact that short-term and long-term value can look quite different, and that a large part of serotonin neuron activity seems to be related to changes in value rather than value per se (Harkin et al., Nature, 2025), the focus of the present study on changes in multi-trial aspects of behaviour seems quite interesting. On the other hand, the authors could consider whether the present results are more consistent with the proposal of Marques et al. (Nature, 2020) that serotonin controls value-independent policy switches.

We are very grateful for the reviewer's extremely thoughtful comments. Although we realise that many of the reviewer's comments are positive, we are also aware that the reviewer suggests that we might have focussed on testing existing accounts of serotonin function. As explained above, already, we feel that our results are actually consistent with a newly emerging picture of DRN function, admittedly partly in papers published since the original submission of our paper. Moreover, we note that our investigations are based on many years of conducting often difficult experiments on related issues in primates, both human and non-human, in our laboratory (Wittmann et al., Nature Communicants, 2020; Khalighinejad et al., Current Biology, 2022; Trier et al., PNAS, 2025; Priestley et al., Science Advances, 2025).

To turn to the more general and philosophical question of how a programme of scientific research should proceed, we appreciate that there are different views about how one should conduct experiments just as there are theories about how one should make choices between different types of options in other areas of human endeavour. One view, however, is that an important element of making effective progress when two or more options are offered is to avoid simply choosing between them but instead to create a new path to consider alongside the existing options. The construction of such a new path, that draws on the most appealing aspects of the options offered, is often the way in which progress is made in science and in other domains (Iyengar, 2009, The Art of Choosing). Just as this approach can be taken in everyday life, so it can be taken in science and this is what we have tried to do in our work.

We contend that we have tried to draw on our own previous work and on the work of the other authorities that the reviewer cites, for example the Miyazakis and Doya, Monosov, Cools, Mainen as well as many others, and we note that we have cited their work throughout our manuscript. We have not previously cited the very interesting paper of Harkin and colleagues (Nature, 2025) simply because it was not published at the time that we submitted our article. Nevertheless, we draw on our own earlier work which has spent a long time under review making a closely related point that the DRN encodes transitions in value. We note that the Harkin paper does a particularly good job of reviewing previous work in the field, perhaps because it critically relies on accessing the data of other laboratories and so we take care to cite this article in our revised manuscript not just as a source of an interesting new proposal but as a useful review of the state of the field.

Our understanding is that it is not possible for us to attempt an extensive review of the literature with tabulations of the sort suggested by the reviewer in the current manuscript and the 5,000-word limit of Nature Communications articles. Such an approach is indeed often valuable but it is perhaps more appropriate for a review article than a report such as the present one and we note that our understanding is that the Discussion section in our current manuscript cannot be even as long as the reviewer's review. However, while we are not able to fully implement all the reviewer's suggestions we acknowledge that the spirit of them

is correct and we have tried to extend our Discussion and to include additional consideration of some of the previously published studies that the reviewer mentions. In the revised manuscript we have included the following sections:

“...Similarly, our results are broadly in-line with Ligneul and Mainen’s hypothesis that DRN mediates behavioural adaptation to changing environments (Ligneul & Mainen, 2023). Ligneul and Mainen emphasise the role of prediction errors in this function and we, also, have reported prediction error-like activity patterns in DRN in our previous work (Trier et al., 2025). Nevertheless, the present results suggest that this is only one aspect of DRN function. The moderate-value options to which DRN responded in the current study always occurred at the same rate in both poor and rich environments, and so in a simple sense were no more surprising in one environment than the other. The moderate-value options did, however, constitute deviations from the long-term reward rate of the environment that prompted periodic changes in behavioural policy. In the present results, DRN activity was most strongly associated with these events.”

We have also added the following sections to the Discussion:

Together with previous studies, this suggests that DRN and serotonin have a conserved role in controlling switches between fundamental aspects of motivated behaviour. For example, Marques and colleagues identified a population of dorsal raphe neurons in zebrafish in which activity covaried with transitions between food-oriented foraging and exploration (Marques et al., 2020).; Priestley and colleagues identified DRN activity in the macaque during transitions between periods of engagement and disengagement with reward-oriented behaviour (Priestley et al., 2025); Trier and colleagues linked activity changes in DRN to transitions between reward-oriented behaviour and threat-oriented behaviour (Trier et al., 2025); finally, Ahmadlou and colleagues linked the median raphe nucleus – an adjacent nucleus with prominent serotonergic projections – to controlling the balance between persisting with the current behaviour or switching to an exploratory mode (Ahmadlou et al., 2025). The aspect of behavioural policy investigated in the present study – whether to accept a reward opportunity of a given magnitude – was comparatively simple and specific, but has nevertheless been a fundamental concern for foraging organisms throughout evolutionary history.

In the present results, changes in behavioural policy were linked to the task’s underlying reward rate. This is consistent with recordings of single-neuron activity implicating DRN in environment-driven changes in behaviour, including changes between patience and impulsivity during intertemporal choice (K. Miyazaki et al., 2011; K. W. Miyazaki et al., 2014b), exploitation and exploration during reversal learning (Clarke et al., 2004; Grossman et al., 2022; Lottem et al., 2018b; Matias et al., 2017). Moreover, it complements a string of recent studies indicating that the reward and/or changes in the reward rate are fundamental determinants of DRN activity (Harkin et al., 2025; Luo et al., 2016; Wittmann et al., 2020). The implications of the background reward rate for adaptive behaviour have long been emphasised in behavioural ecology, and the present study was indeed inspired by the so-called diet-selection task – a ecology canonical scenario which involves deciding which food items are worth pursuing and which are not (Stephens & Krebs, 1986b). In a previous experiment, Lottem and colleagues drew inspiration from a different foraging paradigm called the patch leaving problem, in which animals must decide how long to persist with food-seeking at one location given the possibility of foraging elsewhere (Lottem et al., 2018a). Optogenetic stimulation of DRN neurons in this task promoted active foraging in the current patch for further potential rewards, similar to the way that DRN activity increased in the present study when participants actively sampled a moderate-value option that was worthwhile only in poor environments. Taken together, these observations emphasise that DRN activity’s behavioural effect is not reducible to behavioural inhibition, as posited in classic theories of serotonin

function (Soubrié, 1986a). Instead, they cohere with an emerging body of work in which DRN and adjacent raphe nuclei control changes along fundamental dimensions of behavioural policy (Ahmadlou et al., 2025; Marques et al., 2020; Priestley et al., 2025; Trier et al., 2025).

2. Need for a new theory.

This paper proposes that the DRN initiates changes in behaviour that are specifically related to long-term changes in reward availability. While this perspective touches on some common themes in the serotonin literature — that serotonin has something to do with changes in behaviour, reward availability, foraging, or adapting to adverse circumstances — this precise idea seems new to me. In my view, this novelty is a double-edged sword: On the one hand, this work provides a new way of thinking about DRN function with the potential to inspire new theories and experiments. On the other hand, since this paper does not seem to rule out any of the existing theories of DRN function, it risks becoming one more serotonergic idea among many. I believe the paper would be strengthened if the authors could clarify whether the policy switching model should be interpreted as an extension of some existing model, or, if not, exactly why a new perspective is needed. I would find it particularly helpful to understand how the policy switching model relates to the following:

- **Marques et al. (Nature, 2020)** also propose that the DRN controls behavioural state transitions during foraging behaviour in zebrafish. In their case, DRN serotonin neurons promote a switch from an exploration-like global search for prey to an exploitation-like local search. Interestingly, this effect does not seem to be directly related to environmental richness, since the DRN periodically initiates local search-like behaviour even in a prey-free environment.

In the revised manuscript we have tried to address the links and differences between our current results and the previously published work of Marques and colleagues as shown below. We have also included references to the study by Harkin and colleagues (2025) and the study by Ahmadlou and colleagues (Nature, 2025) both of which were published some time after our manuscript was submitted.

Together with previous studies, this suggests that DRN and serotonin have a conserved role in controlling switches between fundamental aspects of motivated behaviour. For example, Marques and colleagues identified a population of dorsal raphe neurons in zebrafish in which activity covaried with transitions between food-oriented foraging and exploration (Marques et al., 2020); Priestley and colleagues identified DRN activity in the macaque during transitions between periods of engagement and disengagement with reward-oriented behaviour (Priestley et al., 2025); Trier and colleagues linked activity changes in DRN to transitions between reward-oriented behaviour and threat-oriented behaviour (Trier et al., 2025); finally, Ahmadlou and colleagues linked the median raphe nucleus – an adjacent nucleus with prominent serotonergic projections – to controlling the balance between persisting with the current behaviour or switching to an exploratory mode (Ahmadlou et al., 2025). The aspect of behavioural policy investigated in the present study – whether to accept a reward opportunity of a given magnitude – was comparatively simple and specific, but has nevertheless been a fundamental concern for foraging organisms throughout evolutionary history.

- **Lottem et al. (Nat Comm, 2018)** also study serotonergic function from a foraging perspective. However, whereas the present work focuses on a diet selection problem and shows that the DRN controls changes in “diet” (increase in activity =>

start accepting medium option, decrease in activity => start rejecting medium option), Lottem et al. focus on a patch foraging

In response to the reviewer's comments we have added the following section:

In a previous experiment, Lottem and colleagues drew inspiration from a different foraging paradigm called the patch leaving problem, in which animals must decide how long to persist with food-seeking at one location given the possibility of foraging elsewhere (Lottem et al., 2018a). Optogenetic stimulation of DRN neurons in this task promoted active foraging in the current patch for further potential rewards, similar to the way that DRN activity increased in the present study when participants actively sampled a moderate-value option that was worthwhile only in poor environments. Taken together, these observations emphasise that DRN activity's behavioural effect is not reducible to behavioural inhibition, as posited in classic theories of serotonin function (Soubrié, 1986a). Instead, they cohere with an emerging body of work in which DRN and adjacent raphe nuclei control changes along fundamental dimensions of behavioural policy (Ahmadlou et al., 2025; Marques et al., 2020; Priestley et al., 2025; Trier et al., 2025).

3. Presentation of the policy switching behavioural effect in text vs. figures

The premise of the paper is that humans make decisions about whether to accept or reject reward offers based on durable inferences about whether these offers are worth their time. These inferences are reflected in patterns of behaviour (policies), with policy transitions being influenced by the DRN. The strength of this story hinges on whether a (potentially sceptical) reader could reasonably be convinced that the changes in behaviour in this task reflect such durable inferences. In my view, the presentation of the results in the main text is clear and compelling, but the key figures (Fig. 1E, 1G, 2B, and 2D) are less convincing. I'm unsure whether this means that the text should be reconciled to the richness of the figures by toning down some of the language, or whether the figures should be reconciled to the text by visually drawing out key results, so I've highlighted the areas where I felt there was a mismatch and will leave it to the authors to judge how best to adjust.

We thank the reviewer for highlighting these points. In response, we have made the following changes to address the areas identified:

- 1. As the reviewer rightly points out, it is unrealistic to expect participants to behave optimally. In Figure 1E, our aim was to examine whether the type of environment (rich vs. poor) influenced pursue-vs-reject decisions. The results showed that participants were more likely to pursue the 10-point option—but not the 5- or 50-point options—in poor compared to rich environments. This pattern is consistent with predictions from an optimal policy, which would suggest adapting behaviour based on environmental richness.*

Moreover, in fig.1F we show that the subtle difference in pursue-rates illustrated in fig.1E becomes more pronounced as participants accumulate experience with their environmental context. This suggests that participants do, indeed, ultimately converge on an adaptive policy, but that they need time to disambiguate which environment they are in. In line with the reviewer's suggestions, we have clarified our interpretation of the data in the main text and fig.1F captions as follows:

“...This pattern of results was confirmed by a conceptually similar analysis in which the environment was operationalised as the average value of recently encountered reward-options, as distinct from the experimentally defined environments (GLM1.3; see supplementary fig.S1C-D). Taken together, these analyses show that adaptive changes in behaviour toward the 10-point option emerged gradually within each block, indicating that participants required time and experience to ascertain the richness of the present context.”

“...(F) Environments were not explicitly signalled to subjects and needed to be inferred from the relative frequency of reward-options over time. We therefore tested whether adaptive changes in behaviour toward the 10-point option emerged gradually over time within each block. As predicted, the difference in middle-option pursue-rates (y-axis) for poor-environments and rich-environments increased as a function of time spent in an environment (x-axis). There were no such changes for 5-point and 50-point options. Dots and whiskers indicate mean \pm SEM of pursue-rate in each level of trial-within-environment (x-axis)...”

2. *There is indeed a statistically significant relationship between the total reward earned during the experiment and the degree to which participants change their behaviour toward the 10-point option between rich and poor environments. To emphasise this, we support the claim with a new GLM analysis in which total reward earned is predicted as a function of (i) the pursue-rate for the 10-point option in the poor environment; (ii) the pursue-rate for the 10-point option in the rich environment, and; (iii) the two-way interaction between the aforementioned predictors. We have added this GLM analysis for two reasons: (i) to provide a more robust test of the claim that behavioural change is related to task performance than the previous correlation analysis, and; (ii) to avoid confusion about whether fig.1G should be directly interpreted as a statistical test. The results of this analysis appear in the main text as follows:*

Importantly, pursue-rates for the 10-point option in poor environments – but not pursue-rates for the 10-point option in rich environments – were positively associated with total reward accumulated in the experiment ($\beta_{\text{Poor}}=3.94$; $SE=1.63$; $p=.024$; $\beta_{\text{Rich}}=1.63$; $SE=1.92$; $p=.405$). This was complemented by a two-way interaction whereby the benefit of pursuing the 10-point option in the poor environment was inversely proportional to pursue-rates in the rich environment ($\beta_{\text{Poor*Rich}}=-5.42$; $SE=2.29$; $p=.027$; fig.1G). The degree to which participants changed their behaviour between environments thus determined their performance on the task, as envisaged in the experimental design.

We have updated the figure legend to explicitly state that the relationship is significant when analysed in a GLM, and that the figure is intended only to illustrate the conceptual inferences drawn from this GLM. We have also modified the Y-axis label to make the figure clearer. The y-axis now shows z-scored total-points earned in the experiment and x-axis shows mean difference in middle-option (10point) pursue-rates between poor-environments and rich-environment [poor – rich]. The updated figure and figure legend are as follows:

“... (G) The degree to which subjects changed their behaviour between environments predicted their success on the task when analysed in a GLM. Y-axis denotes z-scored total-points earned in the experiment and x-axis denotes mean difference in middle-option pursue-rates between poor-environments and rich-environments [poor – rich]. Dots represent data from individual subjects.

3. *Importantly, behavioural policies are operationalised with respect to individual reward options. Only one reward option is presented on each trial. The 10-point reward option occurred with a constant probability of $p(10\text{-pt})=0.33$ across both rich and poor environments. The length of an average block was 17.0 trials ($SD=1.96$), implying that participants should encounter the 10-pt option 5.66 times on average, which was indeed the case ($M=5.89$, $SD=1.76$). As a result, the reviewer’s logic is consistent with the notion that participants adopted durable policies that lasted approximately the length of a block. In further support of this, the average number of policy-switches per block was 1.56 ($SD=1.65$).*

*It is difficult to pinpoint which policy-switches were undertaken specifically due to changes in the environment. This is because policy-switch decisions would have depended on a participant’s latent belief about the richness of the present context, rather than experimentally programmed changes in block-type. One way of probing the question is to quantify **when**, within a block, policy-switches tend to occur. In the figure below, we show the probability density function for policy-switches as a function of time-within-block. This suggests that policy-switches tended to occur at the midpoint of blocks, consistent with our earlier argument that behavioural adaptations to the environment tend to occur after participants have accumulated sufficient evidence that the environment has changed.*

Taken together, these analyses suggest that participants tended to change their policy in relation to the 10-point option on 1-2 occasions per block, and that most changes occurred near the midpoint of blocks at which point it would be clear that the richness of the environment had changed. We believe that these observations are consistent with the claim that participants adopted durable behavioural policies that were determined by the perceived richness of the environment.

We have now added this new figure and analysis to fig.S2:

(C) Policy switches tended to occur at or after the midpoint of blocks, consistent with the hypothesis that behavioural adaptations to the environment required participants to accumulate evidence that the environment has changed. Further analysis indicated that congruent policy-switches in rich blocks ($M=7.81$, $SD = 2.51$) occurred earlier than congruent policy-switches in poor blocks ($M=10.8$, $SD=2.94$), and this difference was statistically significant ($\beta_{rich-vs-poor}=-0.23$, $SE=0.06$, $p<.001$). Squares indicate mean timepoint at which switches occur at the sample level, and dots indicate means for individual participants.

Minor

4. Relationship between BOLD signal and serotonergic output

As the authors note in the discussion, the fMRI BOLD signal is not cell-type specific, so it is difficult to know how this signal relates to serotonergic output. I am certainly no fMRI expert, but it seems plausible to me that the BOLD signal might even be the mirror image of serotonergic output (in which case the data are still valuable!) for two reasons. First, a large proportion of the cells in the DRN are GABAergic. These cells are much more excitable than serotonin neurons (Harkin et al., eLife, 2023), they send inhibitory input to serotonin neurons (Weissbourd et al., Cell, 2018), and they exhibit activity patterns sometimes opposite those of serotonin neurons (Li, Zhong, Wang et al., Nat Comm, 2016), so the BOLD signal may reflect metabolic activity due to local inhibition of serotonin neurons. Second, serotonin earned its name for being a vasoconstrictor, so it seems possible that local serotonin release in the DRN associated with increased serotonin neuron activity might paradoxically decrease the BOLD signal. If the authors have reason to believe that this mirror possibility is implausible, I would encourage them to add this to the discussion. Otherwise, this possibility should be mentioned early in the paper so that readers bear in mind that the sign of the effect might be flipped. I would like to emphasize that these results are still interesting even if there is a possibility of a sign flip. For example, given that two of the major theories of serotonergic function are related to valence and surprise (see

comments under References), it seems interesting that DRN activity may reflect a signed rather than unsigned change in environmental richness. In my view, this favours a valence over a surprise interpretation, even if it remains unclear whether the DRN signals positive changes in valence (as in Feng et al., 2024 and others) or negative changes (as the present results would suggest if taken at face value).

We agree with the reviewer that interpreting the directionality of the BOLD signal in the DRN is inherently challenging. A positive BOLD signal does not necessarily indicate increased neuronal firing, nor does a negative signal imply inhibition. This ambiguity is precisely why we have been cautious not to directly equate BOLD changes with serotonergic output. Throughout the manuscript, we avoid attributing the observed effects specifically to serotonergic activation or inhibition.

The reviewer's proposed mechanisms are entirely plausible and align well with our central conclusion: that DRN activity—regardless of its cellular source—tracks changes in behavioural policy. As the reviewer notes, this activity could reflect increased firing of serotonergic neurons, suppression via local GABAergic interneurons (which are highly excitable and numerous in the DRN), or even non-neuronal processes such as vasoconstrictive effects of serotonin itself.

We have now added a statement to the Discussion to explicitly acknowledge these possibilities and to emphasize that the observed signal may reflect complex and potentially opposing contributions from different cell types:

It is unclear whether this function arises from serotonin neurons specifically, as many DRN neurons are non-serotonergic and it is not possible to discriminate serotonergic from non-serotonergic populations in fMRI recordings (Fu et al., 2010; Jacobs & Azmitia, 1992). **Moreover, the BOLD signal may not only reflect activity from serotonergic neurons but also local GABAergic inhibition, making the direction of the effect difficult to interpret.** Future experiments in human participants could address this by examining whether pharmacological manipulations of the serotonin system influence patterns of behavioural policy change.

5. Cosine dissimilarity

I'm not sure what the authors mean when they say that cosine dissimilarity is "unaffected by changes in mean univariate BOLD signal". The difference being that the Pearson correlation involves centering each variable around its mean while the cosine similarity does not. This would make cosine (dis)similarity sensitive to the mean. Could the authors clarify how they are calculating cosine similarity (clearly define x and y and any pre-processing) and whether this concern is applicable in their case?

We thank the reviewer for raising this point. The difference between cosine similarity and Pearson correlation is not in mean centering in one case and not doing so in the other. Cosine similarity measures the angle between two vectors. In general, multiplying a vector by a scalar does not change the direction of the vector and hence angular difference by definition is independent of the magnitude of the vectors. We can show this point using an example and the formula that the reviewer wrote down in the review:

Let $A=[1,2]$, $B=[2,1]$, and $B'=3\cdot B=[6,3]$.

B' is in the same direction as B but scaled. Even here, it is obvious that the angular difference between B' and A remains the same as the angular distance between B and A .

However, we can also show it using the formula:

Cosine similarity between A and B

$A \cdot B = 1 \cdot 2 + 2 \cdot 1 = 4$ (this is the dot product between A and B).

$$\|A\| = \sqrt{1^2 + 2^2} = \sqrt{5}$$

$$\|B\| = \sqrt{2^2 + 1^2} = \sqrt{5}$$

$$\text{cosine similarity}(A, B) = \frac{4}{\sqrt{5} \cdot \sqrt{5}} = \frac{4}{5} = .8$$

Similarly, we can compute the cosine similarity between A and B':

$$A \cdot B' = 1 \cdot 6 + 2 \cdot 3 = 6 + 6 = 12$$

$$\|A\| = \sqrt{1^2 + 2^2} = \sqrt{5}$$

$$\|B'\| = \sqrt{6^2 + 3^2} = \sqrt{36 + 9} = \sqrt{45}$$

$$\text{cosine similarity}(A, B') = \frac{12}{\sqrt{5} \cdot \sqrt{45}} = \frac{12}{\sqrt{225}} = \frac{12}{15} = \frac{4}{5} = .8$$

We have now clarified this point in the Methods section of the manuscript:

For each ROI r , and each predictor i in β_{1-30} , GLM5.1 produced a vector of regression weights capturing the effect of β_i at each voxel v in r . Each vector of regression-weights constituted a multivariate neural representation of a reward-option. We then quantified pairwise distances between multivariate representations using cosine dissimilarity – i.e., **angular difference between two vectors**. We used cosine dissimilarity instead of standard parametric (e.g. Pearson's R) and non-parametric (e.g. Kendall's Tau) methods to avoid artefacts arising from univariate differences in activity (see Discussion for further details). **Because cosine dissimilarity measures the angle between two vectors, it remains intact if one vector is scaled, hence it is insensitive to the magnitude of the vectors.**

6. Cortical option representations

H1 and H2 do not seem mutually-exclusive to me — quite the opposite! The statistics for H1 also seem to be missing from the text.

We did not intend to suggest that H1 and H2 are mutually exclusive—on the contrary, we considered both hypotheses as potentially valid and tested whether one or both were supported by the data. Our RSA results indicated that H2—the representational distance between the 10-point and 50-point options—was significantly correlated with rational task performance in both dACC and AI (Fig. 4C). In contrast, H1—the representational distance between the 10-point and 5-point options—was not correlated with behaviour in dACC or AI ($t_{dACC}(21) = -1.12, p = .271$; $t_{AI}(21) = -0.45, p = .650$), nor in any other ROI.

These statistics for H1 were originally reported in Supplementary Figure S5, but in response to the reviewer's comment, we have now moved them to the main text:

We confirmed this interpretation by contrasting $d_4 - d_2$ in participants who were sensitive and insensitive to task environments, which indicated that the $d_4 - d_2$ difference was greater in the environment-sensitive group than in the insensitive group ($t_{dACC}(11) = 3.36, p = .006$; $t_{AI}(11) = 3.15, p = .012$; fig.4C–D). There was no evidence of these patterns in any other ROI. **In addition, H1 – the representational distance between 10-point and 5-point options – was not correlated with behaviour in dACC or AI ($t_{dACC}(21) = -1.12, p = .271$; $t_{AI}(21) = -0.45, p = .650$), nor in any other ROI.**

7. References

We are grateful to the reviewer for their thoughtful comments and for providing a comprehensive list of relevant citations. Unfortunately, due to word limits, we are unable to discuss all of the papers cited. However, we have now incorporated an expanded set of references and related discussion into the manuscript. Please see our responses to comments 1 and 2, and the modified Discussion of the manuscript.

8. Typos

We thank the reviewer for highlighting these typos. We have now corrected them.

Reviewer #3 (Remarks to the Author):

In this paper the authors use 7T fMRI and a novel behavioral task to show a role of Dorsal Raphe Nucleus (DRN) in signaling in policy changes, and effects of reward context on representations in dorsal Anterior Cingulate Cortex (dACC) and Anterior Insula (AI).

They further show that the reward context influences connectivity between DRN and dACC. I particularly liked this part since context and context switching is an important (low order) approach to understanding human cognitive representations.

This is an interesting paper that makes a very significant contribution to understand the role of DRN signaling in human decision making. 7T functional imaging lends extra weight to the findings. The paper is well-written and provides an appropriate amount of detail.

There are only a few minor points to clarify. Also, it is interesting that the analysis relies on variables that are not based on any particular model of the behavioral data. While this avoids debates about such models it would be interesting to see if, for example, a hidden Markov model could capture the switches in behavioral policy.

We are pleased that the reviewer found our paper interesting and recognised its significant contribution. The suggestion regarding the use of a hidden Markov model is indeed very insightful. We have previously applied similar models to decode latent motivational states in non-human primates (Priestley et al., Science Advances, 2025). However, in this study, we relate brain activity to a participant's behavioural policy, which is an observable phenomenon that is indicated in the decision to pursue or reject a reward opportunity on each trial. Because behavioural policies can be directly measured and do not need to be inferred, we believe that analysing behavioural data with an HMM that aims to identify latent states is not necessary to support the claims we make in the manuscript.

Minor Points

1. Line 233. "high field" – Why not just sat 7T? I don't think this is stated until the methods.

We have clarified that the field strength is 7T from the outset in the revised manuscript.

2. Line 260. "fig.3D-E" There is no figure 3E. Akso, it is not clear what the GLM (and beta) is used in (the submitted) figure 3C.

Thank you for pointing out the incorrect reference to figure 3E. This has now been deleted in the revised manuscript. The GLM used to create the results shown in figure 3C was GLM 4.2 and so the beta illustrated is the beta for the policy switch term in this GLM. The revised manuscript now makes this clear as follows:

We first examined brain activity related to policy-switches. We did this by contrasting brain activity on policy-switch trials with brain activity on policy-stay trials – i.e., we asked whether specific patterns of brain activity occurred when a policy-switch was exhibited in behaviour. This indicated that activity in DRN – but no other subcortical ROI – represented policy-switch trials (GLM4.1a; $t_{DRN; policy-change}(26)=2.84, p=.043$; fig.3B).

Figure 3. Brain activity in dorsal raphe nucleus represents environment-driven changes in behavioural policy. (A) The fMRI analysis focussed on anatomical regions of interest

(ROIs) in the ascending neuromodulatory systems (ANS). ROIs comprised the dorsal raphe nucleus (DRN), habenula (Hb), locus coeruleus (LC), substantia nigra and ventral tegmental area (MDB), and medial septal nucleus (MS). See Methods for details of mask construction. **(B)** Distribution of peak regression-weight for policy-switch (purple; β_1 from GLM4.1) and action-switch (grey; β_1 from GLM4.2) events in subcortical ROIs. Policy-switches – but not action-switches – modulate DRN activity, suggesting that DRN is involved in option-specific forms of behavioural change. No other ROIs showed representations of behavioural change. **(C)** Timecourse of the relationship between BOLD signal and: (i) policy-switch events (top; purple; β_1 from GLM4.1), and; (ii) action-switch events (bottom; grey; β_1 from GLM4.2) in DRN. Time=0 (x-axis) corresponds to onset of reward-opportunity stimulus. Given haemodynamic delay, it is likely that policy-switch related neural activity in DRN occurs in ITI periods, when subjects are integrating reward feedback from the previous trial and planning their future behaviour. **(D)** Peak regression-weights for control analyses linking DRN to environment-driven policy switches: **(i)** shows DRN activity during policy changes that are congruent to the environment – i.e., pursue-switch events in poor environments (dark-pink; β_1 from GLM4.2a) and reject-switch events in rich environments (light-pink; β_1 from GLM4.2c); **(ii)** shows DRN activity during pursue-switch events in poor environments (dark-pink; β_1 from GLM4.2a) and rich-environments (grey; β_1 from GLM4.2b) – i.e., events matched on motor output, but that differ in environment-congruence; **(iii)** shows DRN activity during reject-switch events in rich environments (light-pink; β_1 from GLM4.2c) and poor-environments (grey; β_1 from GLM4.2d) – i.e., events matched on motor output, but that differ in environment-congruence; **(iv)** shows DRN activity during pursue-switch events for the 10-point option (dark-pink) and 5-point option (grey), where only policy-switches for the 10-point were modulated by the environment in behaviour (fig.2C–D; supplementary fig.S2; β_1 from GLM4.2a applied to relevant trials); **(v)** shows DRN activity during reject-switch events for the 10-point option (light-pink; β_1 from GLM4.2c applied to relevant trials) and 5-point option (grey; β_1 from GLM4.2c applied to relevant trials). In timecourse graphs (C), lines and shadings show the mean and standard error (SE) of the β weights across participants. In effect-size graphs (B, D), bars show sample-mean magnitude of the peak regression weight in the timecourse according to an unbiased leave-one-out procedure (see methods). Dots indicate peak regression weights for individual participants.

3. Line 357. It would be clarifying to add here that the event is “opportunity onset”.

This has been added as suggested to the revised manuscript:

First, we searched for relevant ROIs in our own data using whole-brain fMRI analysis testing for regions that coded: (i) the pursue-vs-reject decision made on each trial, and; (ii) the reward-magnitude of the opportunity on each trial (GLM3.1 & GLM3.2 which were time-locked to the time of the opportunity onset; see supplementary fig.S4).

4. Line 424. There is no GLM 4.3 in the methods. It is not clear what is being shown in figure 5B.

We have clarified this by adding the following section to the methods:

Finally, we tested for psychophysiological interactions (PPI) between DRN and dACC/Al as a function of environment-type with the following GLM:

GLM4.3

$$BOLD_{ROI} = \beta_0 + \beta_1 BOLD_{seed} + \beta_2 PPI + \beta_3 environment\text{-}type + \beta_4 trial\text{-}number$$

Where $BOLD_{seed}$ is time-series data for the seed region in PPI analysis (here, DRN), PPI is the interaction between $BOLD_{seed}$ and *environment-type* regressors, and *environment-type* is the experimentally specific block-type (i.e., rich-vs-poor).

5. Line 1143. “normalized” – how?

We used the standard normalization process, sometimes referred to as “z-scoring”. We have, however, summarized this as transparently as possible as follows:

To prepare ROI data for analysis, the filtered time-series of BOLD signal from each voxel was averaged, normalised (so that the time course’s mean activity was centred at zero and its standard deviation was one) and up-sampled by a factor of 20 with spline interpolation (Behrens et al., 2008).

6. Line 1146. Should say the GLMs were performed.

Changed as suggested.

7. Line 1151. I think everything here was done within subject and then averaging beta values over the N-1 subjects then maximizing. I think this should be clarified. Also, I think “effect size for “could be confusing. I think beta value is what is meant since it is later stated “N x 1 vector of parameter estimates”.

The reviewer’s interpretation is correct. We have modified the paragraph to avoid confusion.

The effect of r in s was then estimated by calculating the parameter estimates for r at time t in subject s . This was repeated iteratively for each subject, leaving an $N \times 1$ vector of parameter estimates that were not biased by the timecourse of data of individual participants.

8. Line 1232. “cluster-corrected Beta-weight”. I think cluster corrected p-value (or T or z value) is meant.

Thank you for pointing out that this is not clear. The reviewer’s suggested meaning here is not quite correct and the sentence has been corrected as follows:

Each ROI was constructed as a sphere with 7mm radius centred on the voxel with the highest β -weight in the area of activity identified after cluster-correction.

9. Line 1276. “vector of v ...” v is an index, not the number of voxels.

This has been corrected as follows:

For each ROI r , and each predictor i in β_{1-30} , GLM5.1 produced a vector of regression weights capturing the effect of β_i at each voxel v in r .

Reviewer #4 (Remarks to the Author):

In the current study, Priestley and colleagues investigated the neural correlates of behavioral policy change in a foraging-like task. In the task, participants chose between pursuing vs. rejecting rewards in either reward-rich or -poor environments. Pursuing entailed opportunity costs of time, so that participants had to estimate whether pursuing lower-level rewards was worth the time cost of doing so. The authors report that activity in the dorsal raphe nucleus (DRN), but not in other nuclei in the ascending neuromodulatory systems tracks environment-congruent policy switches. Moreover, activity in the dorsal anterior cingulate cortex (dACC) and anterior insula (AI) seemed to represent offer value in relation to the average reward rate of the environment.

Overall, I think that this experiment was expertly conducted and that the results are convincing. The findings would be of broad appeal across disciplines interested in decision-making and fit the scope of the journal very well. Before I can fully endorse the manuscript for publication, however, I have a few comments and concerns, of which I am certain the authors can adequately address.

- 1. My most major concern relates to the conceptualization of policy switching in the behavioral task. The authors defined switches as changes in pursue vs. reject responses in consecutive trials of the same reward magnitude. On page 8, lines 323-337, the authors report that DRN activity reflected environment-congruent switches only, while medial septal nucleus (MS) activity tracked environment-incongruent switches, which they think of as explorative. However, especially early in a block, all switches, including those that are environment-congruent, might reflect exploratory behavior (and learning) instead of deviations from an environment-driven optimal policy (which is indicated by the increasing/decreasing acceptance rates of the 10-point option over time). Unless I misread, all trials were used in the analyses of DRN activity. But how could DRN activity truly reflect environment-driven policy changes at a time when participants could not know yet whether they were currently in a rich or a poor block? Or in other words, how could DRN discriminate whether a chosen action represents an environment-congruent or -incongruent switch when the nature of the environment is yet unknown? As far as I could tell, there was no analysis of how DRN activity changes over time, similar to the behavioral adaptation to the environment. For example, it would be interesting to see how long it takes DRN activity to adapt to a change in the environment (i.e., how many trials it takes until DRN activity reliably scales positively with pursuing vs. rejecting 10-point offers when changing from a poor to a rich environment).**

We thank the reviewer for this suggestion. It is important to first emphasise that, in general, congruent policy-switching events in both rich and poor environments occurred only after several trials had passed (in rich blocks, $M=7.81$, $SD=2.51$; in poor blocks, $M=10.9$, $SD=2.94$; fig S2.C). This suggests that these changes were indeed based on discriminating the richness of the environment.

We agree with the reviewer, however, that it is important to characterise how the DRN activity pattern representing these changes evolves over time. Although it is difficult to pinpoint the precise time at which the DRN activity pattern emerges using fMRI data, we are able to test for the presence of the activity pattern in a moving time window aligned to one, two, three, etc trials after a block onset. We set the width of the time window of 6 trials on the basis that this was approximately the SD of policy-switch time distributions in both rich blocks and poor blocks (see above).

When DRN activity was analysed in this way, we found that representations of policy-switching emerged only in late time windows: specifically, the $(t=5):(t=10)$ and $(t=6):(t=11)$ time windows in rich blocks, and the $(t=9:t=14)$ and $(t=10):(t=15)$ time windows in poor blocks. This is consistent with the hypothesis that neural representations of policy-switches in DRN were based on judgements about the richness of the environment. In the figure below, points indicate the mean effect-size of policy-switches in each time window, error bars indicate the 95%CI for the effect size after Bonferroni correction, and colours indicate policy-switches in poor and rich environments, respectively.

Interestingly, the timing of the emergence of these neural representations matches with the timing of congruent policy-switching events in both rich and poor environments. Congruent policy-switches in rich blocks occurred earlier than congruent policy-switches in poor blocks (please compare fig S2.C and S5.B).

Figure S5. Further details of policy-switch effect in DRN. (B) To characterise how the DRN activity pattern representing policy-switches evolved over time, we tested the effect in time windows aligned to specific trials after the beginning of a block. We set the width of the time window of 6 trials on the basis that this was approximately the SD of policy-switch time distributions in both rich blocks and poor blocks (see supplementary fig. S2C). When DRN activity was analysed in this way, we found that representations of policy-switching emerged only in late time windows: specifically, the $(t=5):(t=10)$ and $(t=6):(t=11)$ time windows in rich blocks, and the $(t=9:t=14)$ and $(t=10):(t=15)$ time windows in poor blocks. This is consistent with the hypothesis that neural representations of policy-switches in DRN were based on

judgements about the richness of the environment. In the figure, points indicate the mean effect-size of policy-switches in each time window, error bars indicate the 95%CI for the effect size after Bonferroni correction, and colours indicate policy-switches in poor and rich environments, respectively.

We have now added the figures and results from this new analysis to the manuscript:

Similarly, there was no evidence for an effect for reject-switches for the 5-point option in rich environments on DRN activity (GLM4.3c, $t_{DRN; low-option\ reject}(26)=-0.73$, $p=.471$; fig.3D-v) in contrast to the significant effect of the 10-point option reject-switches (GLM4.3c, $t_{DRN; middle-option\ reject}(26)=-2.08$, $p=.046$; fig.3D-v) although the difference between signals in this case was not significant ($t_{DRN; low-vs-middle\ reject}(26)= 1.77$, $p=.090$; fig.3D-v). **Furthermore, we found that representations of policy-switching emerged only in late time windows after a block onset (supplementary fig.S5B) in line with the time taken for participants to lean about the value of the environment (see fig.1F).**

- 2. A second potential problem with the interpretability and generalizability of the study is the relatively low sample size (n = 29, of which only 25 were included in the fMRI analyses). I understand the practical challenges of testing large samples in fMRI studies. However, I was wondering if there were any a priori considerations about statistical power or whether sensitivity analyses have been conducted to ensure the study was sufficiently powered?**

We would first like to apologise for a mistake in the Methods section of the manuscript. We collected data from 31 participants and excluded 2 because they failed to make a serious effort on the task and an additional 2 due to poor brain registration. Therefore, our final sample size is 27, and not 25. This was reported correctly in the Results section but incorrectly in the Methods, which has now been corrected.

We appreciate the reviewer's comment and agree that statistical power is an important consideration. Although our fMRI sample size is modest (n = 27), this is partially offset by our use of ultra-high field (7T) imaging, which offers significantly higher signal-to-noise ratio (SNR), temporal SNR, and spatial specificity compared to standard 3T systems, which are used in many cognitive and computational research studies. These improvements are especially beneficial when investigating small brainstem or subcortical structures.

Several studies have demonstrated that 7T fMRI can achieve greater statistical sensitivity and power than 3T, even with the smaller sample sizes. For example:

- *de Martino et al. (2018) showed that 7T provides enhanced contrast-to-noise and spatial precision, facilitating more accurate mapping of small subcortical areas.*
- *Torrisi et al. (2018) directly compared matched 7T and 3T datasets using a Go/NoGo task and revealed that 7T yielded stronger statistical effects, even with the same number of participants.*
- *Isherwood et al. (2025) performed comparative work to show that 7 T yields superior temporal SNR in subcortical regions relative to 3 T, enhancing sensitivity to BOLD signals in small nuclei*

Consistent with this, prior studies targeting similar regions have typically used comparable sample sizes and reported robust findings. These include studies from our laboratory and from others looking at the dopaminergic midbrain regions of the ventral tegmental area and substantia nigra (Khalighinejad et al., PNAS, 2020; Nature Communications, 2021; Hauser et al., PNAS, 2017; Rigoli et al., Nature Communications, 2016; Torris et al., NeuroImage, 2017), the dorsal raphe nucleus (Trier et al., PNAS, 2025; Torrisi et al., NeuroImage, 2017), medial septum (Khalighinejad et al., PNAS, 2020; Nature Communications, 2021; Torrisi et

al., *NeuroImage*, 2017) and locus coeruleus (Meissner et al., *Nature Human Behaviour*, 2024; Crawford et al., *J. Neuroscience*, 2021; Song et al., *J. Neuroscience*, 2017; LLoyd et al. *Cerebral Cortex*, 2025; Groot et al., *J. Neuroscience*, 2023).

Finally, although we did not conduct a formal a priori power analysis, the observed effect sizes support the adequacy of our sample. For example, our main fMRI contrast – congruent policy switches – yielded a paired-sample *t*-value of 2.39 with $n = 27$, corresponding to a Cohen's *d* of 0.47, which reflects a medium-sized effect. This suggests that, despite the modest sample size, the study was sufficiently powered to detect reliable effects in the regions of interest.

More minor points:

3. I would greatly appreciate if the authors could include some descriptive statistics about participants choice behavior in the text. Specifically, it would be great to see mean (+/-SEM) purse/reject rates for each environment as well as for the total number of policy switches especially in the 10-point condition. I know the former can be inferred from Figure 1E but it would still be interesting to have the exact numbers. Given that only certain trials (those with policy switches) were included in most analyses, it is important to know how many of these existed per participant, to make sure the analyses are adequately powered.

We have now added the requested information to the legend of Figure S1.

Mean purse/reject rates for poor and rich environments was 43.80%(+/-2.43%) and 69.60%(+/-2.15%), respectively. On average, there were 16.9 (+/-1.70) policy switches with respect to the 10-point option per participant, and 422 in total. We note, however, that our fMRI analysis of policy-switches is not confined to 422 trials per se, but rather is based on contrasting activity in the 422 trials on which policy-switches for the 10-point option occurred against other trials in the dataset.

4. I would encourage the authors to please report participants' sex or gender, even if no analysis were done to investigate differences based on these variables.

The revised manuscript reports the participants' self-reported sex as follows:

Thirty-one participants (11 self-reported men and 20 self-reported women) performed the experiment. All were between 18-40 years of age, reported normal or corrected-to-normal vision, and no current diagnosis or treatment for psychiatric or neurological disorder. Participants received £40 for completing the experiment and could earn an additional payment of up to £10 depending on their performance in the decision-making task. All relevant ethical regulations governing research with human participants were observed, and participants provided written informed consent at the beginning of each session. Ethical approval was given by the Oxford University Central University Research Ethics Committee (CUREC) (Ref-Number: MSD-IDREC-R55856/RE001).

5. Similarly, I would appreciate a more detailed explanation for the exclusion of 2 participants based on task performance. How exactly did they perform incorrectly?

The manuscript has been revised as follows:

Two participants were excluded from behavioural analyses because they did not perform the task correctly. In both cases participants clearly failed to follow instructions and attempted to respond to all stimuli on nearly every trial. An additional two participants were excluded from

fMRI analyses due to excessive head motion which prevented accurate registration (motion outliers > 15% of total fMRI volumes). **Self-reported sex of participants was not considered during analysis.**

6. Typos:

- Page 3, line 102: “Upon each, encounter,... (first comma should be deleted)
- Page 6, lines 221-222: “A similar pattern obtained...” (WAS obtained)
- Page 9, line 353: “...on a series of ROIs deriving from...” (I think this should be DERIVED)
- Page 13, lines 551-552: “...might similarly shed light serotonin’s role...” (shed light ON)

Thank you for pointing out these issues. We think that the second of these is actually correct usage and we have retained it as it stood. However, the other three points have all been revised as suggested.

REVIEWER COMMENTS

*We thank the editor and Reviewers 1, 2, 3, and 4 to have found our responses to their previous comments thorough and satisfactory. Below we have addressed the remaining comments from reviewer 2. The reviewer comments are shown in **bold**, our response in italics, and the manuscript text is shown in a standard font with new sections highlighted in red.*

Reviewer #2 (Remarks to the Author):

I would like to begin by thanking the authors for their detailed response to my initial review. The new version of the manuscript improves on an already interesting story. In particular, it is good to see more direct comparisons with other neuromodulatory nuclei (lines 264-270) and new discussion of connections to the most important related works (lines 489-550).

In my initial review, I tried to keep my feedback at a very high level in order to give the authors maximum flexibility in how to address my points, but perhaps this was not helpful. In this round, I have tried to be brief and specific. While I have made a number of suggestions, only the four marked ⊗ are important. The rest are optional.

We are glad that the reviewer found our responses satisfactory. Below, we address in detail the important suggestions (marked by ⊗) raised by the reviewer.

- 1. ⊗ Since the authors do not directly show that their policy switching theory can replace existing theories of DRN function, I believe the statement in the introduction that “we argue that [various DRN functions] involve a common process of setting a behavioural policy” (line 74) is unwarranted. Without changes to the results, this sentence should be removed. On the one hand, primary data and new ideas are essential ingredients for scientific progress, as the authors rightly point out, and in my opinion both the data and ideas in the present work are very nice. On the other hand, the lack of consensus in the serotonin field reflects an abundance of existing options, some of which are fairly vague. In this context, I recognize that it is challenging to find space to rule out existing ideas while also proposing a new one. Finding the right balance is a matter of taste and editorial judgement.**

Our intention with the statement “we argue that [various DRN functions] involve a common process of setting a behavioural policy” was not to suggest that our policy switching theory replaces existing theories of DRN function, but rather that it provides a unifying framework for interpreting these diverse perspectives. We agree, however, that the strength of our original wording may have implied otherwise. As the reviewer notes, this is ultimately a matter of balance and editorial judgement. To address this concern, we have softened the language in the introduction, line 75, as follows:

We propose that these seemingly disparate functions may be linked by a common process of setting a behavioural policy.

2. **⊗ Policy vs. policy switch:** In my view, the most intriguing aspect of the present model is that it relates transient changes in DRN activity to sustained changes in policy. This specific connection is an emerging theme in the work of this research group, but seems fairly novel otherwise. For example, while Marques et al. use a policy switch model conceptually similar to that of the present work, they implicate the DRN in sustaining exploitation and identify a separate “trigger network” responsible for policy switches per se. Other papers from this group make a convincing case for this “transient activity -> sustained behaviour” idea by showing that DRN activity correlates with policy switches but not policy (eg Priestley, Sci Adv, 2025 Fig. 5B), but I couldn’t find a similar control analysis here. I would strongly suggest to repeat the analysis shown in Fig. S6A with DRN rather than MBD and refer to it in the main text alongside a citation to Priestley et al. Sci Adv 2025. I believe that highlighting this analysis would strengthen the titular claim (“Activity in human DRN signals changes in behavioural policy”), further shore up this theme in the authors’ work, and clearly show why existing policy-oriented serotonergic theories are inadequate.

We thank the reviewer for this insightful suggestion. In Priestley et al., we demonstrated that DRN activity correlates with changes in motivational state, but not with the absolute level of motivation. Analogously, the relevant comparison here is between changes in behavioural policy versus the policy itself. As the reviewer notes, one possibility is to repeat the analysis in Fig.S6A. However, the analysis in Fig. S6A captures the motoric component of decision-making – whether to act or not to act – which explains why strong activity is observed in dopaminergic regions. To address the reviewer’s query, a more appropriate control analysis for our current study is to test whether transient changes in DRN activity are associated with ‘sustained changes’ in policy, rather than with rapid changes in action – which is not confounded by motor execution. This is precisely what we show in Fig. 3C, where DRN activity is significantly correlated with sustained changes in policy, but not with rapid changes in action.

To make this clearer, we have now added the following sentence to the Results section, line 280:

Next, we confirmed that this effect was specific to behavioural-policy changes by testing DRN’s relationship with other forms of behavioural change, like action-switches where both the pursue-vs-reject decision and option-identity changed across consecutive trials such that the change in behaviour did not constitute change in option-specific policy (fig.2A). There was no evidence for an effect of action switches on DRN activity (GLM4.1b; $t_{DRN; \text{action-switch}}(26) = -1.66, p = .489$; fig.3B–C), and the policy-switch representation reported earlier was, moreover, significantly different to the null action-switch signal ($t_{DRN; \text{action-switch vs policy-switch}}(26) = 3.53, p = .002$; fig.3B–C). **This suggests that the observed activity in DRN is related to sustained changes in policy, but not to rapid changes in action (see also fig.S5C&D).**

In addition, we have conducted a new analysis to probe whether changes in DRN activity that signal policy shifts are transient or sustained. For each policy-change, we examined activity in our ROIs during two types of subsequent event: (i) the trial immediately following a policy-shift (i.e., regardless of which option featured on that trial), and; (ii) the next encounter with the option that produced the policy-shift. There was no evidence of changes in DRN activity in either case ($t_{\text{trial-after}}(26) = -0.47, p = .999$; $t_{\text{encounter-after}}(26) = 1.99, p = .284$). This new analysis, along with our previous results,

strengthens our claim relating ‘transient’ changes in DRN activity with ‘sustained’ changes in behaviour.

We have added this new analysis to the supplementary figure S5.

Figure S5. Further details of policy-switch effect in DRN. (C&D) To probe whether changes in DRN activity that signal policy shifts are transient or sustained, for each policy-change, we examined activity in our ROIs during two types of subsequent event: **(C)** the trial immediately following a policy-shift (i.e., regardless of which option featured on that trial), and; **(D)** the next encounter with the option that produced the policy-shift. There was no evidence of changes in DRN activity in either case ($t_{\text{trial-after}(26)} = -0.47$, $p = .999$; $t_{\text{encounter-after}(26)} = 1.99$, $p = .284$). In C&D lines and shadings of the time course graph show mean \pm SEM of β weights across participants.

3. \otimes Cosine dissimilarity. The authors are correct that cosine similarity is invariant to multiplication by a scalar, but this is not the same as being invariant to changes in mean. Adding 1 to B in the example given by the authors increases its mean from 1.5 to 2.5 and changes the cosine similarity from 0.8 to 0.87. Conversely, using A =

[1 0 1] and $B = [10\ 0\ 1]$ and adding 1 to A decreases the cosine similarity from 0.77 to 0.73. After consulting a colleague with expertise in fMRI data analysis, my understanding is that the univariate changes in BOLD that the authors intend to control for (lines 595-603) are unlikely to be purely multiplicative in nature. As a result, I am concerned that cosine similarity will be affected by univariate changes in BOLD. As shown above, the direction of the effect is difficult to know in advance. A better alternative would be to use Pearson correlation, since it is unaffected by both changes in mean and multiplicative scaling.

We apologise if we misunderstood the reviewer's previous comment. As the reviewer rightly points out, cosine similarity is invariant to multiplicative scaling but not to additive shifts, and it is correct that univariate changes in BOLD are unlikely to be purely multiplicative.

Therefore, as suggested by the reviewer, we have now repeated the RSA by using Pearson's correlation rather than cosine similarity and showed that it does not affect our main conclusions. We have added this new result and the figures to the supplementary figure S9.

Figure S9. Representational similarity analysis implementation and further results. (E) Cosine dissimilarity is unaffected by multiplicative scaling, while Pearson's correlation is unaffected by both multiplicative scaling and additive shifts. We therefore replicated the RSA results using Pearson's correlation ($r_{ACC}=0.65$, $t(21)= 3.91$, $p<.001$; $r_{AI}=0.53$, $t(21)=2.90$, $p=.008$) rather than cosine dissimilarity, suggesting that the correlation holds regardless of the similarity metric used.

We added the following line to the Results section:

We replicated these findings using Pearson's correlation rather than cosine dissimilarity, suggesting that the correlation holds regardless of the similarity metric used (fig.S9).

In addition, we have modified the lines 1483-1488 in the Methods section:

We then quantified pairwise distances between multivariate representations using cosine dissimilarity and Pearson's correlation to avoid artefacts arising from univariate

differences in activity (see Discussion for further details). Cosine dissimilarity is unaffected by multiplicative scaling, while Pearson's correlation is unaffected by both multiplicative scaling and additive shifts. As a result, both measures are insensitive to absolute differences in the magnitude of the vectors.

and lines 600-602 in the Discussion:

It is important to highlight, therefore, that we measured the distance between multivariate patterns of reward-evoked activity with cosine dissimilarity and **Pearson's correlation** – methods that obviate the influence of univariate changes on representational distance.

4. ⊗ **References.** I appreciate that the authors have made an effort to incorporate additional references and discussion, especially given limited space. At the same time, I do think it is an issue that the authors state in the introduction that “little is known about what DRN activity represents, and how DRN function differs from other neuromodulatory nuclei in the brainstem, midbrain, and basal forebrain” without citing or discussing any papers that directly compare DRN and MBD activity: Cohen et al., eLife, 2015; Zhong et al., J Neurosci, 2017; Cardozo-Pinto, Nature, 2024; Batten et al., Nat Hum Behav, 2024(Batten et al., 2024; Cardozo Pinto et al., 2025; Cohen et al., 2015; Zhong et al., 2017); the work of Matias et al. 2017 is cited, but only in the discussion on a separate point.

We thank the reviewer for highlighting this point. Our intention was not to overlook previous comparative work, but rather to emphasise that the functional significance of DRN activity, and its distinction from other neuromodulatory nuclei, is still not fully resolved. To clarify this, we have revised the introduction to read:

Although several studies have compared DRN activity with that of other neuromodulatory nuclei (Batten et al., 2024; Cardozo Pinto et al., 2025; Cohen et al., 2015; Zhong et al., 2017), what DRN activity specifically represents, and how its function differs from other neuromodulatory nuclei in the brainstem, midbrain, and basal forebrain, remains incompletely understood.

In addition, we have made sure that the studies cited by the reviewer are also cited in our manuscript.

Serotonin is thought to play an important but poorly-defined role in valence-related aspects of behaviour and learning. In this work, Priestley et al. investigate the relationship between the activity of the DRN (the main source of forebrain serotonin) and valence-related changes in human behaviour using high-resolution fMRI and a task inspired by the diet selection problem of foraging theory. Compared with the related literature, this study makes three main contributions: First, the authors directly implicate the DRN in long-term (multi-trial) valence-related changes in behaviour. Long-term aspects of reward-seeking/foraging-like behaviour are difficult to study and have received relatively little attention in the literature, making this result quite significant on its own. Second, the ability to record the output of multiple neuromodulatory nuclei at a time raises exciting possibilities to clearly differentiate their functions, and the authors show that the signal driving the long-term behaviour changes that they study is specific to the DRN. Third, this study tackles a diet selection problem with an interesting inter-temporal choice component that is overall quite different from the waiting, patch residence, bandit, and conditioning tasks in the existing literature, possibly highlighting previously unstudied aspects of DRN function.

On a technical level, the analyses are generally sound, thorough, and support the main conclusions. The methods are generally quite clear, the data are nicely formatted, and the code looks clean (although I didn't run it myself). It is great to see that the authors have put genuine effort into increasing the reproducibility of their work.

For now, I am left feeling that the potential of this manuscript was not fully realized. Although the main text tells a clear story in which policy switches are driven by changes in the environment, a sceptical reader who only skims the figures might get a different impression. Discussion of non-serotonergic neuromodulators is very limited apart from confirmatory results about dopamine. The novel task is not used to rule out any of the many existing theories of serotonergic function. By introducing an intriguing policy switch-based model of DRN function that does not seem to build on previous models, the authors have set themselves up for the daunting task of making a convincing case for a new interpretation of serotonergic function in an already crowded field. In my opinion, harmonizing the figures and main text, placing the new model in greater context, and clarifying the implications of these data for existing theories would significantly strengthen the paper and help move the serotonin field towards consensus.

Major

1. Implications for existing theories of serotonergic function

The authors are correct to say that there is “little consensus” about the behavioural function of the DRN, and it would be fair to say that there is similarly little consensus about what DRN activity represents. These data seem potentially quite interesting in terms of their implications for existing theories of serotonergic function (see below), and it would be nice to see a clearer and more direct presentation of any specific results that do or don't support these theories. It is up to the authors to decide how to present their story, but my suggestion would be to start by preparing a table listing their main results and whether or not they are consistent with each of the dominant theories. Once this is done, the authors could consider incorporating key points from this table into the results and/or discussion, potentially supported by new analysis. I believe that emphasizing the contribution of the present work to the evidence base for and against existing theories would increase the impact of the paper by helping to move the field towards consensus.

A few theories that I think would be particularly important to consider:

- Behavioural inhibition (Soubrié, *Behav Brain Sci*, 1986): Since the behavioural task involves withholding an action in order to proceed to the next opportunity, the authors could comment on whether their data support a general role for the DRN in behavioural inhibition. My initial interpretation is that these results are probably not consistent with behavioural inhibition because DRN activity does not correlate with accept/reject decisions on a trial-by-trial basis.
- Discounting (Doya, *Neural Networks*, 2002): Since the task has an intertemporal choice component, it would be interesting to consider whether a change in discounting initiated by the DRN could explain some of their results. The authors already touch on some relevant work from the Doya lab, and it would be great to expand on this. In my view, a persistent decrease in discounting in poor environments seems like a plausible way to explain the present results.

- Surprise/prediction error (Ligneul and Mainen, Current Biol, 2023): Serotonin has been proposed to signal surprising/salient events in general in order to facilitate behavioural adaptation to changing environments. Since some offers in this task are rarer than others, the authors could comment on whether DRN activity is correlated with surprising offers.
- “Beneficialness” (Luo et al., Neurobiol Learn Memory, 2016): As the authors note, serotonin has been repeatedly implicated in signalling positive valence (along the lines of value or reward). Since policy switches should reflect a change in the estimated long-term value of the environment, long-term value of accepting a particular option, or both, it would be good to consider whether the observed results might be consistent with one of these. In light of the fact that short-term and long-term value can look quite different, and that a large part of serotonin neuron activity seems to be related to changes in value rather than value *per se* (Harkin et al., Nature, 2025), the focus of the present study on changes in multi-trial aspects of behaviour seems quite interesting. On the other hand, the authors could consider whether the present results are more consistent with the proposal of Marques et al. (Nature, 2020) that serotonin controls value-*independent* policy switches.

2. Need for a new theory

This paper proposes that the DRN initiates changes in behaviour that are specifically related to long-term changes in reward availability. While this perspective touches on some common themes in the serotonin literature — that serotonin has something to do with changes in behaviour, reward availability, foraging, or adapting to adverse circumstances — this precise idea seems new to me. In my view, this novelty is a double-edged sword: On the one hand, this work provides a new way of thinking about DRN function with the potential to inspire new theories and experiments. On the other hand, since this paper does not seem to rule out any of the existing theories of DRN function, it risks becoming one more serotonergic idea among many. I believe the paper would be strengthened if the authors could clarify whether the policy switching model should be interpreted as an extension of some existing model, or, if not, exactly why a new perspective is needed.

I would find it particularly helpful to understand how the policy switching model relates to the following:

- Marques et al. (Nature, 2020) also propose that the DRN controls behavioural state transitions during foraging behaviour in zebrafish. In their case, DRN serotonin neurons promote a switch from an exploration-like global search for prey to an exploitation-like local search. Interestingly, this effect does not seem to be directly related to environmental richness, since the DRN periodically initiates local search-like behaviour even in a prey-free environment.
- Lottem et al. (Nat Comm, 2018) also study serotonergic function from a foraging perspective. However, whereas the present work focuses on a diet selection problem and shows that the DRN controls changes in “diet” (increase in activity => start accepting medium option, decrease in activity => start rejecting medium option), Lottem et al. focus on a patch foraging problem and show that activation of DRN serotonin neurons increases patch residence time.

3. Presentation of the policy switching behavioural effect in text vs. figures

The premise of the paper is that humans make decisions about whether to accept or reject reward offers based on durable inferences about whether these offers are worth their time. These inferences are reflected in patterns of behaviour (policies), with policy transitions being influenced by the DRN. The strength of this story hinges on whether a (potentially sceptical) reader could reasonably be convinced that the changes in behaviour in this task reflect such durable inferences. In my view, the presentation of the results in the main text is clear and compelling, but the key figures (Fig. 1E, 1G, 2B, and 2D) are less convincing. I’m unsure whether this means that the text should be reconciled to the richness of the figures by toning down some of the language, or whether the figures should be reconciled to the text by visually drawing out key results, so I’ve highlighted the areas where I felt there was a mismatch and will leave it to the authors to judge how best to adjust.

Figure	Impression from text	Impression from figure	Suggestion
1E	There are clear and pronounced changes in policy as a function of environmental context. L25, L79, L228	There is a statistically significant effect of environmental context on policy switching. However, on a scale from random switching to ideal context-dependent switching, participants are much closer to random.	It is clearly unrealistic to expect ideal behaviour from participants. Adding a more reasonable point of comparison to the figure would help put sub-optimal behaviour in context.
1G	Participants that more closely approximated ideal behaviour performed meaningfully better. L157	There is no statistically significant/meaningful relationship between x and y axes.	I'm not sure what the y axis represents. Fraction of maximum attainable score? More clearly labeled axes and a correlation test/effect size would help.
2B	Behaviour follows rules that are basically stable over time, and changes in behaviour are mainly related to changes in the underlying environment.	It is difficult to tell how the amount of time between policy switches compares with the block length in this experiment. Fig. 2B makes me wonder whether the time between policy switches is around five trials on average, which seems significantly shorter than the block duration? In that case, most policy switching would seem to be unrelated to changes in the environment.	Show the distribution of policy durations as a fraction of block length and/or quantify the fraction of policy switches that are related to changes in the environment.

Minor

Relationship between BOLD signal and serotonergic output

As the authors note in the discussion, the fMRI BOLD signal is not cell-type specific, so it is difficult to know how this signal relates to serotonergic output. I am certainly no fMRI expert, but it seems plausible to me that the BOLD signal might even be the mirror image of serotonergic output (in which case the data are still valuable!) for two reasons. First, a large proportion of the cells in the DRN are GABAergic. These cells are much more excitable than serotonin neurons (Harkin et al., eLife, 2023), they send inhibitory input to serotonin neurons (Weissbourd et al., Cell, 2018), and they exhibit activity patterns sometimes opposite those of serotonin neurons (Li, Zhong, Wang et al., Nat Comm, 2016), so the BOLD signal may reflect metabolic activity due to local *inhibition* of serotonin neurons. Second, serotonin earned its name for being a vasoconstrictor, so it seems possible that local serotonin release in the DRN associated with increased serotonin neuron activity might paradoxically decrease the BOLD signal. If the authors have reason to believe that this mirror possibility is implausible, I would encourage them to add this to the discussion. Otherwise, this possibility should be mentioned early in the paper so that readers bear in mind that the sign of the effect might be flipped.

I would like to emphasize that these results are still interesting even if there is a possibility of a sign flip. For example, given that two of the major theories of serotonergic function are related to valence and surprise (see comments under References), it seems interesting that DRN activity may reflect a signed rather than unsigned change in environmental richness. In my view, this favours a valence over a surprise interpretation, even if it remains unclear whether the DRN signals positive changes in valence (as in Feng et al., 2024 and others) or negative changes (as the present results would suggest if taken at face value).

Cosine dissimilarity

I'm not sure what the authors mean when they say that cosine dissimilarity is “unaffected by changes in mean univariate BOLD signal”. Cosine similarity is

$$\cos \theta = \frac{\sum_i x_i y_i}{(\sum_i x_i^2)^{\frac{1}{2}} (\sum_i y_i^2)^{\frac{1}{2}}}$$

while the Pearson correlation is

$$r = \frac{\sum_i (x_i - \bar{x})(y_i - \bar{y})}{(\sum_i (x_i - \bar{x})^2)^{\frac{1}{2}} (\sum_i (y_i - \bar{y})^2)^{\frac{1}{2}}}$$

the difference being that the Pearson correlation involves centering each variable around its mean while the cosine similarity does not. This would make cosine (dis)similarity sensitive to the mean. Could the authors clarify how they are calculating cosine similarity (clearly define x and y and any pre-processing) and whether this concern is applicable in their case?

Cortical option representations

H1 and H2 do not seem mutually-exclusive to me — quite the opposite! The statistics for H1 also seem to be missing from the text.

References

- L42: The cited Charnov paper introduces the marginal value theorem, which pertains to patch residence time rather than prey selection.
- L45-L54: No citations for this paragraph? It might be helpful to at least point the reader to a good book about foraging theory.
- L61-63: I agree that there is not yet a clear consensus on what DRN activity represents, and even less clarity about how DRN activity relates to activity in other neuromodulatory systems. This is not for lack of effort however, and it is a small travesty that none of the many excellent experimental studies of DRN activity in awake animals are cited here. Cohen et al. (eLife, 2015), Matias et al. (eLife, 2017), Li et al. (Nat Comm, 2016), Zhong et al. (J Neurosci, 2017), and Cardozo Pinto (Nature, 2024), all of which emphasize comparisons between serotonin and dopamine, seem particularly relevant.
- L69-70: A separate line of results from the Mainen lab emphasizes a role for serotonin in signaling something related to surprise rather than opportunity value. This idea has been influential (Feng et al., 2024; Grossman et al., 2022; and Paquelet et al., 2022 seem to partly echo this line of thinking) and probably deserves a mention. Harkin et al. (Nature, 2025) argue that the Mainen lab results can be interpreted in terms of value, however.
- L71: Do the authors see the reward learning-related effects of serotonin as being in the same vein? See, for example, Matias et al., eLife, 2017; Iigaya et al., Nat Comm, 2018.
- L474: Missing a reference to Marques et al. (Nature, 2020). Lottem et al. (2018) is not reversal learning so doesn't quite fit. Grossman et al. (2022) and Matias et al. (2017) do study reversal learning (although Grossman's reversals are quite fast) but I don't see a clear connection to exploration.
- L485: Modern literature on serotonin and escape from threat: Warden et al., Nature, 2012; Amo et al., Neuron, 2014; Andalman et al., Cell, 2019; Seo et al., Science, 2019
- L486: Reward/value-related activity has been observed repeatedly in serotonin neurons, as the authors note in their introduction. In my view, the idea that the *main* function of DRN serotonin neurons is to signal adversity can be comfortably ruled out. For primary data, see Bromberg-Martin et al., J Neurosci, 2010; Miyazaki et al., J Neurosci, 2011; Cohen et al., eLife, 2015; Li, Zhong, Wang, et al., Nat Comm, 2016; Zhong, Li, et al., J Neurosci, 2017; Ren et al., Cell, 2018; Paquelet et al., Neuron, 2022; Spring and Nautiyal, J Neurosci, 2024; Batten et al., Nat Hum Behav, 2024. For summary, see Liu, Lin, and Luo, Ann Rev Neurosci, 2020; Harkin et al., Nature, 2025.
- L497: Mkrtchian et al. (biorxiv, 2025) recently published a systematic review on the effect of serotonin on reward learning in humans. There may be clues in there. (Title: “Differential effects of dopamine and serotonin on reward and punishment processes in humans: A systematic review and meta-analysis”.)

Typos

- L70: “and control” should be “and controls”
- L547: “MDB” should be “MBD”
- L551: “shed light serotonin’s role” missing “on”
- There are many more hyphens than I would use. I suggest double checking whether all of these are necessary.

I would like to begin by thanking the authors for their detailed response to my initial review.

The new version of the manuscript improves on an already interesting story. In particular, it is good to see more direct comparisons with other neuromodulatory nuclei (lines 264-270) and new discussion of connections to the most important related works (lines 489-550).

In my initial review, I tried to keep my feedback at a very high level in order to give the authors maximum flexibility in how to address my points, but perhaps this was not helpful. In this round, I have tried to be brief and specific. While I have made a number of suggestions, only the four marked \otimes are important. The rest are optional.

Major points from round 1

1. Implications for existing theories of serotonergic function

The authors are correct to say that there is “little consensus” about the behavioural function of the DRN, and it would be fair to say that there is similarly little consensus about what DRN activity represents. These data seem potentially quite interesting in terms of their implications for existing theories of serotonergic function (see below), and it would be nice to see a clearer and more direct presentation of any specific results that do or don’t support these theories. [...]

A few theories I think would be particularly important to consider:

- Behavioural inhibition [...]
- Discounting [...]
- Surprise/prediction error [...]
- “Beneficialness” [...]

We are very grateful for the reviewer’s extremely thoughtful comments. Although we realise that many of the reviewer’s comments are positive, we are also aware that the reviewer suggests that we might have focussed on testing existing accounts of serotonin function. As explained above, already, we feel that our results are actually consistent with a newly emerging picture of DRN function, admittedly partly in papers published since the original submission of our paper. Moreover, we note that our investigations are based on many years of conducting often difficult experiments on related issues in primates, both human and non-human, in our laboratory (Wittmann et al., *Nature Communicants*, 2020; Khalighinejad et al., *Current Biology*, 2022; Trier et al., *PNAS*, 2025; Priestley et al., *Science Advances*, 2025).

To turn to the more general and philosophical question of how a programme of scientific research should proceed, we appreciate that there are different views about how one should conduct experiments just as there are theories about how one should make choices between different types of options in other areas of human endeavour. One view, however, is that an important element of making effective progress when two or more options are offered is to avoid simply choosing between them but instead to create a new path to consider alongside the existing options. The construction of such a new path, that draws on the most appealing aspects of the options offered, is often the way in which progress is made in science and in other domains (Iyengar, 2009, *The Art of Choosing*). Just as this approach can be taken in everyday life, so it can be taken in science and this is what we have tried to do in our work.

We contend that we have tried to draw on our own previous work and on the work of the other authorities that the reviewer cites, for example the Miyazakis and Doya, Monosov, Cools, Mainen as well as many others, and we note that we have cited their work throughout our manuscript. We have not previously cited the very interesting paper of Harkin and colleagues (*Nature*, 2025) simply because it was not published at the time that we submitted our article. Nevertheless, we draw on our own earlier work which has spent a long time under review making a closely related point that the DRN encodes transitions in value. We note that the Harkin paper does a particularly good job of reviewing previous work in the field, perhaps because it critically relies on accessing the data of other laboratories and so we take care to cite this article in our revised manuscript not just as a source of an interesting new proposal but as a useful review of the state of the field.

Our understanding is that it is not possible for us to attempt an extensive review of the literature with tabulations of the sort suggested by the reviewer in the current manuscript and the 5,000-word limit of *Nature Communications* articles. Such an approach is indeed often valuable but it is perhaps more appropriate for a review article than a report such as the present one and we note that our understanding is that the Discussion section in our current manuscript cannot be even as long as the reviewer’s review. However, while we are not able to fully implement all the reviewer’s suggestions we acknowledge that the spirit of them is correct and we have tried to extend our Discussion and to include additional consideration of some of the previously published studies that the reviewer mentions. In the revised manuscript we have included the following sections: [...]

I would like to thank the authors for their thoughtful response. While I had hoped that the authors might be able to take a firm position on one or more existing ideas directly in the results, the discussion paragraphs added by the authors cover the most important related work. The nuanced treatment of Ligneul and Mainen’s prediction error hypothesis and how this relates to the relative surprisingness of moderate value options in different environments is particularly good, and I would encourage the authors to prioritize this section when editing the final version of the paper to comply with word limits.

⊗ **Concrete point:** Since the authors do not directly show that their policy switching theory can replace existing theories of DRN function, I believe the statement in the introduction that “we argue that [various DRN functions] involve a common process of setting a behavioural policy” (line 74) is unwarranted. Without changes to the results, this sentence should be removed.

Philosophical point: On the one hand, primary data and new ideas are essential ingredients for scientific

progress, as the authors rightly point out, and in my opinion both the data and ideas in the present work are very nice. On the other hand, the lack of consensus in the serotonin field reflects an abundance of existing options, some of which are fairly vague. In this context, I recognize that it is challenging to find space to rule out existing ideas while also proposing a new one. Finding the right balance is a matter of taste and editorial judgement.

2. Need for a new theory

This paper proposes that the DRN initiates changes in behaviour that are specifically related to long-term changes in reward availability. While this perspective touches on some common themes in the serotonin literature — that serotonin has something to do with changes in behaviour, reward availability, foraging, or adapting to adverse circumstances — this precise idea seems new to me. In my view, this novelty is a double-edged sword: On the one hand, this work provides a new way of thinking about DRN function with the potential to inspire new theories and experiments. On the other hand, since this paper does not seem to rule out any of the existing theories of DRN function, it risks becoming one more serotonergic idea among many. I believe the paper would be strengthened if the authors could clarify whether the policy switching model should be interpreted as an extension of some existing model, or, if not, exactly why a new perspective is needed. I would find it particularly helpful to understand how the policy switching model relates to the following:

- [Work by Marques et al. (2020) on behavioural state transitions in foraging zebrafish.]

In the revised manuscript we have tried to address the links and differences between our current results and the previously published work of Marques and colleagues as shown below. We have also included references to the study by Harkin and colleagues (2025) and the study by Ahmadlou and colleagues (Nature, 2025) both of which were published some time after our manuscript was submitted.

[...]

The added discussion is a nice improvement that largely addresses my initial concern.

I have two concrete points for further improvement:

1. **⊗ Policy vs. policy switch:** In my view, the most intriguing aspect of the present model is that it relates *transient* changes in DRN activity to *sustained* changes in policy. This specific connection is an emerging theme in the work of this research group, but seems fairly novel otherwise. For example, while Marques et al. use a policy switch model conceptually similar to that of the present work, they implicate the DRN in sustaining exploitation and identify a separate “trigger network” responsible for policy *switches* per se. Other papers from this group make a convincing case for this “transient activity -> sustained behaviour” idea by showing that DRN activity correlates with policy switches *but not policy* (eg Priestley, Sci Adv, 2025 Fig. 5B), but I couldn’t find a similar control analysis here. I would strongly suggest to **repeat the analysis shown in Fig. S6A with DRN rather than MBD** and refer to it in the main text alongside a citation to Priestley et al. Sci Adv 2025. I believe that highlighting this analysis would strengthen the titular claim (“Activity in human DRN signals *changes* in behavioural policy”), further shore up this theme in the authors’ work, and clearly show why existing policy-oriented serotonergic theories are inadequate.
2. **Suggest strengthening wording:** I don’t think it is necessary for the authors to downplay their experimental paradigm as “simple and specific” in the added paragraph (it could equally be described as “well controlled”).

- [Patch leaving experiments of Lottem et al. (2018).]

In response to the reviewer’s comments we have added the following section:

[...]

Ok.

I do think that “DRN and adjacent raphe nuclei control changes along fundamental dimensions of behavioural policy” is fairly general. I would suggest something along the lines of “DRN and adjacent raphe nuclei modulate the drive to exploit proximal sources of reward”.

Typo: “forging elsewhere” -> “foraging elsewhere”.

3. Presentation of policy switching behavioural effect in text vs. figures

The premise of the paper is that humans make decisions about whether to accept or reject reward offers based on durable inferences about whether these offers are worth their time. These inferences are reflected in patterns of behaviour (policies), with policy transitions being influenced by the DRN. The strength of this story hinges on whether a (potentially sceptical) reader could reasonably be convinced that the changes in behaviour in this task reflect such durable inferences. In my view, the presentation of the results in the main text is clear and compelling, but the key figures (Fig. 1E, 1G, 2B, and 2D) are less convincing. I’m unsure whether this means that the text should be reconciled to the richness of the figures by toning down some of the language, or whether the figures should be reconciled to the text by visually drawing out key results, so I’ve highlighted the areas where I felt there was a mismatch and will leave it to the authors to judge how best to adjust.

- (1) As the reviewer rightly points out, it is unrealistic to expect participants to behave optimally. In Figure 1E, our aim was to examine whether the type of environment (rich vs. poor) influenced pursue-vs-reject decisions. The results showed that participants were more likely to pursue the 10-point option—but not the 5- or 50-point options—in poor compared to rich environments. This pattern is consistent with predictions from an optimal policy, which would suggest adapting behaviour based on environmental richness.

Moreover, in fig.1F we show that the subtle difference in pursue-rates illustrated in fig.1E becomes more pronounced as participants accumulate experience with their environmental context. This suggests that participants do, indeed, ultimately converge on an adaptive policy, but that they need time to disambiguate which environment they are in. In line with the reviewer’s suggestions, we have clarified our interpretation of the data in the main text and fig.1F captions as follows:

[...]

I find the authors’ interpretation convincing, although this does not resolve the disconnect between the visual impression given by figures and the interpretation in the text.

Concrete suggestion: Since block transitions were not cued, it seems reasonable to expect first and foremost that participant behaviour would adjust to block transitions with a delay, and only secondarily that block-averaged behaviour would reflect block type. I believe this could be shown more clearly if the authors change Fig. 1 to visually emphasize the time course of behaviour. The simplest way to accomplish this would be to enlarge panel F and present it before E. An alternative approach would be to adapt panel C by adding a behavioural element to show that participant behaviour towards the ten point option follows the red line with a delay.

- (2) There is indeed a statistically significant relationship between the total reward earned during the experiment and the degree to which participants change their behaviour toward the 10-point option between rich and poor environments. [We have added a new GLM and adapted the figure.]

This is clear, thanks. Indicating directly on panel G that the result is significant when analyzed with a GLM is helpful.

- (3) Importantly, behavioural policies are operationalised with respect to individual reward options. Only one reward option is presented on each trial. The 10-point reward option occurred with a constant probability of $p(10\text{-pt})=0.33$ across both rich and poor environments. The length of an average block was 17.0 trials ($SD=1.96$), implying that participants should encounter the 10-pt option 5.66 times on average, which was indeed the case ($M=5.89$, $SD=1.76$). As a result, the reviewer’s logic is consistent with the notion that participants adopted durable policies that lasted approximately the length of a block. In further support of this, the average number of policy-switches per block was 1.56 ($SD=1.65$).

It is difficult to pinpoint which policy-switches were undertaken specifically due to changes in the environment. This is because policy-switch decisions would have depended on a participant’s latent belief about the richness of the present context, rather than experimentally programmed changes in block-type. One way of probing the question is to quantify **when,** within a block, policy-switches tend to occur. In the figure below, we show the probability density function for policy-switches as a function of time-within-block. This suggests that policy-switches tended to occur at the midpoint of blocks, consistent with our earlier argument that behavioural adaptations to the environment tend to occur after participants have accumulated sufficient evidence that the environment has changed.

Taken together, these analyses suggest that participants tended to change their policy in relation to the 10-point option on 1-2 occasions per block, and that most changes occurred near the midpoint of blocks at which point it would be clear that the richness of the environment had changed. We believe that these observations are consistent with the claim that participants adopted durable behavioural policies that were determined by the perceived richness of the environment.

The authors’ response is clear and it addresses my concern.

As a suggestion, I do think that directly showing the time course of behaviour towards the ten point option using traces along the lines of Figs. 1 and 2 of Grossman et al. (2022) would tell this story more clearly (see above).

Minor points from round 1

4. Relationship between BOLD signal and serotonergic output

The authors have addressed my concern with the sentence added to the discussion.

5. \otimes Cosine dissimilarity

Unfortunately I believe my initial concern did not come across properly, and this point may turn out to be important for the RSA results.

I’m not sure what the authors mean when they say that cosine dissimilarity is “unaffected by changes in mean univariate BOLD signal”. The difference being that the Pearson correlation involves centering each variable around its mean while the cosine similarity does not. This would make cosine (dis)similarity sensitive to the mean.

We thank the reviewer for raising this point. The difference between cosine similarity and Pearson correlation is not in mean centering in one case and not doing so in the other.

This is not correct. In the Pearson correlation, mean centering is accomplished by subtracting \bar{x} from each x_i and \bar{y} from each y_i . Removing this part of the Pearson correlation formula yields exactly the formula for cosine similarity (see my initial review).

Cosine similarity measures the angle between two vectors. In general, multiplying a vector by a scalar does not change the direction of the vector and hence angular difference by definition is independent of the magnitude of the vectors. We can show this point using an example and the formula that the reviewer wrote down in the review:

[...]

The authors are correct that cosine similarity is invariant to multiplication by a scalar, but this is not the same as being invariant to changes in mean. Adding 1 to B in the example given by the authors increases its mean from 1.5 to 2.5 and changes the cosine similarity from 0.8 to 0.87. Conversely, using $A = [1 \ 0 \ 1]$ and $B = [10 \ 0 \ 1]$ and adding 1 to A decreases the cosine similarity from 0.77 to 0.73.

After consulting a colleague with expertise in fMRI data analysis, my understanding is that the univariate changes in BOLD that the authors intend to control for (lines 595-603) are unlikely to be purely multiplicative in nature. As a result, I am concerned that cosine similarity will be affected by univariate changes in BOLD. As shown above, the direction of the effect is difficult to know in advance.

A better alternative would be to use Pearson correlation, since it is unaffected by both changes in mean and multiplicative scaling. Letting a be a multiplicative factor and b be a change in mean, this can be shown as follows:

$$\begin{aligned} r &= \frac{\sum_i^N ((ax_i + b) - (a\bar{x} + b))(y_i - \bar{y})}{(\sum_i^N ((ax_i + b) - (a\bar{x} + b))^2)^{\frac{1}{2}} (\sum_i^N (y_i - \bar{y})^2)^{\frac{1}{2}}} \\ &= \frac{a \sum_i^N (x_i - \bar{x})(y_i - \bar{y})}{(a^2)^{\frac{1}{2}} (\sum_i^N (x_i - \bar{x})^2)^{\frac{1}{2}} (\sum_i^N (y_i - \bar{y})^2)^{\frac{1}{2}}} \\ &= \frac{\sum_i^N (x_i - \bar{x})(y_i - \bar{y})}{(\sum_i^N (x_i - \bar{x})^2)^{\frac{1}{2}} (\sum_i^N (y_i - \bar{y})^2)^{\frac{1}{2}}} \end{aligned}$$

Notice that the contributions of b to the individual observations x_i and the mean \bar{x} cancel out. This does not happen for cosine similarity.

We have now clarified this point in the Methods section of the manuscript:

“For each ROI r , and each predictor i in β 1-30, GLM5.1 produced a vector of regression weights capturing the effect of β_i at each voxel v in r . Each vector of regression-weights constituted a multivariate neural representation of a reward-option. We then quantified pairwise distances between multivariate representations using cosine dissimilarity – i.e., angular difference between two vectors. We used cosine dissimilarity instead of standard parametric (e.g. Pearson’s R) and non-parametric (e.g. Kendall’s Tau) methods to avoid artefacts arising from univariate differences in activity (see Discussion for further details). Because cosine dissimilarity measures the angle between two vectors, it remains intact if one vector is scaled, hence it is insensitive to the magnitude of the vectors.”

Concrete points:

1. Since both the Pearson correlation and cosine similarity are unaffected by scaling, the rationale for choosing one over the other given in the methods (lines 1474-1478) is not meaningful. It should be removed.
2. Unless univariate changes in BOLD imply multiplicative changes in multivariate activity in the context of this specific analysis, the section of the discussion stating that the use of cosine (dis)similarity ensures that the RSA results are not a spurious consequence of univariate activity (lines 595-603) is not correct. In that case, it should also be removed.
3. If there is a concern that the RSA results may be an artifact of changes in univariate activity (lines 594-597), the authors should show that switching from cosine similarity to Pearson correlation does not affect their conclusions. (I once had to do this myself, unfortunately.) Even if this does not materially impact the present findings, as will hopefully be the case, I would encourage the authors to replace cosine similarity with Pearson correlation in the manuscript so that its readers do not use the former.

6. Cortical option representations

I missed that the statistics for H1 were in Fig. S5, and I appreciate that the authors have addressed my concern by highlighting this result in the main text.

7. ⊗ References

I appreciate that the authors have made an effort to incorporate additional references and discussion, especially given limited space.

At the same time, I do think it is an issue that the authors state in the introduction that “little is known about what DRN activity represents, and how DRN function differs from other neuromodulatory nuclei in the brainstem, midbrain, and basal forebrain” without citing or discussing any papers that directly compare DRN and MBD activity: Cohen et al., *eLife*, 2015; Zhong et al., *J Neurosci*, 2017; Cardozo-Pinto, *Nature*, 2024; Batten et al., *Nat Hum Behav*, 2024; the work of Matias et al. 2017 is cited, but only in the discussion on a separate point.